# FINE-TUNING CAN DISTORT PRETRAINED FEATURES AND UNDERPERFORM OUT-OF-DISTRIBUTION

**Ananya Kumar, Aditi Raghunathan, Robbie Jones, Tengyu Ma, Percy Liang**
Stanford University, Computer Science Department

## ABSTRACT

When transferring a pretrained model to a downstream task, two popular methods are full fine-tuning (updating all the model parameters) and linear probing (updating only the last linear layer—the "head"). It is well known that fine-tuning leads to better accuracy in-distribution (ID). However, in this paper, we find that fine-tuning can achieve worse accuracy than linear probing out-of-distribution (OOD) when the pretrained features are good and the distribution shift is large. On 10 distribution shift datasets (BREEDS-Living17, BREEDS-Entity30, DomainNet, CIFAR → STL, CIFAR-10.1, FMoW, ImageNetV2, ImageNet-R, ImageNet-A, ImageNet-Sketch), fine-tuning obtains on average 2% higher accuracy ID but 7% lower accuracy OOD than linear probing. We show theoretically that this tradeoff between ID and OOD accuracy arises even in a simple setting: fine-tuning overparameterized two-layer linear networks. Our analysis suggests that the easy two-step strategy of linear probing then full fine-tuning (LP-FT), sometimes used as a fine-tuning heuristic, combines the benefits of both fine-tuning and linear probing. Empirically, LP-FT outperforms both fine-tuning and linear probing on the above datasets (1% better ID, 10% better OOD than full fine-tuning).

## 1 INTRODUCTION

Pretraining a model on a large dataset before transferring to a downstream task's training data substantially improves accuracy over training from scratch—for example, pretraining a ResNet-50 on unlabeled ImageNet boosts accuracy on CIFAR-10 from 94% to 98% (Chen et al., 2020a;b). High-stakes applications such as poverty mapping in under-resourced countries (Jean et al., 2016), self-driving cars (Yu et al., 2020), and medical diagnosis (AlBadawy et al., 2018), require models that also generalize to circumstances not seen in the training distribution. In addition to testing on data drawn from the downstream task's training distribution (in-distribution; ID), it is increasingly important to test on data distributions unseen during training (out-of-distribution; OOD).

After initializing with a pretrained model, two popular transfer methods are fine-tuning (running gradient descent on all the model parameters), and linear probing (tuning the head but freezing lower layers). In the ID setting it is well known that fine-tuning leads to better accuracy than linear probing (Kornblith et al., 2019; Zhai et al., 2020; He et al., 2020), and even when testing OOD, prior work usually fine-tunes all parameters of their model (Hendrycks et al., 2019a; Miller et al., 2021; Andreassen et al., 2021). Intuitively, fine-tuning all layers of a network can improve pretrained features by adapting them to the specific task, while linear probing freezes these features.

In this work, we investigate the OOD accuracy of fine-tuning and linear probing and find that surprisingly, fine-tuning can do *worse* than linear probing in the presence of a large distribution shift. We experiment on ten distribution shift benchmarks (BREEDS Living17, BREEDS Entity30, DomainNet, CIFAR → STL, CIFAR10.1, FMoW Geo-shift, ImageNetV2, ImageNet-R, ImageNet-A, ImageNet-Sketch), initializing with good pretrained features from MoCo-v2 (Chen et al., 2020b) and CLIP (Radford et al., 2021). While both methods offer gains over training from scratch, fine-tuning improves the average ID accuracy relative to linear probing from 83% to 85% but brings down the OOD accuracy from 66% to 59% (Figure 1).

When and why does fine-tuning underperform linear probing? We theoretically consider fine-tuning a two-layer linear network in an overparameterized regression setting where the feature extractor layer has been pretrained to map high-dimensional inputs to useful, lower-dimensional, features. We prove that fine-tuning is worse than linear probing on directions outside the span of the training data when using "good" pretrained features. Even with an infinitesimally small learning rate, fine-tuning distorts pretrained features—the features of ID training data are updated while those of OOD data

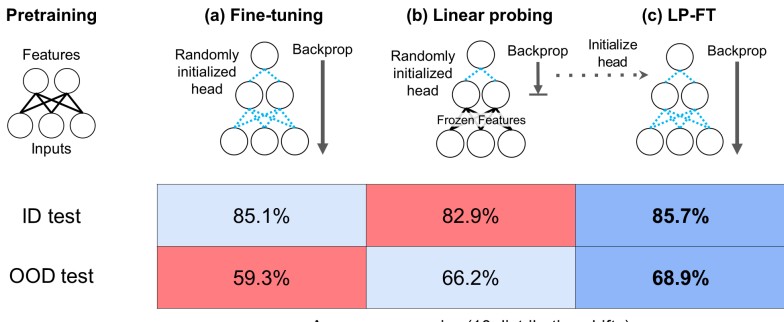

Figure 1: Given a good feature extractor (top-left), a randomly initialized head is added to map features to outputs and we can (a) fine-tune all the model parameters or (b) linear probe, which freezes the feature extractor and trains only the head. We run experiments on ten distribution shifts. Fine-tuning does well when the test example is sampled from the fine-tuning distribution (ID), but can underperform on test examples sampled from OOD distributions (when the distribution shift is large). (c) Our theory indicates that fine-tuning can distort the pretrained feature extractor and lead to poor OOD accuracy, but initializing with a linear probed head can fix this—empirically LP-FT gets better accuracies both ID and OOD.

change less. Since the head and feature extractor are simultaneously optimized during fine-tuning to a configuration that works well on ID training data, the head only accomodates the distorted features of ID points and performs poorly (relative to linear probing) on the less changed features of OOD points. Interestingly, we show that this feature distortion issue cannot be simply fixed by early stopping—throughout the entire process of fine-tuning, we never pass through parameters that do well OOD (relative to linear probing). On the other hand, given "good" features, linear probing extrapolates better OOD because it preserves pretrained features, but does worse than fine-tuning ID because linear probing cannot adapt the features to the downstream task.

**Technical challenges.** Existing theoretical work on transfer learning focuses on linear probing (Wu et al., 2020; Tripuraneni et al., 2020; Du et al., 2020). In contrast, analyses of fine-tuning is scarce and challenging because it requires understanding the training dynamics, instead of only the loss function and its global minimizers. In fact, fine-tuning and training from scratch optimize the *same* training loss and only differ in their initializations (pretrained vs random). A mathematical analysis that distinguishes them needs to capture properties of the different minima that these algorithms converge to, a phenomenon that is sometimes theoretically referred to as the implicit regularization effect of initialization (Neyshabur et al., 2014). Accordingly, our analysis reasons about the parameters that gradient methods pass through starting from the pretrained initialization, which is challenging because this is a non-convex optimization problem and there is no known closed form for this trajectory. Two-layer linear networks are widely studied in the literature on implicit regularization (Saxe et al., 2014; Gunasekar et al., 2017; Gidel et al., 2019; Arora et al., 2018). However, they analyze random and often small initializations, which don't capture pretraining.

**Algorithmic implications.** Our theory shows that fine-tuning underpeforms because when trying to fit ID training data with a randomly initialized head, the feature extractor changes significantly for ID examples, making features for ID and OOD examples largely inconsistent. This can be fixed by initializing with a good head that does not need to be updated much during fine-tuning, reducing how much the feature extractor changes. This suggests a simple two-step strategy of first linear probing to find a good head and then full fine-tuning (LP-FT). *Empirically, LP-FT outperforms fine-tuning and linear probing, both ID and OOD*. Even on CIFAR-10.1 (small distribution shift), where fine-tuning is better for both ID and OOD, we find LP-FT outperforms fine-tuning on both metrics. LP-FT and vanilla fine-tuning use similar amounts of compute because the first step of linear probing is relatively very cheap. Prior work has used LP-FT (Levine et al., 2016; Kanavati & Tsuneki, 2021) (or variants such as layerwise fine-tuning (Howard & Ruder, 2018) or larger learning rates for the head layer (Prabhu et al., 2021))—however it has not been used for robustness / OOD accuracy, and we show that it addresses the ID-OOD tradeoff theoretically and empirically. Note that LP-FT is not meant to be a SOTA method but rather a simple, principled way to get good ID and OOD accuracy—we hope our analysis inspires even better methods for robust fine-tuning.

**Empirical validation.** Finally, we find that fine-tuning fails and LP-FT works, for the reasons predicted by our feature distortion theory: (1) fine-tuning changes the features for ID examples more than for OOD examples, leading to distortions; (2) LP-FT indeed changes both ID and OOD features $10-100\times$ less than fine-tuning does; (3) LP-FT gets the best of both worlds, achieving better accuracies than fine-tuning and linear probing both ID and OOD (Figure 1).

## 2 SETUP

**Task and evaluation.** Given training examples sampled from some distribution $P_{\mathsf{id}}$, our goal is to learn a predictor $f : \mathbb{R}^d \to \mathcal{Y}$ to map inputs $x \in \mathbb{R}^d$ to outputs $y \in \mathcal{Y}$. We evaluate predictors on their standard "in-distribution" (ID) performance $L_{\mathsf{id}}$ on new test samples drawn from $P_{\mathsf{id}}$ that the training data is also sampled from. We also evaluate classifiers on their "out-of-distribution" (OOD) performance $L_{\mathsf{ood}}$ on test samples drawn from a new distribution $P_{\mathsf{ood}}$ that is different from $P_{\mathsf{id}}$. Formally, for some loss function $\ell$, we evaluate classifiers on:

$$L_{\mathsf{id}}(f) = \mathbb{E}_{(x,y)\sim P_{\mathsf{id}}}[\ell(f(x),y)] \text{ and } L_{\mathsf{ood}}(f) = \mathbb{E}_{(x,y)\sim P_{\mathsf{ood}}}[\ell(f(x),y)]. \tag{2.1}$$

**Models.** In this work, we focus on predictors that leverage pretrained representations. We parameterize the final predictor $f$ as follows: given features $g_B(x) \in \mathbb{R}^k$ for some feature extractor parameters $B \in \mathcal{B}$, and a linear "head" $v \in \mathcal{V}$, we have $f_{v,B}(x) = v^\top g_B(x)$. In our experiments (Section 4), $g_B$ is a deep network and in our theory (Section 3), $g_B$ is a linear projection.

We assume access to some initial pretrained feature extractor $B_0$ that is obtained by training on potentially large amounts of data from a distribution that contains unlabeled or weakly supervised $x$ inputs from $P_{\mathsf{id}}$ and $P_{\mathsf{ood}}$. We focus on two popular methods to learn a predictor $f_{v,B}$ given training data from $P_{\mathsf{id}}$: (i) linear probing where $B = B_0$ and the linear head is obtained by minimizing some loss (e.g., logistic loss for classification, squared loss for regression) on the training data, and (ii) fine-tuning where both $v$ and $B$ are updated by performing gradient descent on some loss on the training data with $B$ initialized at $B_0$.

## 3 THEORY: FINE-TUNING DISTORTS PRETRAINED FEATURES

Our goal is to understand under what conditions fine-tuning does worse than linear probing out-of-distribution (OOD). We consider a linear setting (feature extractor $g_B$ is linear) where the pretrained features are "good" and the OOD shift is large (Section 3.1). We prove our main result: that fine-tuning, in which all model parameters are updated, distorts features and gets suboptimal OOD error (Section 3.2, Theorem 3.2). We use this result to show that linear probing gets better OOD error but worse ID error than fine-tuning (Section 3.3). Finally, we explain why linear probing then fine-tuning can mitigate this ID-OOD tradeoff (Section 3.4).

Our analysis handles two key challenges which distinguishes it from prior work on transfer learning in linear models (Wu et al., 2020; Tripuraneni et al., 2020; Du et al., 2020; Xie et al., 2021a). Prior work focuses on linear probing, while we study fine-tuning where the resulting optimization problem is *non-convex*. We also study *overparameterized models* where the training loss alone does not determine test performance—this captures the fact that both training neural networks from scratch and fine-tuning them have the same training loss but very different test performance. However, it also makes the analysis challenging because we need to reason about the trajectory of gradient methods starting from a pretrained initialization, which has no known closed form.

### 3.1 LINEAR OVERPARAMETERIZED SETTING

For our analysis, we focus on regression, where $\mathcal{Y} = \mathbb{R}$ and $\ell(\widehat{y},y) = (\widehat{y}-y)^2$ is the squared loss.

**Models.** Recall from Section 2 that we parameterize predictors in terms of the feature extractor and head parameters. In this section, we study models where the feature extractor is linear, i.e. $f_{v,B}(x) = v^\top B x$ where $B \in \mathcal{B} = \mathbb{R}^{k \times d}$, and $v \in \mathcal{V} = \mathbb{R}^k$.

**Good pretrained features.** For simplicity, we assume the models are well-specified i.e. $y = v_\star^\top B_\star x$ where $v_\star \in \mathbb{R}^k$ and $B_\star \in \mathbb{R}^{k \times d}$. [1] Note that $B_\star$ and $v_\star$ are only unique up to rotations, i.e., for any rotation matrix $U$, $(Uv_\star)^T(UB_\star)x = v_\star^T B_\star x$. As in prior work (Tripuraneni et al., 2020) suppose $B_\star$

---

[1]Our main contribution, analysis of fine-tuning (Theorem 3.2), does not require well-specification. We compare FT with LP by adapting earlier work on linear probing which requires well-specification.

and $B_0$ have been orthogonalized to have orthonormal rows. Suppose we have a pretrained feature extractor $B_0$ close to $B_\star$, so $d(B_0, B_\star) \leq \epsilon$ where the distance $d$ is defined as (where the min is over rotation matrices $U \in \mathbb{R}^{k \times k}$):

$$d(B, B') = \min_U \| B - UB' \|_2. \tag{3.1}$$

**Training data.** Let $X \in \mathbb{R}^{n \times d}, X \neq 0$ be a matrix encoding $n$ training examples from $P_{\text{id}}$ where each of the $n$ rows is a training input. Let $Y \in \mathbb{R}^n$ be the corresponding outputs. Let $S = \text{rowspace}(X)$ be the $m$-dimensional subspace spanning the training examples. We consider an overparameterized setting where $1 \leq m < d - k$. Intuitively, the input dimension $d$ is high (e.g., 10K), feature dimension $k$ is lower (e.g., 100) and $m$ is in the middle (e.g., 5K).

**Large OOD shift.** We assume that the OOD data contains examples outside the span of the training data. Formally, let $P_{\text{ood}}$ have second moment $\Sigma = \mathbb{E}[xx^\top]$ where $x \sim P_{\text{ood}}$, for invertible $\Sigma$.

**Training methods.** Given training data and a pretrained feature extractor $B_0$, we study the two popular methods of linear probing (LP) and fine-tuning (FT) to learn the final predictor. Both methods involve optimizing the training loss via gradient descent (or variants). In order to effectively analyze these gradient based algorithms, we study vanishing step sizes leading to gradient flows. Gradient flows can be thought of as a continuous time analogue of gradient based methods and have been extensively studied in recent years as a way to understand gradient based methods (Gunasekar et al., 2017; Arora et al., 2018; Du et al., 2018). Formally, for training loss $\widehat{L}(v, B) = \| XB^\top v - Y \|_2^2$, the gradient flow differential equations for LP and FT are as follows:

$$\partial_t v_{\text{ft}}(t) = -\nabla_v \widehat{L}(v_{\text{ft}}(t), B_{\text{ft}}(t)), \ \partial_t B_{\text{ft}}(t) = -\nabla_B \widehat{L}(v_{\text{ft}}(t), B_{\text{ft}}(t)), \tag{3.2}$$

$$\partial_t v_{\text{lp}}(t) = -\nabla_v \widehat{L}(v_{\text{lp}}(t), B_0), \ \partial_t B_{\text{lp}}(t) = 0, \tag{3.3}$$

initialized with $B_{\text{ft}}(0) = B_{\text{lp}}(0) = B_0$ and $v_{\text{ft}}(0) = v_{\text{lp}}(0) = v_0$. In practice, the head parameter $v_0$ is initialized randomly—our results hold for any standard random initialization (Glorot & Bengio, 2010), for example $v_0 \sim \mathcal{N}(0, \sigma^2 I)$ for any $\sigma^2$, or zero initialization where $v_0 = 0$. Recall that the initial value of the feature extractor $B_0$ is obtained via pretraining.

The final LP and FT solutions are the limit points of the corresponding gradient flows:

$$v_{\text{ft}}^\infty = \lim_{t \to \infty} v_{\text{ft}}(t) \text{ and } B_{\text{ft}}^\infty = \lim_{t \to \infty} B_{\text{ft}}(t), \tag{3.4}$$

$$v_{\text{lp}}^\infty = \lim_{t \to \infty} v_{\text{lp}}(t) \text{ and } B_{\text{lp}}^\infty = \lim_{t \to \infty} B_{\text{lp}}(t) = B_0. \tag{3.5}$$

### 3.2 FINE-TUNING DISTORTS PRETRAINED FEATURES

The more common method of using a pretrained feature extractor is fine-tuning (FT) which typically improves ID performance relative to linear probing (LP). In this section, we show that FT can distort features leading to poor OOD performance. We first explain the key intuitions and then present our formal theorem lower bounding the OOD error of FT (Section 3.2.2).

#### 3.2.1 KEY INTUITIONS

We use two main observations to characterize when and why FT has higher OOD error than LP.

*1. Features get distorted: representations change only in the ID subspace (i.e., subspace spanned by the training data) and are unchanged in the orthogonal subspace.* To see this, we take the derivative of the training loss $\widehat{L}(v, B) = \| XB^\top v - Y \|_2^2$ with respect to the feature extractor parameter $B$:

$$\nabla_B \widehat{L}(v, B) = 2v(Y - XB^\top v)^\top X. \tag{3.6}$$

By definition, if $u$ is a direction orthogonal to the training subspace $S = \text{rowspace}(X)$, then $\nabla_B \widehat{L}(v, B)u = 0$, that is the gradient updates to $B$ do not modify $Bu$ for $u \in S^\perp$. However, the gradient is non-zero for directions $u$ in the ID subspace and the corresponding features $Bu$ change across the fine-tuning process. We call this feature distortion: the features in some directions are changed but not others. Next, we explain why this can lead to high OOD error.

*2. Distorted features can lead to higher OOD error.* Consider a toy example (Figure 2) where $d = 2$ and the dimensionality of the representations $k = 1$. The linear head $v$ is a scalar quantity that denotes how much the feature extractor $B$ has to be scaled by. Suppose the ID-subspace is the $x$-axis. There are different ways of fitting the ID subspace depending on the feature extractors $B$ as

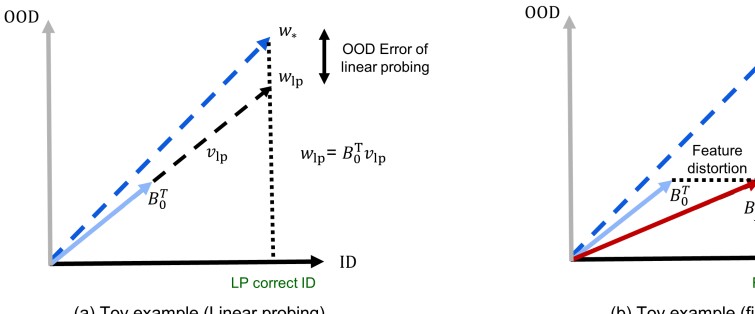

(a) Toy example (Linear probing)   (b) Toy example (fine-tuning)

Figure 2: A toy version of our theory illustrating why fine-tuning distorts features, with inputs in 2D. Given input $x$, the ground truth output is $y = w_\star^\top x$. The ID data is along the $x$-axis and the pretrained feature extractor is $B_0$. (a) Linear probing learns $w_{\mathsf{lp}}$, a scaling of the pretrained feature extractor that gets the ID data correct ($w_{\mathsf{lp}}$ and $w_\star$ have the same $x$ coordinate as indicated by the vertical dotted line). (b) Fine-tuning updates the pretrained feature extractor along the ID data (so horizontally) to get $B_{\mathsf{ft}}$, and then learns a scaling of these features that gets the ID data correct. While both methods get ID data correct, fine-tuning makes large errors perpendicular to the ID data, because fine-tuning updates $B_0$ along the ID direction but not the perpendicular direction.

shown in the Figure—both fine-tuned and linear probed estimators match the true parameter in the ID subspace (since $w_{\mathsf{lp}}, w_{\mathsf{ft}}, w_\star$ have the same projection on the $x$-axis). If the feature extractor were optimal or scaled versions of the optimal, good performance on the ID subspace would translate to good performance everywhere, even in directions orthogonal to the ID subspace. However, in FT, the features change only for inputs in the ID subspace (see (1)) and thus the updated features are *not* simply scaled but distorted. In Figure 2, this corresponds to the feature extractor $B_0$ changing along the $x$-axis. In this case even if the ID error is low, error in directions orthogonal to the ID subspace can be high, leading to high OOD error.

The only way the pretrained features are not distorted and only scaled during FT is if the initial feature extractor $B_0$ is exactly aligned with the ID subspace. In Figure 2, if $B_0$ is along the $x$-axis (the ID subspace), then updating the features exclusively along the $x$-axis would simply scale the initial features. In this case linear probing and fine-tuning will have identical behavior. However, if the angle between $B_0$ and the $x$-axis is non-zero, the updates would lead to distortions. In high dimensions, we measure the alignment between $B_0$ and the ID subspace with the largest principal angle:

**Definition 3.1** (largest principal angle). *Let $A$ and $B$ be arbitrary subspaces, and $E$ and $F$ be matrices with orthonormal columns that span $A$ and $B$ respectively, with $r = \min(\dim(A), \dim(B))$. Then $\cos\theta_{\mathsf{max}}(A,B) = \sigma_r(E^\top F)$, which is the $r$-th largest singular value of $E^\top F$.*

### 3.2.2 GENERAL RESULT ON THE OOD ERROR OF FINE-TUNING

Our main theorem lower bounds the OOD error of fine-tuning outside the span of the training data.

**Theorem 3.2.** *In the overparameterized linear setting, let $S^\perp = \mathrm{rowspace}(X)^\perp$, $R_0 = \mathrm{rowspace}(B_0)$, and $v_\star, B_\star$ be the optimal parameters with $w_\star = B_\star v_\star$. If $\cos\theta_{\mathsf{max}}(R_0, S^\perp) > 0$, then for all time steps $t$, the OOD error of the fine-tuning iterates $(B_{\mathsf{ft}}(t), v_{\mathsf{ft}}(t))$ is lower bounded:*

$$\sqrt{L_{\mathsf{ood}}(v_{\mathsf{ft}}(t), B_{\mathsf{ft}}(t))} \geq \sqrt{\sigma_{\mathsf{min}}(\Sigma)} \left( \frac{\cos\theta_{\mathsf{max}}(R_0, S^\perp)}{\sqrt{k}} \frac{\min(\varphi, \varphi^2/\|w_\star\|_2)}{(1+\|w_\star\|_2)^2} - \epsilon \right), \qquad (3.7)$$

*where $\varphi^2 = |(v_0^\top v_\star)^2 - (v_\star^\top v_\star)^2|$ is defined to be inital head alignment error and $\epsilon \geq d(B_0, B_\star)$ is the error in the pretrained feature extractor.*

**Proof sketch.** Since the features do not change for examples in $S^\perp$ (perpendicular to the training data), we show that in order to achieve low error on $S^\perp$ the linear head $v_{\mathsf{ft}}(t)$ would have to become very similar to the optimal $v_\star$ at some time $t$. The head initialization $v_0$ is random (or zero) and likely to be far from $v_\star$ (measured by the alignment error $\varphi$), so the head would have to change a lot to get close to $v_\star$. As we see from the fine-tuning gradient flow (3.2), $v_{\mathsf{ft}}(t)$ and $B_{\mathsf{ft}}(t)$ change in a "coupled" manner, and a "balancedness" invariant in Du et al. (2018) holds across the fine-tuning trajectory. Correspondingly, if $v_{\mathsf{ft}}(t)$ changes a lot and gets close to $v_\star$, the features $B_{\mathsf{ft}}(t)$ also change a lot for

examples in $S$—we show that this would lead to high error on examples in $S$. Either way, fine-tuning would get some subspace ($S$ or $S^\perp$) of examples wrong, leading to high OOD error. The full proof appears in Appendix A.

**Interpretations of various quantities.** *Quality of pretrained features ($\epsilon$).* To unpack the bound consider a special case where the pretrained features are perfect ($\epsilon = 0$). With perfect features, Proposition A.21 shows that linear probing gets zero OOD error. Theorem 3.2 shows that $L_{\text{ood}}(v_{\text{ft}}(t), B_{\text{ft}}(t)) > 0$ at all times $t$—so fine-tuning underperforms when the features are perfect.

*Alignment error of random head initialization ($\varphi^2$).* The lower bound (Equation A.14) increases as $\varphi^2$ increases, because the gradient updates to the head and feature extractor are coupled. If the head were somehow initialized perfectly at $v_\star$, fine-tuning updates may not increase the OOD error. However, when the head is randomly initialized as is standard in fine-tuning, the alignment error is high, leading to high OOD error. We use this insight in Section 3.4 to show that better head initialization (via linear probing) improves OOD performance of fine-tuning.

### 3.3 LINEAR PROBING VS. FINE-TUNING

In this section, we use our main theorem on fine-tuning (Theorem 3.2) and adapt prior work on linear probing to show that linear probing is better than fine-tuning OOD, but worse ID, when the ID distribution has density on a lower $m < d$ dimensional subspace $S$, and $B_0$ is close to $B_\star$.

**Assumption 3.3** (ID subspace assumption). *We assume that the ID data lies on an $m$-dimensional subspace $S$ where $k < m < d - k$, and we have $n \geq m$ training examples. Formally, let $P_z$ be a distribution on $\mathbb{R}^m$ which has density, and let the columns of $F \in \mathbb{R}^{d \times m}$ form an orthonormal basis for $S$. Then $P_{\text{id}}$ has the distribution of $Fz$ where $z \sim P_z$.*

Recall that the ID error is the expected mean-squared error over the ID distribution $P_{\text{id}}$:

$$L_{\text{id}}(v, B) = \mathop{\mathbb{E}}_{x \sim P_{\text{id}}} [(v_\star^\top B_\star x - v^\top B x)^2] \tag{3.8}$$

**OOD comparison**: Under mild non-degeneracy conditions, we show that as the feature extractor error $\epsilon$ goes to $0$, linear probing does much better than fine-tuning OOD: the ratio of the losses goes to $0$. The non-degeneracy conditions are similar to Section 3.2—we require that the training data cannot be exactly in the same direction or orthogonal to the pretrained features, formally that $\cos\theta_{\text{max}}(R_*, S)$ and $\cos\theta_{\text{max}}(R_*, S^\perp)$ are not $0$ where $R_* = \text{rowspace}(B_\star)$.

**Theorem 3.4** (Informal version of Theorem A.9). *In the linear overparameterized setting, under the ID subspace assumption (Assumption 3.3), if $\cos\theta_{\text{max}}(R_*, S) \neq 0$ and $\cos\theta_{\text{max}}(R_*, S^\perp) \neq 0$ where $R_* = \text{rowspace}(B_\star)$, then,*

$$\frac{L_{\text{ood}}(v_{\text{lp}}^\infty, B_0)}{L_{\text{ood}}(v_{\text{ft}}(t), B_{\text{ft}}(t))} \xrightarrow{p} 0, \text{as } B_0 \to B_\star. \tag{3.9}$$

*This holds for all times $t$ for FT (and therefore also for the limit $v_{\text{ft}}^\infty, B_{\text{ft}}^\infty$) and the LP iterates converge to $v_{\text{lp}}^\infty, B_0$ as a result of the gradient flow on a convex problem.*

Intuitively, if the pretrained features are good, LP learns a near optimal linear head which has small OOD error (Lemma A.15) but fine-tuning has high OOD error (Theorem 3.2). We give a more formal version of Theorem 3.4 and a proof in Appendix A.3.

**ID comparison**: When the pretrained features have some error, we show that fine-tuning does better than linear probing ID because fine-tuning can update the features to fit the ID data. The non-degeneracy condition on $R_{\text{aug}}$ below is similar to our previous results, and holds with probability $1$ if the ID subspace is chosen randomly, from Lemma A.17.

**Proposition 3.5.** *In the linear overparameterized setting, under the ID subspace assumption (Assumption 3.3), let $R_0 = \text{rowspace}(B_0)$, and $R_{\text{aug}} = \text{Span}(\{w_\star\} \cup R_0)$. Suppose $w_\star \notin R_0$, $\cos\theta_{\text{max}}(S, R_{\text{aug}}) \neq 0$, and that fine-tuning converges to a local minimum of its loss, then fine-tuning does better ID almost surely: $L_{\text{id}}(v_{\text{ft}}^\infty, B_{\text{ft}}^\infty) < L_{\text{id}}(v_{\text{lp}}^\infty, B_0)$ with probability $1$ (over the randomness of the training examples).*

To summarize, we proved that there are tradeoffs between ID and OOD error: FT has lower ID error but higher OOD error than LP. In the next section, we extend our theoretical insights to illustrate why a simple variant of FT may mitigate such tradeoffs.

### 3.4 LINEAR PROBING THEN FINE-TUNING: A SIMPLE VARIANT TO MITIGATE TRADEOFFS

The advantage of fine-tuning is it can adapt the feature extractor to fit the downstream task. Can we keep this benefit while ensuring that our OOD error is low when we have good pretrained features?

Going back to Theorem 3.2, we see that the alignment error in the head initialization $\varphi^2 = |(v_0^\top v_\star)^2 - (v_\star^\top v_\star)^2|$ plays an important role. The issue with FT was that under random or zero initialization, $\varphi^2$ is usually large and since the gradient updates to the feature extractor parameter are coupled with that of the head parameter, the features get distorted in a manner that increases the OOD error. This suggests that we should use a better head initialization—one obtained from linear probing. If the pretrained features are decent, a linear probed head would be much better aligned with $v_\star$ allowing the features to be updated in a manner that does not increase the OOD error much.

We formally prove this intuition in a simple setting where we have perfect pretrained features. Note that in this case, linear probing alone gets zero OOD error—so Proposition 3.6 is just a first cut result to illustrate that if initialized well, full fine-tuning does not distort features.

**Proposition 3.6.** *Given perfect pretrained features* $B_0 = UB_\star$ *for some rotation* $U$. *Let* $R_0 = rowspace(B_0)$. *Under the non-degeneracy conditions* $\cos\theta_{\mathsf{max}}(R_0,S) \neq 0$, $\cos\theta_{\mathsf{max}}(R_0,S^\perp) \neq 0$:

$$\forall t, L_{\mathsf{ood}}(B_{\mathsf{ft}}(t)^\top v_{\mathsf{ft}}(t)) > 0, \textit{ if } v_0 \sim \mathcal{N}(0,\sigma^2 I) \textit{ is randomly initialized (FT)}, \quad (3.10)$$

$$\forall t, L_{\mathsf{ood}}(B_{\mathsf{ft}}(t)^\top v_{\mathsf{ft}}(t)) = 0, \textit{ if } v_0 \textit{ is initialized to } v_{\mathsf{lp}}^\infty \textit{ (LP-FT)}. \quad (3.11)$$

## 4 EXPERIMENTS

We run experiments on ten benchmark datasets with deep neural networks and see that given good pretrained features, fine-tuning (FT) does better ID but worse OOD than linear probing (LP). As predicted by the theory, we find that LP-FT does better than both methods. Finally, we see that a number of predictions from the feature distortion theory hold up in practice. For more details on datasets, pretraining models, and experiment protocols, see Appendix B.

We use standard distribution shift datasets: **DomainNet** (Peng et al., 2019; Tan et al., 2020), **BREEDS-Living-17** (Santurkar et al., 2020), **BREEDS-Entity-30** (Santurkar et al., 2020), **CIFAR-10 → STL** (Krizhevsky, 2009; Coates et al., 2011; French et al., 2018), **CIFAR-10 → CIFAR-10.1** (Recht et al., 2018), **ImageNet-1K** (Russakovsky et al., 2015)—where the OOD test sets are **ImageNetV2** (Recht et al., 2019), **ImageNet-R** (Hendrycks et al., 2020), **ImageNet-A** (Hendrycks et al., 2019b), and **ImageNet-Sketch** (Wang et al., 2019)—, and **FMoW Geo-shift** which is adapted from the satellite remote sensing dataset *Functional Map of the World* (Christie et al., 2018; Koh et al., 2021). See Appendix B for more details on the datasets.

**Pretraining and models.** We use a CLIP pretrained ViT-B/16 for ImageNet. For the other datasets we use a ResNet-50 architecture and consider a diverse range of pretraining methods and datasets: MoCo-v2 (Chen et al., 2020b), CLIP (Radford et al., 2021), and MoCo-TP (Ayush et al., 2020). In Appendix B, we also show results for a CLIP-ViT-B/16 and more fine-tuning baselines on Living-17.

### 4.1 LINEAR PROBING VS FINE-TUNING

**Experiment protocols.** We initialize with the pretrained model, and fine-tune or linear probe on ID training examples. For fine-tuning on each dataset we swept over 6 learning rates, using a cosine learning rate schedule and batch size of 64. We early stop and choose the best learning rate using ID validation accuracy. For linear probing we train an $\ell_2$-regularized logistic regression classifier on frozen features from the penultimate layer of the pretrained model, selecting the best $\ell_2$-regularization hyperparameter based on ID validation accuracy. For all methods, we run each hyperparameter configuration 3 times (with different random seeds), and take the average accuracy. We used a slightly different protocol for ImageNet because the dataset is much larger and running these experiments involves more computational resources: we used a batch size of 128, swept over 3 learning rates for both fine-tuning and linear probing (we did not sweep over $\ell_2$-regularization), and ran each hyperparameter configuration once. In all cases, OOD data was only used for evaluation.

**Results.** Fine-tuning (FT) does better than linear probing (LP) on 5 out of 6 ID datasets (average accuracy of 85.1% for FT vs. 82.9% for LP, see Table 1). This is consistent with prior work and intuitions. However, linear probing does better on 8 out of 10 OOD datasets (average accuracy of 66.2% for LP vs. 59.3% for FT, see Table 2)—LP does better on all datasets except CIFAR-10.1 and ImageNetV2, where the OOD is designed to closely replicate the ID dataset. This matches

Table 1: **ID accuracies** with 90% confidence intervals over 3 runs—fine-tuning does better than linear probing on all datasets except DomainNet (which could be because the version of the DomainNet training dataset from Tan et al. (2020) is fairly small, with around 20K examples). LP-FT does the best on all except FMoW where it is in between linear probing and fine-tuning.

|  | CIFAR-10 | Ent-30 | Liv-17 | DomainNet | FMoW | ImageNet | Average |
|---|---|---|---|---|---|---|---|
| FT | **97.3 (0.2)** | **93.6 (0.2)** | 97.1 (0.2) | 84.5 (0.6) | **56.5 (0.3)** | **81.7 (-)** | 85.1 |
| LP | 91.8 (0.0) | 90.6 (0.2) | 96.5 (0.2) | 89.4 (0.1) | 49.1 (0.0) | 79.7 (-) | 82.9 |
| LP-FT | **97.5 (0.1)** | **93.7 (0.1)** | **97.8 (0.2)** | **91.6 (0.0)** | 51.8 (0.2) | **81.7 (-)** | **85.7** |

Table 2: **OOD accuracies** with 90% confidence intervals over 3 runs. Linear probing does better than fine-tuning on all datasets except CIFAR-10.1 and ImageNetV2, where the ID and OOD are similar (consistent with our theory). LP-FT does the best on all 10 datasets.

|  | STL | CIFAR-10.1 | Ent-30 | Liv-17 | DomainNet | FMoW |
|---|---|---|---|---|---|---|
| FT | 82.4 (0.4) | 92.3 (0.4) | 60.7 (0.2) | 77.8 (0.7) | 55.5 (2.2) | 32.0 (3.5) |
| LP | 85.1 (0.2) | 82.7 (0.2) | **63.2 (1.3)** | 82.2 (0.2) | 79.7 (0.6) | **36.6 (0.0)** |
| LP-FT | **90.7 (0.3)** | **93.5 (0.1)** | 62.3 (0.9) | **82.6 (0.3)** | **80.7 (0.9)** | 36.8 (1.3) |

|  | ImNetV2 | ImNet-R | ImNet-Sk | ImNet-A | Average |
|---|---|---|---|---|---|
| FT | **71.5 (-)** | 52.4 (-) | 40.5 (-) | 27.8 (-) | 59.3 |
| LP | 69.7 (-) | 70.6 (-) | 46.4 (-) | 45.7 (-) | 66.2 |
| LP-FT | **71.6 (-)** | **72.9 (-)** | **48.4 (-)** | **49.1 (-)** | **68.9** |

our theoretical predictions. Our training datasets vary in size from 20K examples to over a million examples, so LP does not appear to perform better than FT simply because of a small training set.

## 4.2 LINEAR PROBING THEN FINE-TUNING (LP-FT)

**Experiment protocols.** For LP-FT, we initialize the neural network head using the linear probed solution, and then fine-tune the model. LP-FT and fine-tuning use similar compute because the linear probing step is much faster than fine-tuning. As with fine-tuning, we swept over 6 learning rates, early stopping using ID validation accuracy. For the ImageNet experiments we swept over 3 learning rates, and explicitly ensured that LP-FT and fine-tuning use exactly the same compute (we ran each stage of LP-FT for half as many epochs as we ran vanilla fine-tuning).

**Results.** We find that LP-FT gets the best accuracy ID (average: 85.7%) and OOD (average: 68.9%). This is true for 5/6 ID and 10/10 OOD datasets—every dataset except FMoW ID, where LP-FT is better than linear probing but worse than fine-tuning. Since the ID accuracy on FMoW is low (56.5%), this could be because the pretrained features are not good.

## 4.3 EXAMINING THE FEATURE DISTORTION THEORY

**Early stopping does not mitigate feature distortion.** Our theory predicts that fine-tuning can do worse OOD (than linear probing) throughout the process of fine-tuning, and not just at the end. To test this, we early stop each fine-tuning method and choose the best learning rate based on OOD test accuracy. As expected, fine-tuning does improve a little, but linear probing (average accuracy: 67.1%) is still better than fine-tuning (average accuracy: 61.3%). See Appendix B for per-dataset results.

**ID-OOD features get distorted from fine-tuning.** The feature distortion theory predicts that fine-tuning changes features for ID examples more than for OOD examples, which is why fitting a head on ID examples performs poorly OOD. To test this, for each example $x$ in Living-17 (results for other datasets are in Appendix B), we took the Euclidean distance of the ResNet-50 features before and after fine-tuning: $\|g_B(x) - g_{B_0}(x)\|_2$. As expected, the average distance for ID examples $(0.0188 \pm 0.0001)$ is more than for OOD examples $(0.0167 \pm 0.0001)$. The theory also predicts that LP-FT changes features less than fine-tuning does. As expected, the average distance changed by LP-FT both ID $(0.0011 \pm 0.0001)$ and OOD $(0.0009 \pm 0.0001)$ is $20\times$ smaller than for fine-tuning.

**Pretrained features must be good, ID-OOD far apart.** Our theory says that linear probing does better than fine-tuning OOD, but only if the OOD and ID data are quite different, and the pretrained features are good—otherwise fine-tuning can do better OOD by adjusting the feature extractor ID.

*Feature quality*: We use a checkpoint of MoCo-v1 that got 10% worse accuracy (on ImageNet) and compare linear probing and fine-tuning on Living-17. With worse features, both methods do worse, but fine-tuning (96% ID, 71% OOD) does better than linear probing (92% ID, 66% OOD).

*ID ≈ OOD*: We fine-tune / linear probe on CIFAR-10, and test on CIFAR-10.1, a dataset collected using a similar protocol to CIFAR-10. As expected, fine-tuning (92.3%) outperforms linear probing OOD (82.7%). Even in this case, where we have no tradeoffs, LP-FT does the best (93.5%).

## 5  RELATED WORK AND DISCUSSION

**Fine-tuning vs. linear probing.** Fine-tuning (FT) and linear probing (LP) are popular transfer learning algorithms. There is substantial evidence of FT outperforming LP in-distribution (ID) including recent large-scale investigations (Kornblith et al., 2019; Chen et al., 2021; Zhai et al., 2020; Chen et al., 2020b) (the only notable exception is in Peters et al. (2019) where LP performs better than FT when using ELMo representations, but worse using BERT). FT is therefore the method of choice for improving accuracy, while LP is used to analyze properties of representations (Peters et al., 2018; Belinkov et al., 2017; Hewitt & Manning, 2019). In our work, we find that FT can underperform LP especially when using high quality pretrained features in the presence of a large distribution shift. There are a variety of other fine-tuning heuristics (Ge & Yu, 2017; Guo et al., 2019; Zhang et al., 2020; Zhu et al., 2020; Jiang et al., 2021; Aghajanyan et al., 2021)—combining our insights with these ideas might lead to better methods.

**The benefit of preserving pretrained features.** Our work adds to growing evidence that *lightweight* fine-tuning, where only a small part of a pretrained model are updated, can perform better under distribution shifts—and we give a theoretical grounding to why this might be the case. Zero-shot language prompting in vision (Radford et al., 2021) and other lightweight fine-tuning approaches in NLP (Houlsby et al., 2019; Li & Liang, 2021; Xie et al., 2021b; Lester et al., 2021; Utama et al., 2021; Zhou et al., 2021) have been shown to improve OOD performance. Andreassen et al. (2021) observe that through the course of fine-tuning, ID accuracy increases but OOD accuracy plateaus.

**Mitigating ID-OOD tradeoffs.** While LP-FT has sometimes been used as a fine-tuning heuristic (Levine et al., 2016; Kanavati & Tsuneki, 2021; fastai), it has not been used for robustness / OOD accuracy, and we show that it addresses the ID-OOD tradeoff theoretically and empirically. Tradeoffs between ID and OOD accuracy are widely studied and prior work self-trains on large amounts of unlabeled data to mitigate such tradeoffs (Raghunathan et al., 2020; Xie et al., 2021a; Khani & Liang, 2021). In contrast, LP-FT uses no extra unlabeled data and is a simple variant of fine-tuning. In concurrent and independent work, Wortsman et al. (2021) show that ensembling the weights of a zero-shot and fine-tuned model mitigates the ID-OOD tradeoff between these approaches, and this method could be promising for our datasets as well.

**Theoretical analysis of transfer learning.** Prior works on transfer learning mainly analyze linear probing (Wu et al., 2020; Tripuraneni et al., 2020; Du et al., 2020). Recent works (Chua et al., 2021; Shachaf et al., 2021) study fine-tuning, but in the underparameterized regime (where there is a unique global optimum) or assuming a balanced initialization. Prior works also focus on ID error, while we analyze OOD error. See Section C for additional related work on theory of overparameterized models.

## 6  CONCLUSION.

There is a strong trend towards leveraging pretrained models to improve downstream performance, and whenever feasible, it is common to fine-tune all model parameters. In this work, we show theoretically and empirically that preserving features might be important for robustness, and simpler approaches like linear probing can improve out-of-distribution (OOD) performance. *This OOD gap between fine-tuning and linear probing grows as the quality of pretrained features improve, so we believe our results are likely to gain significance over time with growing innovations and scale of pretraining.*

Finally, we showed LP-FT can mitigate tradeoffs between ID and OOD accuracy in our context. LP-FT could be useful in other situations, for example in CLIP we could initialize the final layer with the zero-shot classifier and then fine-tune the entire model, as done in concurrent work (Wortsman et al., 2021). In NLP, linear probing is not as good—here we could first prompt-tune (Lester et al., 2021) and then fine-tune the entire model. LP-FT is just a first step in leveraging the intuition from our theoretical analysis and we hope that this work inspires new methods of leveraging powerful pretrained models.

**Proofs and Reproducibility**: We include proofs for our theoretical results in Appendix A and additional experiment details in Appendix B. Updated code is available at `https://github.com/AnanyaKumar/transfer_learning` and this CodaLab worksheet.

**Acknowledgements**: We would like to thank Kumar Ayush and Burak Uzkent for MoCo checkpoints pretrained on unlabeled FMoW images, Nilesh Tripuraneni for clarifications on his work and references on principal angles, Daniel Levy for useful suggestions on experiments to run, Niladri Chatterji, Jeff Z. HaoChen, and Colin Wei for useful papers and comments on figures, Niladri Chatterji and Kaidi Cao for reviewing the paper at ML paper swap, Kevin Yang for his help with analyzing differential equations, Tri Dao and Pang Wei Koh for help with writing, Suriya Gunasekar, Adam Kalai, Simon Kornblith, Ting Chen, Sang Michael Xie, Albert Gu, and Kendrick Shen for useful discussions, and Pang Wei Koh, Niladri Chatterji, and Tri Dao for suggestions on framing our results better.

Ananya Kumar was supported by the Rambus Corporation Stanford Graduate Fellowship. Percy Liang was supported by the Open Philanthropy Project and NSF Award Grant No. 1805310. Aditi Raghunathan was supported by a Google PhD Fellowship and Open Philanthropy Project AI Fellowship. Tengyu Ma acknowledges support of a Google Faculty Award, NSF IIS 2045685, the Sloan Fellowship, JD.com, SAIL, and SDSI.

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

# A PROOFS FOR SECTION 3

## A.1 PRELIMINARIES ON IMPORTANT NOTATIONS AND PRINCIPAL ANGLES

**Big-Oh Notation**: For convenience, we use big-oh notation in a way that differs from standard theoretical computer science texts. When we say $O(\text{<expr1>})$ we mean that this can be replaced by $c\,\text{<expr1>}$ for some universal constant such that the statement holds. As an example, we can say $5x^2 \leq O(x^2)$ because there exists some universal constant ($c = 5$) such that $5x^2 \leq 5x^2$. More examples: we can also say $5x^2 \geq O(x^2)$ or if $x \geq 1$ then $7x^2 \leq O(x^3)$ and $0.1x^2 \geq O(x)$.

**Singular Values**: Given a rectangular matrix $A \in \mathbb{R}^{m \times n}$, let $r = \min(m,n)$. The minimum singular value is defined as the $r$-th largest singular value of $A$, so $\sigma_{\min}(A) = \sigma_r(A)$.

Working with minimum singular values requires more care than maximum singular vectors. In particular, when we have rectangular matrices some bounds depend on whether the matrix is 'fat' (has more columns than rows) or 'tall' (has more rows than columns).

Given a matrix $A$, the operator norm $\|A\|_2$ is the maximum singular value: $\|A\|_2 = \sigma_{\max}(A)$.

**Projectors**: Given a subspace $R$ of $\mathbb{R}^d$, let $\Pi_R$ denote the orthogonal projection onto $R$, satisfying that for all $x \in \mathbb{R}^d$:

$$\Pi_R(x) \in R \text{ and } \forall r \in R, \|x - \Pi_R(x)\|_2 \leq \|x - r\|_2. \tag{A.1}$$

If $E \in \mathbb{R}^{d \times \dim(R)}$ has orthonormal columns that form a basis for $R$, then we have:

$$\Pi_R = EE^\top \tag{A.2}$$

From this we can easily check that $\Pi_R^2 = \Pi_R$ and $\Pi_R^\top = \Pi_R$. See e.g., Chapter 2.5.1 Golub & Loan (2013) for more information.

**Principal Angles**: Given two non-zero vectors $x$ and $y$, the cosine of the angle between them, $\cos\theta$, is:

$$\cos\theta = \frac{x^\top y}{\|x\|_2 \|y\|_2} \tag{A.3}$$

If we consider the 1-dimensional subspaces (so basically lines) $S_x$ and $S_y$ spanned by $x$ and $y$ respectively, then the angle between them, $\cos\theta'$ is given by the absolute value (since lines are undirected):

$$\cos\theta' = \frac{|x^\top y|}{\|x\|_2 \|y\|_2} \tag{A.4}$$

Principal angles generalize this notion to higher dimensions. See e.g., Chapter 6.4.3 in Golub & Loan (2013) for more information on principal angles.

**Definition A.1.** *Given two non-empty subspaces $R$ and $S$ of $\mathbb{R}^d$, where $r = \min(\dim(R), \dim(S))$, we have $r$ principal angles:*

$$0 \leq \theta_1 \leq ... \leq \theta_r \leq \pi/2. \tag{A.5}$$

*The directions of the inequalities swap when we take the cosine of the principal angles:*

$$1 \geq \cos\theta_1 \geq ... \geq \cos\theta_r \geq 0. \tag{A.6}$$

*The cosines of the principal angles are given by the SVD—let $E \in \mathbb{R}^{d \times \dim(R)}$ and $F \in \mathbb{R}^{d \times \dim(S)}$ have orthonormal columns which span $R$ and $S$ respectively. Then we have:*

$$\cos\theta_i = \sigma_i(E^\top F), \tag{A.7}$$

*where $\sigma_i$ denotes the $i$-th largest singular value. In this paper, we are interested in the cosine of the largest angle between them, given by:*

$$\cos\theta_{\mathsf{max}}(R,S) = \cos\theta_r \tag{A.8}$$

We can massage this into a variational characterization of the maximum principal angle, which is important for lower bounding the error of fine-tuning outside the span of the training data.

**Lemma A.2.** *Suppose $\dim(R) \leq \dim(S)$, and let $F \in \mathbb{R}^{d \times \dim(S)}$ have orthonormal columns that form a basis for $S$. We have:*

$$\cos\theta_{\mathsf{max}}(R,S) = \min_{r \in R, \|r\|_2 = 1} \|F^\top(r)\|_2 \tag{A.9}$$

*Proof.* Let $E \in \mathbb{R}^{d \times \dim(R)}$ and $F \in \mathbb{R}^{d \times \dim(S)}$ have orthonormal columns that span $R$ and $S$ respectively. Since $\dim(R) \leq \dim(S)$ (a crucial condition!), $F^\top E$ is a 'tall' matrix (it has more rows than columns) so we have:

$$\sigma_{\min}(F^\top E) = \min_{\|v\|_2 = 1} \|F^\top E v\|_2. \tag{A.10}$$

The result now follows from some algebra:

$$\cos\theta_{\mathsf{max}}(R,S) = \sigma_{\min}(F^\top E) \tag{A.11}$$

$$= \min_{\|v\|_2 = 1} \|F^\top E v\|_2 \tag{A.12}$$

$$= \min_{r \in R, \|r\|_2 = 1} \|F^\top (r)\|_2. \tag{A.13}$$

$\square$

## A.2 FEATURE DISTORTION THEOREM

We first prove our core theorem, that fine-tuning distorts pretrained features.

**Restatement of Theorem 3.2.** *In the overparameterized linear setting, let $S^\perp = rowspace(X)^\perp$, $R_0 = rowspace(B_0)$, and $v_\star, B_\star$ be the optimal parameters with $w_\star = B_\star v_\star$. If $\cos\theta_{\mathsf{max}}(R_0, S^\perp) > 0$, then for all time steps $t$, the OOD error of the fine-tuning iterates $(B_{\mathsf{ft}}(t), v_{\mathsf{ft}}(t))$ is lower bounded:*

$$\sqrt{L_{\mathsf{ood}}(v_{\mathsf{ft}}(t), B_{\mathsf{ft}}(t))} \geq \sqrt{\sigma_{\min}(\Sigma)} \Big( \frac{\cos\theta_{\mathsf{max}}(R_0, S^\perp)}{\sqrt{k}} \frac{\min(\varphi, \varphi^2/\|w_\star\|_2)}{(1 + \|w_\star\|_2)^2} - \epsilon \Big), \tag{A.14}$$

*where $\varphi^2 = |(v_0^\top v_\star)^2 - (v_\star^\top v_\star)^2|$ is defined to be inital head alignment error and $\epsilon \geq d(B_0, B_\star)$ is the error in the pretrained feature extractor.*

We follow the sketch in the main paper. We begin with a few lemmas, showing that certain quantities are preserved throughout the fine-tuning process.

Our first lemma says that the representations $B_{ft}^t x$ do not change for examples perpendicular the span of the training examples. Note that the final output $v_{ft}^t{}^\top B_{ft}^t x$ still changes, because $v_{ft}^t$ changes.

**Lemma A.3.** *For all times $t$ and all $x \in S^\perp$, we have:*

$$B_0 x = B_{ft}^t x \tag{A.15}$$

*Proof.* We initialized fine-tuning with the feature extractor $B_{\mathsf{ft}}(0) = B_0$. It suffices to show that $\partial_t B_{ft}^t x = 0$ for all $x \in S^\perp$. Recall that $\partial_t B_{ft}^t$ is given by the gradient flow update equation:

$$\partial_t B_{ft}^t = -\partial_B \widehat{L}(v_{ft}^t, B_{ft}^t) = -\partial_B \|XB^\top v - Y\|_2^2 \tag{A.16}$$

Computing the RHS explicitly using multivariable chain rule, we get:

$$\partial_t B_{ft}^t = -2v(XB^\top v - Y)^\top X \tag{A.17}$$

Since $x$ is a constant, we get:

$$\partial_t B_{ft}^t x = -2v(XB^\top v - Y)^\top X x \tag{A.18}$$

But $Xx = 0$ for $x \in S^\perp$, since $x \in S^\perp$ is defined as $x$ is perpendicular to the rowspace of $X$ (i.e., perpendicular to the rows of $X$). So the RHS is 0—that is, $\partial_t B_{ft}^t x = 0$, as desired. $\square$

Next, we show that the change in the head and feature extractor are 'coupled'. So if the head changes in a certain way, then the feature extractor cannot just stay the same. In the literature, this is sometimes called the "balancedness" lemma, and has been proved in prior work on two layer linear networks.

**Lemma A.4.** *For all $t$ we have:*

$$v_0 v_0^\top - B_0 B_0^\top = v_{ft}^t v_{ft}^t{}^\top - B_{ft}^t B_{ft}^t{}^\top \tag{A.19}$$

*Proof.* This follows by showing that the derivative is 0:

$$\partial_t[v_{ft}^t {v_{ft}^t}^\top - B_{ft}^t {B_{ft}^t}^\top] = 0 \tag{A.20}$$

Which can be verified by direct calculation. See Theorem 2.2 in Du et al. (2018) and the proof of Theorem 1 in Arora et al. (2018). □

For our proof we will require that every feature $r \in R$ can be generated from some OOD direction, that is $r = B_0 u$ for some $u \in S^\perp$. We will show that this is implied by the condition on the principal angle: $\cos\theta_{\max}(R, S^\perp) > 0$ where $R = \text{rowspace}(B_0)$, which we assumed in Theorem 3.2. The following lemma shows this (and also quantifies that the norm of $u$ does not shrink too much when projected onto $R$).

**Lemma A.5.** *Let $R, S$ be subspaces of $\mathbb{R}^d$ with $\dim(R) \leq \dim(S)$. For all $r \in R$ with $\|r\|_2 = \cos\theta_{\max}(R,S)$, there exists $s \in S$ with $\Pi_R(s) = r$ and $\|s\|_2 \leq 1$. Here $\Pi_R \in \mathbb{R}^{d \times d}$ projects a vector onto $R$.*

*Proof.* Let $c = \cos\theta_{\max}(R, S)$. Firt, we get rid of an easy case—if $c = 0$, then we need to show the claim for all $r \in R$ with $\|r\|_2 = c = 0$, which means $r = 0$. Then we can just pick $s = 0$, and $\Pi_R(s) = 0 = r$ and $\|s\|_2 = 0 \leq 1$. So for the rest of the proof we assume $c > 0$.

Consider arbitrary vector $r \in R$ with $\|r\|_2 = c$. Let $E \in \mathbb{R}^{d \times \dim(S)}, F \in \mathbb{R}^{d \times \dim(R)}$ have orthonormal columns, which form a basis for $R$ and $S$ respectively.

**Step 1: Finding $s$:** Since the columns of $E$ span $R$, $r = Ez$ for some $z \in \mathbb{R}^{\dim(R)}$. $c = \sigma_{\min}(E^\top F) > 0$, which means that $E^\top F \in \mathbb{R}^{\dim(R) \times \dim(S)}$ has rank $\dim(R)$ since $\dim(R) \leq \dim(S)$—in other words, $E^\top F$ has full column rank since the column dimension is smaller than the row dimension. So $z = E^\top F w$ for some $w \in \text{rowspace}(E^\top F)$. Then we set $s = Fw$—this means $s \in S$ because the columns of $F$ form a basis for $S$. In addition, following the steps above we have $r = Ez = EE^\top F w = EE^\top s$. We note that $\Pi_R = EE^\top$ is the projection onto $R$ (see e.g., Chapter 2.5.1 of Golub & Loan (2013)).

**Step 2: Bounding norm of $s$:** It suffices to show that $\|s\|_2 \leq 1$. Since $F$ has orthonormal columns, $\|s\|_2 = \|Fw\|_2 = \|w\|_2$, so it suffices to show that $\|w\|_2 \leq 1$. Since $E$ has orthonormal columns, $\|r\|_2 = \|z\|_2$. Recall that $z = E^\top F w$—since $w \in \text{rowspace}(E^\top F)$, from Lemma A.6 we have:

$$\|z\|_2 \geq \sigma_{\min}(E^\top F)\|w\|_2 = c\|w\|_2. \tag{A.21}$$

Rearranging, we get $\|w\|_2 \leq \|z\|_2/c = 1$, as desired.

□

In the lemma above, we used a standard linear algebraic result that we include for completeness. This says that $A$ cannot shrink vectors in its rowspace too much, where the shrinkage factor is given by the minimum singular value of $A$.

**Lemma A.6.** *Let $A \in \mathbb{R}^{m \times n}$. Let $r = \min(m, n)$. Then if $x \in \text{rowspace}(A)$, we have $\|Ax\|_2 \geq \sigma_r(A)\|x\|_2$.*

*Proof.* We bound the norm of $x$ using the SVD. Consider the singular value decomposition (SVD) of $A$:

$$A = UDV^\top \tag{A.22}$$

Where $U \in \mathbb{R}^{m \times r}, D \in \mathbb{R}^{r \times r}, V^\top \in \mathbb{R}^{r \times n}$, where $U$ and $V$ have orthonormal columns, and $D = \text{diag}(\sigma_1, ..., \sigma_r)$ is a diagonal matrix with $\sigma_1 \geq ... \geq \sigma_r \geq 0$.

$$
\begin{aligned}
\|Ax\|_2 &= \|UDV^\top x\|_2 && [\text{Definition of } r] && \text{(A.23)} \\
&= \|DV^\top x\|_2 && [U \in \mathbb{R}^{m \times r} \text{ has orthonormal columns}] && \text{(A.24)} \\
&\geq \sigma_r\|V^\top x\|_2 && [D \text{ is diagonal}] && \text{(A.25)} \\
&= \sigma_r\|x\|_2 && [\text{rows of } V^\top \text{ are orthonormal, } x \text{ is in rowspace}] && \text{(A.26)} \\
&= \sigma_r(A)\|x\|_2 && && \text{(A.27)}
\end{aligned}
$$

Where for the fourth step, we used the fact that if $x \in \text{rowspace}(V^\top)$ and the rows of $V^\top$ are orthonormal, then $\|V^\top x\|_2 = \|x\|_2$. One way to see this is by writing $x = \sum_i \alpha_i v_i$, where $v_i$ are rows of $V^\top$, and then noting that $V^\top x = (\alpha_1, ..., \alpha_r)$ and so $x$ and $V^\top x$ have the same norm. $\qquad\square$

We recall that $P_{\text{ood}}$ has second moment $\Sigma$: $\mathbb{E}[xx^\top] = \Sigma$ when $x \sim P_{\text{ood}}$, where $\Sigma$ is invertible. So with some simple algebra we can write the OOD error $L_{\text{ood}}$ in terms of $\Sigma$ (the proof is standard and basic, but we include it just for completeness):

**Lemma A.7.**

$$L_{\text{ood}}(v, B) = (B_\star^\top v_\star - B^\top v)^\top \Sigma (B_\star^\top v_\star - B^\top v) \le \sigma_{\min}(\Sigma) \|B_\star^\top v_\star - B^\top v\|_2^2. \tag{A.28}$$

*Proof.* Let $x \sim P_{\text{ood}}$. We have,

$$L_{\text{ood}}(v, B) = \mathbb{E}[(v_\star^\top B_\star x - v^\top B x)^2] \tag{A.29}$$

$$= \mathbb{E}[(B_\star^\top v_\star - B^\top v)^\top xx^\top (B_\star^\top v_\star - B^\top v)] \tag{A.30}$$

$$= (B_\star^\top v_\star - B^\top v)^\top \mathbb{E}[xx^\top](B_\star^\top v_\star - B^\top v) \tag{A.31}$$

$$= (B_\star^\top v_\star - B^\top v)^\top \Sigma (B_\star^\top v_\star - B^\top v). \tag{A.32}$$

The inequality follows immediately because $\sigma_{\min}(A)$ (for a square matrix $A$) is simply the min over $x$ with unit $\ell_2$ norm of $x^\top A x$. $\qquad\square$

We now prove Theorem 3.2, following the 3 steps outlined in the main text.

*Proof of Theorem 3.2.* Let $c = \cos\theta_{\max}(R, S^\perp)$. From Lemma A.7, we have $L_{\text{ood}}(v_{ft}^t, B_{ft}^t) \le \sigma_{\min}(\Sigma)\|B_\star^\top v_\star - B_{ft}^t{}^\top v_{ft}^t\|_2^2$ so it suffices to bound $\|B_\star^\top v_\star - B_{ft}^t{}^\top v_{ft}^t\|_2$.

Because it makes the proof much easier, we will prove the contrapositive, and then convert back to the original theorem statement. We assume $\|B_\star^\top v_\star - B_{ft}^t{}^\top v_{ft}^t\|_2 \le \Delta$, and will show that:

$$|(v_0^\top v_\star)^2 - (v_\star^\top v_\star)^2| \le \frac{\Delta + \epsilon}{c} g_1(\|w\|_2)\sqrt{k} + \frac{(\Delta + \epsilon)^2}{c^2} g_2(\|w\|_2) k \tag{A.33}$$

Where $g_1$ and $g_2$ are non-negative polynomials we will bound in the proof.

We gave a basic outline of the proof in the main paper, and here we are just trying to be careful about capturing all the dependencies. We also give intuition for each step before diving into algebra (which we include for completeness).

Recall that in the overparameterized linear setting we assumed we have orthonormal $B_0$ with $\|B_0 - UB_\star\|_2 \le \epsilon$ for some $U$. We note that the setup is rotationally symmetric so without loss of generality we can suppose $\|B_0 - B_\star\|_2 \le \epsilon$. This is because we can let $B_\star' = UB_\star$ and $v_\star' = Uv_\star$, and we have $w_\star = B_\star^\top v_\star = (UB_\star)^\top (Uv_\star)$, where $w_\star$ is the optimal classifier—so we can now write the entire proof in terms of $B_\star'$ and $v_\star'$.

**Step 1: Show that** $\|v_{ft}^t - v_\star\|_2 \le \Delta/c$: We first give intuition and then dive into the math. The key insight is to use the fact that in 'many' directions $B_{ft}^t$ and $B_0$ are the same (formally, for all $x \in S^\perp$, $B_{ft}^t x = B_0 x$). But $B_0$ and $B_\star$ are close by assumption, which means that $B_{ft}^t$ and $B_\star$ are close in 'many' directions. Then since we assumed in the contrapositive that $v_{ft}^t{}^\top B_{ft}^t$ and $v_\star^\top B_\star$ are close, we get that $v_{ft}^t$ and $v_\star$ are close in 'many' directions. Because $S^\perp$ covers the rowspace of $B_0$, we get that 'many' is $k$, which is precisely the dimensionality of $v_\star$, so the two vectors $v_{ft}^t$ and $v_\star$ must be close.

We now dive into the math. Since $B_0$ has orthogonal rows, $B_0$ has full column rank.

Let $z$ be given by:

$$z = \frac{c}{\|v_{ft}^t - v_\star\|_2}(v_{ft}^t - v_\star) \tag{A.34}$$

We note that $\|z\|_2 = c$. Then, we can find $y \in R = \text{rowspace}(B_0)$ such that $B_0 y = z$ (since $B_0$ has full column-rank) and then $\|y\|_2 = \|z\|_2 = c$ (since $B_0$ has orthonormal rows).

Since $c = \cos\theta_{\max}(R, S^\perp) > 0$, and $y \in R$ with $\|y\| = c$, from Lemma A.5 we can choose $x \in S^\perp$ with $\|x\|_2 \le 1$ and $\Pi_R(x) = y$. Then, we have $B_0 x = z$.

From Proposition A.3, since $x \in S^\perp$, $B_0$ does not change in directions of $x$ when fine-tuning so we have: $B_0 x = B_{ft}^t x$.

The claim now follows from simple algebraic manipulation, following the intuition we described. The algebra just captures what 'close' means and adds up the error terms.

$$\|v_{ft}^t - v_\star\|_2 = \frac{1}{c}(v_{ft}^t - v_\star)^\top \left( \frac{c(v_{ft}^t - v_\star)}{\|v_{ft}^t - v_\star\|_2} \right) \qquad \text{[Algebra]} \qquad (A.35)$$

$$= \frac{1}{c}(v_{ft}^t - v_\star)^\top z \qquad \text{[Definition of } z] \qquad (A.36)$$

$$= \frac{1}{c}(v_{ft}^t - v_\star)^\top B_0 x \qquad \text{[Since } B_0 x = z] \qquad (A.37)$$

$$= \frac{1}{c}(v_{ft}^{t\,\top} B_0 x - v_\star^\top B_0 x) \qquad \text{[Algebra]} \qquad (A.38)$$

$$= \frac{1}{c}(v_{ft}^{t\,\top} B_{ft}^t x - v_\star^\top B_0 x) \qquad [B_{ft}^t x = B_0 x \text{ since } x \in S^\perp]$$
$$\tag{A.39}$$

$$= \frac{1}{c}(v_{ft}^{t\,\top} B_{ft}^t - v_\star^\top B_0)x \qquad \text{[Algebra]} \qquad (A.40)$$

$$\le \frac{1}{c}\|v_{ft}^{t\,\top} B_{ft}^t - v_\star^\top B_0\|_2 \|x\|_2 \qquad \text{[Cauchy-Schwarz]} \qquad (A.41)$$

$$\le \frac{1}{c}\|v_{ft}^{t\,\top} B_{ft}^t - v_\star^\top B_0\|_2 \qquad \text{[since } \|x\|_2 \le 1] \qquad (A.42)$$

$$\le \frac{1}{c}\|v_{ft}^{t\,\top} B_{ft}^t - v_\star^\top B_\star\|_2 + \frac{1}{c}\|v_\star^\top B_\star - v_\star^\top B_0\|_2 \qquad \text{[Triangle inequality]} \qquad (A.43)$$

$$\le \frac{1}{c}\|B_{ft}^{t\,\top} v_{ft}^t - B_\star^\top v_\star\|_2 + \frac{1}{c}\|v_\star^\top B_\star - v_\star^\top B_0\|_2 \qquad \text{[Taking transpose]} \qquad (A.44)$$

$$= \frac{1}{c}\|B_{ft}^{t\,\top} v_{ft}^t - B_\star^\top v_\star\|_2 + \frac{1}{c}\sigma_{\max}(B_0 - B_\star)\|v_\star\|_2 \qquad \text{[definition of max singular value]}$$
$$\tag{A.45}$$

$$= \frac{1}{c}\|B_{ft}^{t\,\top} v_{ft}^t - B_\star^\top v_\star\|_2 + \frac{1}{c}\epsilon \|v_\star\|_2 \qquad \text{[since } \sigma_{\max}(B_0 - B_\star) \le \epsilon]$$
$$\tag{A.46}$$

$$\le \frac{\Delta + \epsilon\|v_\star\|_2}{c} \qquad \text{[since } \|B_\star^\top v_\star - B_{ft}^{t\,\top} v_{ft}^t\|_2 \le \Delta]$$
$$\tag{A.47}$$
$$\tag{A.48}$$

Which shows that $\|v_{ft}^t - v_\star\|_2 \le (\Delta + \epsilon\|v_\star\|_2)/c$.

**Step 2A: Show that $\|B_{ft}^t\|_F^2$ is small:** The key insight is to take the trace on both sides of Proposition A.4, which bounds the Frobenius norm of $B_{ft}^t$ and therefore the operator norm.

Rearranging Proposition A.4, we have:

$$B_{ft}^t B_{ft}^{t\,\top} = B_0 B_0^\top + v_\star v_\star^\top - v_0 v_0^\top \tag{A.49}$$

Taking the trace everywhere, we get:

$$\text{Tr}(B_{ft}^t B_{ft}^{t\,\top}) = \text{Tr}(B_0 B_0^\top) + \text{Tr}(v_\star v_\star^\top) - \text{Tr}(v_0 v_0^\top) \tag{A.50}$$

For any matrix $A$, $\text{Tr}(AA^\top) = \|A\|_F^2$, and for a vector $v$ the Frobenius norm is just the $\ell_2$-norm, so $\text{Tr}(vv^\top) = \|v\|_2^2$. So we have:

$$\|B_{ft}^t\|_F^2 = \|B_0\|_F^2 + \|v_\star\|_2^2 - \|v_0\|_2^2 \tag{A.51}$$

Squares are non-negative, so we get the inequality:

$$\|B_{ft}^t\|_F^2 \le \|B_0\|_F^2 + \|v_\star\|_2^2 \tag{A.52}$$

**Step 2B: Show that** $\|B_0^\top v_\star\|_2^2 - \|B_{ft}^t{}^\top v_\star\|_2^2$ **is small**: This step doesn't involve much insight, and is standard peturbation analysis—we simply factor the difference of squares and bound each term.

First, we bound $\|B_{ft}^t{}^\top v_{ft}^t - B_{ft}^t{}^\top v_\star\|_2$:

$$\|B_{ft}^t{}^\top v_{ft}^t - B_{ft}^t{}^\top v_\star\|_2 \leq \sigma_{\max}(B_{ft}^t)\|v_{ft}^t - v_\star\|_2 \tag{A.53}$$

$$\leq \|B_{ft}^t\|_F \|v_{ft}^t - v_\star\|_2 \tag{A.54}$$

$$\leq \sqrt{\|B_0\|_F^2 + \|v_\star\|_2^2}\|v_{ft}^t - v_\star\|_2 \tag{A.55}$$

$$\leq \sqrt{\|B_0\|_F^2 + \|v_\star\|_2^2}\left(\frac{\Delta + \epsilon\|v_\star\|_2}{c}\right) \tag{A.56}$$

Next, we bound $\|B_0^\top v_\star - B_{ft}^t{}^\top v_\star\|_2$:

$$\|B_0^\top v_\star - B_{ft}^t{}^\top v_\star\|_2 \leq \|B_0^\top v_\star - B_\star^\top v_\star\|_2 + \|B_\star^\top v_\star - B_{ft}^t{}^\top v_\star\|_2 \tag{A.57}$$

$$\leq \sigma_{\max}(B_0 - B_\star)\|v_\star\|_2 + \|B_\star^\top v_\star - B_{ft}^t{}^\top v_\star\|_2 \tag{A.58}$$

$$\leq \epsilon\|v_\star\|_2 + \|B_\star^\top v_\star - B_{ft}^t{}^\top v_\star\|_2 \tag{A.59}$$

$$\leq \epsilon\|v_\star\|_2 + \|B_\star^\top v_\star - B_{ft}^t{}^\top v_{ft}^t\|_2 + \|B_{ft}^t{}^\top v_{ft}^t - B_{ft}^t{}^\top v_\star\|_2 \tag{A.60}$$

$$\leq \epsilon\|v_\star\|_2 + \Delta + \sqrt{\|B_0\|_F^2 + \|v_\star\|_2^2}\left(\frac{\Delta + \epsilon\|v_\star\|_2}{c}\right) \tag{A.61}$$

$$=: \Delta_2 \tag{A.62}$$

Finally, we bound $\left|\|B_0^\top v_\star\|_2^2 - \|B_{ft}^t{}^\top v_\star\|_2^2\right|$, using the identity:

$$\left|\|u\|_2^2 - \|v\|_2^2\right| = |(u - v)^\top(u + v)| \tag{A.63}$$

$$\leq \|u - v\|_2 \|u + v\|_2 \tag{A.64}$$

$$\leq \|u - v\|_2(2\|u\|_2 + \|u - v\|_2) \tag{A.65}$$

Applying this:

$$\left|\|B_0^\top v_\star\|_2^2 - \|B_{ft}^t{}^\top v_\star\|_2^2\right| \leq \|B_0^\top v_\star - B_{ft}^t{}^\top v_\star\|_2(2\|B_0^\top v_\star\|_2 + \|B_0^\top v_\star - B_{ft}^t{}^\top v_\star\|_2) \tag{A.66}$$

$$\leq \Delta_2(2\|B_0^\top v_\star\|_2 + \Delta_2) \tag{A.67}$$

$$\leq \Delta_2(2\|B_\star^\top v_\star\|_2 + 2\|B_0^\top v_\star - B_\star^\top v_\star\|_2 + \Delta_2) \tag{A.68}$$

$$\leq \Delta_2(2\|w_\star\|_2 + 2\epsilon\|v_\star\|_2 + \Delta_2) \tag{A.69}$$

$$=: \Delta_3 \tag{A.70}$$

**Step 3: Use Proposition A.4 to show** $v_0$ **and** $v_\star$ **must be close**: The key insight is that we start from Proposition A.4, and left and right multiply by $v_\star$, after that we use the previous steps and do some some standard perturbation analysis.

We start from Proposition A.4:

$$v_0 v_0^\top - B_0 B_0^\top = v_{ft}^t v_{ft}^t{}^\top - B_{ft}^t B_{ft}^t{}^\top \tag{A.71}$$

The key step is to left multiply both sides by $v_\star^\top$ and right multiply both sides by $v_\star$ to get:

$$(v_0^\top v_\star)^2 - \|B_0^\top v_\star\|_2^2 = (v_{ft}^t{}^\top v_\star)^2 - \|B_{ft}^t{}^\top v_\star\|_2^2 \tag{A.72}$$

Rearranging, and then using Equation A.66, we get:

$$\left|(v_{ft}^t{}^\top v_\star)^2 - (v_0^\top v_\star)^2\right| = \left|\|B_{ft}^t{}^\top v_\star\|_2^2 - \|B_0^\top v_\star\|_2^2\right| \leq \Delta_3 \tag{A.73}$$

This is close to what we want, except we have $(v_{ft}^t{}^\top v_\star)^2$ on the LHS instead of $(v_\star^\top v_\star)^2$. We previously showed that $v_{ft}^t$ and $v_\star$ are close, in Step 1, so with some algebra we can bound the

difference between $(v_{ft}^t{}^\top v_\star)^2$ and $(v_\star^\top v_\star)^2$:

$$|(v_{ft}^t{}^\top v_\star)^2 - (v_\star^\top v_\star)^2| = |(v_{ft}^t{}^\top v_\star - v_\star^\top v_\star)^\top (v_{ft}^t{}^\top v_\star + v_\star^\top v_\star)| \tag{A.74}$$

$$= |(v_{ft}^t{}^\top v_\star - v_\star^\top v_\star)^\top [2v_\star^\top v_\star + (v_{ft}^t{}^\top v_\star - v_\star^\top v_\star)]| \tag{A.75}$$

$$= |(v_\star^\top (v_{ft}^t - v_\star))^\top [2v_\star^\top v_\star + (v_\star^\top (v_{ft}^t - v_\star))]| \tag{A.76}$$

$$\leq \|v_{ft}^t - v_\star\|_2 \|v_\star\|_2^2 [2\|v_\star\|_2 + \|v_{ft}^t - v_\star\|_2] \tag{A.77}$$

$$= (\Delta/c)\|v_\star\|_2^2 (2\|v_\star\|_2 + (\Delta/c)) := \Delta_4 \tag{A.78}$$

Above, from the third line to the fourth line, we used triangle inequality and Cauchy-Schwarz.

So finally, by triangle-inequality we can now bound $|(v_\star^\top v_\star)^2 - (v_0^\top v_\star)^2|$:

$$|(v_\star^\top v_\star)^2 - (v_0^\top v_\star)^2| \leq |(v_\star^\top v_\star)^2 - (v_{ft}^t{}^\top v_\star)^2| + |(v_{ft}^t{}^\top v_\star)^2 - (v_0^\top v_\star)^2| \tag{A.79}$$

$$\leq \Delta_4 + \Delta_3 \tag{A.80}$$

**Wrap up i.e., writing out $\Delta_4 + \Delta_3$ explicitly**: This is basically the bound we want, but we would like to express $\Delta_3, \Delta_4$ in terms of $\Delta$ and $\epsilon$. Note that this step has no insight, and is just algebra—we include the details for reference and verifiability. We recall:

$$\Delta_4 = (\Delta/c)\|v_\star\|_2^2 (2\|v_\star\|_2 + (\Delta/c)) \tag{A.81}$$

$$\Delta_3 = \Delta_2 (2\|w_\star\|_2 + 2\epsilon\|v_\star\|_2 + \Delta_2) \tag{A.82}$$

$$\Delta_2 = \epsilon\|v_\star\|_2 + \Delta + \sqrt{\|B_0\|_F^2 + \|v_\star\|_2^2}\left(\frac{\Delta + \epsilon\|v_\star\|_2}{c}\right) \tag{A.83}$$

Since $B_0$ has orthogonal rows (by assumption), $B_0^\top$ has orthogonal columns, so $\|w_\star\|_2 = \|B_0^\top v_\star\|_2 = \|v_\star\|_2$. In addition, since $B_0$ has $k$ orthogonal rows, $\|B_0\|_F = \sqrt{k}$. We also note that $\sqrt{\|B_0\|_F^2 + \|v_\star\|_2^2} \leq \|B_0\|_F + \|v_\star\|_2 = \sqrt{k} + \|w_\star\|_2$. Since $c \leq 1$, we have:

$$\epsilon\|v_\star\|_2 + \Delta \leq \left(\frac{\Delta + \epsilon\|v_\star\|_2}{c}\right) \tag{A.84}$$

So for $\Delta_2$, up to constant factors we can ignore the $\epsilon\|v_\star\|_2 + \Delta$ term—this means we get:

$$\Delta_2 \leq O\left((\sqrt{k} + \|w_\star\|_2)\left(\frac{\Delta + \epsilon\|w_\star\|_2}{c}\right)\right) \tag{A.85}$$

Using the fact that $\sqrt{k} + \|w_\star\|_2 \leq \sqrt{k}(1 + \|w_\star\|)$ we get:

$$\Delta_2 \leq O\left(\sqrt{k}(1 + \|w_\star\|)\left(\frac{\Delta + \epsilon\|w_\star\|_2}{c}\right)\right) \tag{A.86}$$

Then since $\Delta + \epsilon\|w_\star\|_2 \leq (1 + \|w_\star\|_2)(\Delta + \epsilon)$, we get:

$$\Delta_2 \leq O\left(\sqrt{k}(1 + \|w_\star\|)^2\left(\frac{\Delta + \epsilon}{c}\right)\right) \tag{A.87}$$

Now for $\Delta_3$, first note that $\epsilon \leq 2$, since $B_\star$ and $B_0$ have orthogonormal rows so $\|B_\star - B_0\|_2 \leq 2$. This means that $\epsilon\|w_\star\|_2 \leq \|w_\star\|_2$, so $\Delta_3$ simplifies to:

$$\Delta_3 \leq O(\Delta_2(\|w_\star\|_2 + \Delta_2)) = O(\Delta_2\|w_\star\|_2 + \Delta_2) \tag{A.88}$$

Substituting the bound for $\Delta_2$ into $\Delta_3$, we get:

$$\Delta_3 \leq O\left(\sqrt{k}\|w_\star\|_2(1 + \|w_\star\|)^2\left(\frac{\Delta + \epsilon\|w_\star\|_2}{c}\right) + k(1 + \|w_\star\|)^4\left(\frac{\Delta + \epsilon\|w_\star\|_2}{c}\right)^2\right) \tag{A.89}$$

For $\Delta_4$, we get:

$$\Delta_4 \leq O\left(\|w_\star\|_2^3\frac{\Delta}{c} + \|w_\star\|_2(\frac{\Delta}{c})\right) \tag{A.90}$$

Since $\Delta/c \leq (\Delta + \epsilon)/c$ and $\|w_\star\|_2^2 \leq (1 + \|w_\star\|_2)^2$ we have for the final error $\Delta_3 + \Delta_4$:

$$\Delta_3 + \Delta_4 \leq \sqrt{k}w(1 + \|w_\star\|_2^2)^2\left(\frac{\Delta + \epsilon}{c}\right) + k(1 + \|w_\star\|_2^2)^4\left(\frac{\Delta + \epsilon}{c}\right)^2 \tag{A.91}$$

**Wrap up i.e., taking the contrapositive:** So we've shown that if $\|B_\star^\top v_\star - B_{ft}^t{}^\top v_{ft}^t\|_2^2 \le \Delta$, then:

$$|(v_\star^\top v_\star)^2 - (v_0^\top v_\star)^2| \le \frac{\Delta + \epsilon}{c} w(1 + \|w_\star\|_2^2)^2 \sqrt{k} + \frac{(\Delta + \epsilon)^2}{c^2}(1 + \|w_\star\|_2^2)^4 k \quad (A.92)$$

We'd like to flip this around: suppose $|(v_\star^\top v_\star)^2 - (v_0^\top v_\star)^2| \ge \varphi^2$ for some $\varphi$. To lower bound $\|B_\star^\top v_\star - B_{ft}^t{}^\top v_{ft}^t\|_2^2$, we simply take the contrapositive of what we have proved. Let $\Delta$ be given by:

$$\Delta = \min\left(\frac{c}{w(1 + \|w_\star\|_2^2)^2 \sqrt{k}} \varphi^2, \frac{c}{\sqrt{(1 + \|w_\star\|_2^2)^4 k}} \varphi\right) - \epsilon \quad (A.93)$$

In this case with some algebra, we can show that:

$$|(v_\star^\top v_\star)^2 - (v_0^\top v_\star)^2| \ge \varphi^2 \ge \frac{\Delta + \epsilon}{c} w(1 + \|w_\star\|_2^2)^2 \sqrt{k} + \frac{(\Delta + \epsilon)^2}{c^2}(1 + \|w_\star\|_2^2)^4 k \quad (A.94)$$

To see this, we bound each of the terms in the RHS separately using our definition of $\Delta$. Then, from the contrapositive of what we proved (compare with Equation A.92, we get:

$$\|B_\star^\top v_\star - B_{ft}^t{}^\top v_{ft}^t\|_2^2 \ge \Delta \quad (A.95)$$

Finally, we can massage $\Delta$ to combine terms and make it look slightly nicer:

$$\Delta \ge \frac{c}{\sqrt{k}} \frac{\min(\varphi, \varphi^2/\|w_\star\|_2)}{(1 + \|w_\star\|_2)^2} - \epsilon \quad (A.96)$$

Then applying Lemma A.7 we get the desired result. For even more interpretability, if $\|w\|_2 = 1$ and $\varphi$ is bounded above by some constant, then you can think of $\Delta$ as approximately $\frac{c}{\sqrt{k}} \varphi^2 - \epsilon$. This completes the proof. $\qquad\square$

## A.3 LP VS. FT (OOD)

We now prove Theorem 3.4, which compares linear probing and fine-tuning in the linear overparameterized setting, when the ID data lies in a lower dimensional subspace. We first define a distance (formally, a pseudometric) between pretrained feature exactors, which measures the operator norm difference between them up to rotations (e.g., if $B = UB'$ for a rotation matrix $U$ then $B$ and $B'$ are equivalent since this means we can obtain one feature extractor's representations from another just by rotating):

**Definition A.8** (Feature Extractor Distance). *The distance between feature extractors $B, B' \in \mathbb{R}^{k \times d}$ (with orthonormal rows) is given by (where the min is over rotation matrices $U \in \mathbb{R}^{k \times k}$):*

$$d(B, B') = \min_U \|B - UB'\|_2. \quad (A.97)$$

We state a more precise version of Theorem 3.4—basically we fix all problem parameters except $B_0$ (which limits to $B_\star$). To define the limit, we consider a sequence of pretrained feature extractors: $\{B_0^i\}_{i=1}^\infty$. We define the corresponding limit points of fine-tuning and linear probing when we start from the $i$-th pretrained feature extractor. That is, let $v_{ft}^i(t), B_{ft}^i(t)$ denote the parameters at time $t$ of fine-tuning if we initialize with $v_0, B_0^i$ (see Equation 3.2 for the fine-tuning updates). Let $v_{lp}^{\infty i}, B_0^i$ be the linear probing solution when initialized with $v_0, B_0^i$ (see Equation 3.5 for the linear probing updates). We note that the LP iterates converge to $v_{lp}^{\infty i}, B_0^i$ as a result of gradient flow on a convex problem.

Finally, Theorem 3.4 says that as the pretrained representations get better, linear probing does much better than fine-tuning OOD:

**Theorem A.9** (Formal statement of Theorem 3.4). *In the linear overparameterized setting, under the ID subspace assumption, fix the dimensions of the setting $d, k, m$, number of examples $n$, the ID subspace $S$, ID distribution $P_{id}$, the distribution over the head $v_0$, and the ground truth parameters $v_\star, B_\star$. Assume the non-degeneracy conditions $\cos\theta_{max}(R_*, S) > 0$ and $\cos\theta_{max}(R_*, S^\perp) > 0$ where $R_* = rowspace(B_\star)$. Given a sequence of pretrained feature extractors $\{B_0^i\}_{i=1}^\infty$ with $B_0^i \to B_\star$, where the limit is in the pseudometric given by Definition A.8, the ratio of OOD errors of linear probing and fine-tuning converges in probability to $0$:*

$$\frac{L_{ood}(v_{lp}^{\infty i}, B_0^i)}{\inf_{t \ge 0} L_{ood}(v_{ft}^i(t), B_{ft}^i(t))} \xrightarrow{p} 0, as \ i \to \infty. \quad (A.98)$$

*The purpose of the infimum is to capture the fact that the bound holds for all times $t$ for fine-tuning (and therefore also for the limit $v_{\mathsf{ft}}^\infty, B_{\mathsf{ft}}^\infty$ when it exists). Note that the ratio is a random variable because the training data is sampled from $P_{\mathsf{id}}$ and the head is sampled ($v_0 \sim \mathcal{N}(0, \sigma^2 I)$ for some $\sigma^2$).*

*Proof.* Recall that we say a sequence of real-valued random variables converges in probability to $0$ (written as $X_i \xrightarrow{p} 0$) if for every $\epsilon', \delta > 0$, for all large enough $i$ (that is, for all $i \geq N_i$ for some $N_i$), we have:

$$P(|X_i| > \epsilon') \leq \delta. \tag{A.99}$$

Accordingly, fix arbitrary $\epsilon', \delta > 0$, and we will show that the ratio of errors is eventually smaller than $\epsilon'$ with probability at least $1 - \delta$.

**Lower bounding fine-tuning error**: Since $B_0^i \to B_\star$, from Lemma A.11 we have that $\cos\theta_{\mathsf{max}}(R^i, S^\perp) \to \cos\theta_{\mathsf{max}}(R_*, S^\perp)$ where $R^i = \mathrm{rowspace}(B_0^i)$. Since $\cos\theta_{\mathsf{max}}(R_*, S^\perp) > 0$, this means that for all large enough $i$ we have:

$$\cos\theta_{\mathsf{max}}(R^i, S^\perp) > \cos\theta_{\mathsf{max}}(R_*, S^\perp)/2. \tag{A.100}$$

Next, from Lemma A.13, we have that with probability at least $1 - \delta/2$, Head-Error$(v_0, v_\star) = |(v_0^\top v_\star)^2 - (v_\star^\top v_\star)^2| \geq c_\delta$ for some $c_\delta > 0$. Plugging this into the fine-tuning bound in Theorem 3.2, this means that for all large enough $i$ with probability at least $1 - \delta/2$:

$$\inf_{t \geq 0} \sqrt{L_{\mathsf{ood}}(v_{\mathsf{ft}}{}^i(t), B_{\mathsf{ft}}{}^i(t))} \geq c_\delta' - d(B_0^i, B_\star), \tag{A.101}$$

for some $c_\delta' > 0$. But since $B_0^i \to B_\star$ we have $d(B_0^i, B_\star) \to 0$ as $i \to \infty$. So this means that for all large enough $i$ with probability at least $1 - \delta/2$:

$$\inf_{t \geq 0} L_{\mathsf{ood}}(v_{\mathsf{ft}}{}^i(t), B_{\mathsf{ft}}{}^i(t)) \geq c_\delta'', \tag{A.102}$$

for some $c_\delta'' > 0$.

**Upper bounding the linear probing error**: Since $B_0^i \to B_\star$, from Lemma A.11 we have that $\cos\theta_{\mathsf{max}}(R^i, S) \to \cos\theta_{\mathsf{max}}(R_*, S)$ and so since $\cos\theta_{\mathsf{max}}(R_*, S) > 0$, for all large enough $i$ we have:

$$\cos\theta_{\mathsf{max}}(R^i, S) > \cos\theta_{\mathsf{max}}(R_*, S)/2. \tag{A.103}$$

Plugging this into the RHS of Lemma A.15, Equation A.133, which upper bounds the OOD error of linear probing, we get that for all large enough $i$, with probability at least $1 - \delta/2$:

$$L_{\mathsf{ood}}(v_{\mathsf{lp}}^{\infty i}, B_0^i) \leq u_\delta (d(B_0^i, B_\star))^2, \tag{A.104}$$

for some $u_\delta > 0$. Again since $d(B_0^i, B_\star) \to 0$ as $i \to \infty$, this means for all large enough $i$, with probability at least $1 - \delta/2$, $d(B_0^i, B_\star)$ will be small enough so that:

$$L_{\mathsf{ood}}(v_{\mathsf{lp}}^{\infty i}, B_0^i) \leq c_\delta'' \epsilon. \tag{A.105}$$

**Taking the ratio**: So taking the ratio of the lower bound for fine-tuning, and upper bound for linear probing, we get with with probability at least $1 - \delta$:

$$\frac{L_{\mathsf{ood}}(v_{\mathsf{lp}}^{\infty i}, B_0^i)}{\inf_{t \geq 0} L_{\mathsf{ood}}(v_{\mathsf{ft}}{}^i(t), B_{\mathsf{ft}}{}^i(t))} \leq \epsilon, \tag{A.106}$$

as desired.

$\square$

We now prove the Lemmas that we used in the above proof.

### A.3.1 CONVERGENCE OF PRINCIPAL ANGLE

Theorem 3.4 assumes conditions on the angle between the perfect feature extractor $B_\star$ and the ID subspace $S$. However, fine-tuning and linear probing start from features $B_0$ with some error, and do not get access to $B_\star$. We show that if $B_0$ and $B_\star$ are close, then the angles between their rowspaces to a third subspace $T$ (which could be the the ID subspace $S$) is similar.

**Lemma A.10.** *Given two feature extractors $B_0, B_\star \in \mathbb{R}^{k \times d}$ with orthonormal rows, where $R_0 = rowspace(B_0), R_* = rowspace(B_\star)$, and a subspace $T$ with dimension at least 1, we have:*

$$|\cos\theta_{\max}(R_0,T) - \cos\theta_{\max}(R_*,T)| \leq d(B_0,B_\star) \tag{A.107}$$

*Proof.* Recall that $k = \dim(R_0) = \dim(R_*)$. Let $r = \min(k, \dim(T))$ and let $F$ be a $d$-by-$\dim(T)$ matrix with orthonormal columns that form a basis for $T$. We have, for arbitrary rotation matrix $U \in \mathbb{R}^{k \times k}$:

$$\cos\theta_{\max}(R_0,T) = \sigma_r(B_0 F) \tag{A.108}$$
$$= \sigma_r(UB_0 F) \tag{A.109}$$
$$= \sigma_r(B_\star F + (UB_0 - B_\star)F) \tag{A.110}$$
$$\geq \sigma_r(B_\star F) - \sigma_1((UB_0 - B_\star)F) \tag{A.111}$$
$$\geq \sigma_r(B_\star F) - \sigma_1(UB_0 - B_\star) \tag{A.112}$$
$$= \sigma_r(B_\star F) - \|UB_0 - B_\star\|_2 \tag{A.113}$$
$$= \cos\theta_{\max}(R_*,T) - \|UB_0 - B_\star\|_2 \tag{A.114}$$

Here in the first step we used the definition of $\cos\theta_{\max}$ (Definition 3.1), and the fact that $B_0^\top$ has orthonormal columns which form a basis for $R_0$ (the rowspace of $B_0$), so in Definition 3.1 we can subtitute $E = B_0^\top$. To get Equation A.111 we used Weyl's theorem, which bounds the singular value under perturbations: $\sigma_r(A + B) \geq \sigma_r(A) - \sigma_1(B)$. To get Equation A.112 we used the fact that $\|Fv\|_2 = \|v\|$ since $F$ has orthonormal columns.

Since this holds for all rotation matrices $U$, we can take the minimum over $U$ to get:

$$\cos\theta_{\max}(R_0,T) \geq \cos\theta_{\max}(R_*,T) - \min_U \|UB_0 - B_\star\|_2 = \cos\theta_{\max}(R_*,T) - d(B_0^i,B_\star) \tag{A.115}$$

Since the relationship between $B_0$ and $B_\star$ are symmetric (and the distance $d$ is symmetric), this gives us the desired result:

$$|\cos\theta_{\max}(R_0,T) - \cos\theta_{\max}(R_*,T)| \leq d(B_0,B_\star) \tag{A.116}$$

$\square$

**Lemma A.11.** *Given a sequence of pretrained feature extractors $\{B_0^i\}_{i=1}^\infty$ with $B_0^i \to B_\star$, where $B_0^i, B_\star \in \mathbb{R}^{k \times d}$ have orthonormal rows, let $R^i = rowspace(B_0^i), R_* = rowspace(B_\star)$. Then for any subspace $T$, we have:*

$$\cos\theta_{\max}(R^i,T) \to \cos\theta_{\max}(R_*,T), as \ i \to \infty. \tag{A.117}$$

*Proof.* This follows directly from Lemma A.10. $B_0^i \to B_\star$ means $d(B_0^i, B_\star) \to 0$. Then from Lemma A.10:

$$|\cos\theta_{\max}(R^i,T) - \cos\theta_{\max}(R_*,T)| \to 0, as \ i \to \infty \tag{A.118}$$

This means $\cos\theta_{\max}(R^i,T) \to \cos\theta_{\max}(R_*,T)$ as $i \to \infty$ $\square$

### A.3.2 BOUNDING THE HEAD ERROR

We prove a lower bound on Head-Error$(v_0, v_\star) = |(v_0^\top v_\star)^2 - (v_\star^\top v_\star)^2|$, which was a key term in the fine-tuning lower bound (Theorem 3.2). Note that if the head is initialized as $v_0 = 0$, then Head-Error$(v_0, v_\star) = \|v_\star\|_2^2 = \|w_\star\|_2^2$. In practice, the head is usually initialized randomly, for example normally distributed. Intuitively, the head error is still high because we do not know which direction the head is pointing in, so most of the time the initial (randomly sampled) head will be pointing in the wrong direction. If $v_0 \sim N(0, \sigma^2 I)$ can show that for any $\sigma^2$, the head error will still typically be at least $\Omega(\|v_\star\|_2)$ This is an illustrative result, one can show similar results for other random initializations as well.

We first prove an anti-concentration lemma, which says that if $u$ is univariate Gaussian, then it cannot be too close to any particular constant $a$, no matter how the variance of the Gaussian is chosen.

**Lemma A.12.** *For some universal constant c, given $a > 0$, for all $\nu^2$ if $u \sim N(0, \nu^2)$ then for all $0 \leq \delta \leq 1$:*

$$P(|u - a| \leq c\delta a) \leq \delta \tag{A.119}$$

*Proof.* Consider $\delta$ such that $\delta \leq 1/10$. Then for all $u$ with $|u - a| \leq \delta a$, we have $u \geq 9a/10$. For all $u \geq 9a/10$, the density $f(u)$ is upper bounded (from the formula for the density of a Gaussian random variable) by:

$$f(u) \leq O(\frac{1}{v}\exp\frac{-9^2a^2}{2 \cdot 10^2 v^2}) \tag{A.120}$$

We can maximize this explicitly (e.g., use Mathematica or by taking the logarithm and then setting the derivative to 0) and we get for some universal constant $c' \geq 10$ (it is OK to choose a larger universal constant than needed):

$$f(u) \leq \frac{c'}{a} \tag{A.121}$$

Since the density is less than $c'/a$ and if $|u - a| \leq \delta a$ the size of the interval is $2\delta a$, we get for all $\delta \leq 1/10$:

$$P(|u - a| \leq \delta a) \leq \frac{2c'\delta a}{a} = 2c'\delta \tag{A.122}$$

Now, we substitute $\delta' = 2c'\delta$. We get for all $\delta' \leq 2c'/10$:

$$P(|u - a| \leq \frac{1}{2c'}\delta' a) \leq \delta' \tag{A.123}$$

Since $c' \geq 10$, $2c'/10 \geq 1$, so the statement is true for all $0 \leq \delta' \leq 1$. $\qquad\square$

We now bound the error in the head if the initialization is Gaussian. This bound holds for all initialization variances $\sigma^2$. Similar bounds can be shown for other (non-Gaussian) head initializations using similar anti-concentration arguments.

**Lemma A.13.** *For some universal constant c, for all $v_\star \in \mathbb{R}^k$ with $v_\star \neq 0$, $\sigma \in \mathbb{R}^+$, $\delta \in [0, 1]$, if $v_0 \sim N(0, \sigma^2 I_k)$, we have with probability at least $1 - \delta$:*

$$(\text{Head-Error}(v_0, v_\star))^2 := |(v_0^\top v_\star)^2 - (v_\star^\top v_\star)^2| \geq c\delta(v_\star^\top v_\star)^2 \tag{A.124}$$

*Proof.* First note that Head-Error$(v_0, v_\star) = $ Head-Error$(-v_0, v_\star)$ and $v_0$ is symmetric around 0 ($v_0$ and $-v_0$ have the same probability), and is almost surely not exactly 0. So without loss of generality, we can suppose that $v_0^\top v_\star \geq 0$.

**Suffices to bound $|v_0^\top v_\star - v_\star^\top v_\star|$:** We decompose the error:

$$|(v_0^\top v_\star)^2 - (v_\star^\top v_\star)^2| = |v_0^\top v_\star - v_\star^\top v_\star|(|v_0^\top v_\star + v_\star^\top v_\star|) \tag{A.125}$$

$$\geq |v_0^\top v_\star - v_\star^\top v_\star|(v_\star^\top v_\star)| \tag{A.126}$$

So we bound $|v_0^\top v_\star - v_\star^\top v_\star|$.

$v_0^\top v_\star$ **is normally distributed**: We note that $v_0^\top v_\star$ is distributed as:

$$v_0^\top v_\star \sim N(0, \sigma^2 v_\star^\top v_\star) \tag{A.127}$$

In other words, a normal with mean 0, and *variance* $\sigma_1^2 = \sigma^2 v_\star^\top v_\star$, and therefore standard deviation $\sigma_1 = \sigma\sqrt{v_\star^\top v_\star}$.

**Apply Gaussian anti-concentration lemma**: Then, from Lemma A.12, we have for some universal constant $c$ that with probability at least $1 - \delta$:

$$|v_0^\top v_\star - v_\star^\top v_\star| \geq c\delta v_\star^\top v_\star \tag{A.128}$$

So substituting this back into Equation A.125, we get the desired result:

$$|(v_0^\top v_\star)^2 - (v_\star^\top v_\star)^2| \geq c\delta(v_\star^\top v_\star)^2 \tag{A.129}$$

$\square$

### A.3.3 UPPER BOUNDING LINEAR PROBING ERROR

We showed a lower bound for the OOD error of fine-tuning in Theorem 3.2. To compare this with linear probing, we prove an *upper bound* on the OOD error of linear probing.

For completeness we include an elementary lemma (note that the condition that the matrices are tall is important for composing $\sigma_{\min}$, unlike for $\sigma_{\max}$, and we included this lemma to be careful about these conditions):

**Lemma A.14.** *Suppose we have two matrices $A$, $B$ of shape $(r,s)$ and $(s,t)$ respectively, and they are tall matrices so $r \geq s \geq t$. Then we have:*

$$\sigma_{\min}(AB) \geq \sigma_{\min}(A)\sigma_{\min}(B) \tag{A.130}$$

*Proof.* For a tall matrix $A$, we have:

$$\sigma_{\min}(A) = \min_{\|x\|_2 \leq 1} \|Ax\|_2 \tag{A.131}$$

So we have:

$$\sigma_{\min}(AB) = \min_{\|x\|_2 \leq 1} \|ABx\|_2 \geq \sigma_{\min}(A)\sigma_{\min}(B) \min_{\|x\|_2 \leq 1} \|x\|_2 \tag{A.132}$$

And $\min_{\|x\|_2 \leq 1} \|x\|_2 = 1$ which completes the proof. $\qquad\square$

**Lemma A.15.** *In the linear overparameterized setting, under the ID subspace assumption, fix arbitrary $P_z$. Then there exists $c_\delta$ such that with probability at least $1-\delta$, for all $d,n,m,k,w_\star$, feature extractors $B_\star, B_0$, and ID subspaces $S$ with corresponding $F$ (whose columns are orthonormal and form a basis for S), if $\cos\theta_{\max}(S,R) > 0$, we have:*

$$\sqrt{L_{\text{ood}}(v_{\text{lp}}^\infty, B_0)} \leq \left(\frac{c_\delta}{\cos\theta_{\max}(S,R)}\right)^2 d(B_0, B_\star)\|w*\|_2 \tag{A.133}$$

*If $P_z$ is isotropic Gaussian so $\mathcal{N}(0, I_m)$, then we derive a bound for $c_\delta$ analytically: if $n \geq 5m$ and $n \geq 10\log\frac{1}{\delta}$ then with probability at least $1-\delta$, the linear probing OOD error is upper bounded by:*

$$\sqrt{L_{\text{ood}}(v_{\text{lp}}^\infty, B_0)} \leq O\left(\frac{\log(n/\delta)}{(\cos\theta_{\max}(R,S))^2} d(B_0, B_\star)\|w_\star\|_2\right) \tag{A.134}$$

*Proof.* From the ID subspace assumption, the data matrix $X$ of shape $(n, d)$ can be written as $X = ZF^\top$ where $Z$ be a matrix of shape $(n, m)$ with each row $Z_i$ sampled iid from $P_z$, and $F$ is a matrix of shape $(d, m)$ whose columns are orthonormal and form a basis for the ID subspace $S$.

Let $\epsilon = \|B_\star - B_0\|_2 \leq$. We first prove the bounds for $\epsilon$, in terms of $d(B_0, B_\star)$ and we later handle the fact that the feature extractor distance involves the min over rotation matrices $U$: $d(B_0, B_\star) = \min_U \|UB_0 - B_\star\|_2$.

**Bounding key singular values**: Before proceeding with the proof, we examine a key quantity $XB_0^\top = ZF^\top B_0^\top$ which comes up in the Hessian of the loss function. We will show that this is invertible almost surely, and get a lower bound on its min singular value.

First, we examine the shapes of the matrices. $ZF^\top B_0^\top$ has shape $(n, d)$ where $Z$ has shape $(n, m)$ and $F^\top B_0^\top$ has shape $(m, k)$. Since $n \geq m > k$ we have that $Z$ and $F^\top B_0^\top$ are tall matrices, and so from Lemma A.14 we can write the min singular value of $ZF^\top B_0^\top$ as:

$$\sigma_{\min}(ZF^\top B_0^\top) \geq \sigma_{\min}(Z)\sigma_{\min}(F^\top B_0^\top) \tag{A.135}$$

Now from the definion of the principal angle (Definition 3.1), we have:

$$\sigma_{\min}(F^\top B_0^\top) = \cos\theta_{\max}(R,S) > 0. \tag{A.136}$$

Since we assumed $P_z$ has density in the ID subspace assumption, from Lemma 3 in Xie et al. (2021a) we get that for some $c_\delta' > 0$ that depends on $\delta$ and $P_z$, with probability at least $1-\delta$:

$$\sigma_{\min}(Z) \geq c_\delta' \tag{A.137}$$

Note that this also means that $\sigma_{\min}(ZF^\top B_0^\top) > 0$ and so $XB_0^\top = ZF^\top B_0^\top$ has full rank $k$ almost surely. This also implies that $B_0 X^\top X B_0^\top$ is a matrix of shape $(k,k)$ that is invertible almost surely.

**Main proof** Since $B_0 X^\top X B_0^\top$ is invertible almost surely, there is a unique global minimum (minimizing over $v$) to the loss optimized by linear probing:

$$\underset{v}{\mathrm{argmin}}\|X B_0^\top v - X B_\star^\top v_\star\|_2^2 = (B_0 X^\top X B_0^\top)^{-1} B_0 X^\top X B_\star^\top v_\star \tag{A.138}$$

We can see this by noting that the loss function on the LHS is strongly convex in $v$ since the Hessian $B_0 X^\top X B_0^\top$ is invertible. Then, gradient flow converges to the unique minimizer on the RHS, so:

$$v_{\mathsf{lp}}^\infty = (B_0 X^\top X B_0^\top)^{-1} B_0 X^\top X B_\star^\top v_\star \tag{A.139}$$

We now bound the square-root OOD error (taking the square root makes it easier to apply triangle inequalities), starting with the definition:

$$\sqrt{L_{\mathsf{ood}}(v_{\mathsf{lp}}^\infty, B_0)} = \|B_\star^\top v_\star - B_0^\top v_{\mathsf{lp}}^\infty\|_2 \tag{A.140}$$

$$\leq \|(B_\star^\top v_\star - B_0^\top v_\star) + (B_0^\top v_\star - B_0^\top v_{\mathsf{lp}}^\infty)\|_2 \tag{A.141}$$

$$\leq \underbrace{\|B_\star^\top v_\star - B_0^\top v_\star\|_2}_{(1)} + \underbrace{\|B_0^\top v_\star - B_0^\top v_{\mathsf{lp}}^\infty)\|_2}_{(2)} \tag{A.142}$$

We bound each term on the RHS of the last line. For term $(1)$:

$$\|B_\star^\top v_\star - B_0^\top v_\star\|_2 \leq \sigma_{\max}(B_\star - B_0)\|v_\star\|_2 \tag{A.143}$$

$$\leq \epsilon\|v_\star\|_2 \tag{A.144}$$

$$= \epsilon\|w_\star\|_2. \tag{A.145}$$

Where we note that $\|v_\star\|_2 = \|w_\star\|_2$ because $w_\star = B_\star^\top v_\star$ where the rows of $B_\star$ (columns of $B_\star^\top$) are orthonormal.

Let $\Sigma = X^\top X$. For term $(2)$, we first subtitute $v_{\mathsf{lp}}^\infty$ and do some algebra (again noting that $\|v_\star\|_2 = \|w_\star\|_2$) to get:

$$\|B_0^\top v_\star - B_0^\top v_{\mathsf{lp}}^\infty\|_2 = \|B_0^\top (B_0 \Sigma B_0^\top)^{-1} B_0 \Sigma B_0^\top v_\star - B_0^\top v_{\mathsf{lp}}^\infty\|_2 \tag{A.146}$$

$$= \|B_0^\top (B_0 \Sigma B_0^\top)^{-1} B_0 \Sigma (B_0 - B_\star)^\top v_\star\|_2 \tag{A.147}$$

$$\leq \sigma_{\max}(B_0^\top (B_0 \Sigma B_0^\top)^{-1} B_0 \Sigma)\sigma_{\max}(B_0 - B_\star)\|w_\star\|_2 \tag{A.148}$$

$$\leq \sigma_{\max}(B_0^\top (B_0 \Sigma B_0^\top)^{-1} B_0 \Sigma)\epsilon\|w_\star\|_2 \tag{A.149}$$

$$\leq \sigma_{\max}(B_0)^2 \sigma_{\max}(\Sigma)\frac{1}{\sigma_{\min}(B_0 \Sigma B_0^\top)}\epsilon\|w_\star\|_2 \tag{A.150}$$

$$\leq \frac{\sigma_{\max}(B_0)^2 \sigma_{\max}(X)^2}{\sigma_{\min}(X B_0^\top)^2}\epsilon\|w_\star\|_2 \tag{A.151}$$

$$= \frac{\sigma_{\max}(B_0)^2 \sigma_{\max}(Z F^\top)^2}{\sigma_{\min}(Z F^\top B_0^\top)^2}\epsilon\|w_\star\|_2 \tag{A.152}$$

$$\leq \frac{\sigma_{\max}(B_0)^2 \sigma_{\max}(Z)^2}{\sigma_{\min}(Z)^2 (\cos\theta_{\max}(R, S))^2}\epsilon\|w_\star\|_2 \tag{A.153}$$

$$\tag{A.154}$$

Where in the first line we substituted in the closed form for $v_{\mathsf{lp}}^\infty$ from Equation A.138, and in the last line we used the fact that $\sigma_{\max}(Z F^\top) \leq \sigma_{\max}(Z)$ since $F^\top$ has orthonormal rows, and $\sigma_{\min}(Z F^\top B^\top) = \sigma_{\min}(Z)\cos\theta_{\max}(R, S)$ as explained in Equation A.135 and Equation A.136.

So it suffices to bound the quantities in the RHS. Since $B_0$ has orthonormal rows, $\sigma_{\max}(B_0) = 1$.

**No Gaussian assumption**: For the first part of the Theorem (Equation A.133 where we make no Gaussian assumptions, but give a less quantitative bound), we just use the fact that $\sigma_{\max}(Z)$ is upper bounded almost surely, and $\sigma_{\min}(Z) \geq c_\delta'$ with probability at least $1 - \delta$. This implies that for some $c_\delta > 0$ with probability at least $1 - \delta$:

$$\sqrt{L_{\mathsf{ood}}(v_{\mathsf{lp}}^\infty, B_0)} \leq \left(\frac{c_\delta}{\cos\theta_{\max}(S, R)}\right)^2 \epsilon\|w*\|_2, \tag{A.155}$$

where $\epsilon = \|B_0 - B_\star\|_2$.

**Gaussian assumption**: For the second part of the Theorem (Equation A.134 where we assume $P_z$ is Gaussian), we use results in random matrix theory to lower bound and upper bound $\sigma_{\min}(Z)$. For the lower bound we use a result from Rudelson & Vershynin (2009) (see page 4, in the equation below Equation 1.11), since $Z \in \mathbb{R}^{n \times m}$ is a matrix with each entry sampled from $\mathcal{N}(0,1)$, we get for all $t > 0$:

$$\mathbb{P}(\sigma_{\min}(Z) \leq \sqrt{n} - \sqrt{m} - t) \leq e^{-t^2/2} \tag{A.156}$$

With a bit of algebra, this gives us that with probability at least $1 - \delta$:

$$\sigma_{\min}(Z) \geq \sqrt{n} - \sqrt{m} - \sqrt{2\log\frac{1}{\delta}} \tag{A.157}$$

We assumed $n \geq 5m$ and $n \geq 10\log\frac{1}{\delta}$, so we get:

$$\sigma_{\min}(Z) \geq O(\sqrt{n}) \tag{A.158}$$

The upper bound is a standard matrix concentration bound—we use the high probability bound in Theorem 4.1.1 from Tropp (2015) (see Section 4.2.2 which calculates the variance statistic for rectangular Gaussian matrices, also notice the square on the LHS below):

$$\sigma_{\max}(Z)^2 \leq O(n\log\frac{n}{\delta}) \tag{A.159}$$

Substituting the lower and upper bounds on $\sigma_{\min}(Z)$ into Equation A.146 we get:

$$\|B_0^\top v_\star - B_0^\top v_{\mathsf{lp}}^\infty\|_2 \leq O\Big(\frac{\log(n/\delta)}{(\cos\theta_{\max}(R,S))^2}\epsilon\|w_\star\|_2\Big) \tag{A.160}$$

Substituting into equation A.140, we have:

$$\sqrt{L_{\mathsf{ood}}(v_{\mathsf{lp}}^\infty, B_0)} \leq O\Big(\frac{\log(n/\delta)}{(\cos\theta_{\max}(R,S))^2}\epsilon\|w_\star\|_2\Big), \tag{A.161}$$

where $\epsilon = \|B_0 - B_\star\|_2$. Which completes the proof of the second part (Equation A.134).

**Handling the rotation matrix $U$**: We now handle the fact that the feature extractor distance involves the min over rotation matrices $U$: $d(B_0, B_\star) = \min_U \|UB_0 - B_\star\|_2$. Let $v_{\mathsf{lp}}^\infty(B_0)$ denote the linear probing head solution if we use a pretrained feature extractor $B_0$. We first note that for any $k$-by-$k$ rotation matrix $U$, we have:

$$L_{\mathsf{ood}}(v_{\mathsf{lp}}^\infty(B_0), B_0) = L_{\mathsf{ood}}(v_{\mathsf{lp}}^\infty(UB_0), UB_0). \tag{A.162}$$

This follows from using the closed form we derived above for $v_{\mathsf{lp}}^\infty(B_0)$ and some simple algebraic manipulation (e.g., recall that $U^{-1} = Y^\top$):

$$(UB_0)^\top v_{\mathsf{lp}}^\infty(UB_0) = (UB_0)^\top (UB_0 X^\top X B_0^\top U^\top)^{-1} UB_0 X^\top X B_\star^\top v_\star \tag{A.163}$$

$$= B_0^\top U^\top U (B_0 X^\top X B_0^\top)^{-1} U^\top U B_0 X^\top X B_\star^\top v_\star \tag{A.164}$$

$$= B_0^\top (U^\top U)(B_0 X^\top X B_0^\top)^{-1} (U^\top U) B_0 X^\top X B_\star^\top v_\star \tag{A.165}$$

$$= B_0^\top (B_0 X^\top X B_0^\top)^{-1} B_0 X^\top X B_\star^\top v_\star \tag{A.166}$$

$$= B_0^\top v_{\mathsf{lp}}^\infty(B_0) \tag{A.167}$$

So the final predictors in both cases, $(UB_0)^\top v_{\mathsf{lp}}^\infty(UB_0)$ and $B_0^\top v_{\mathsf{lp}}^\infty(B_0)$ are identical. This means that the OOD error $L_{\mathsf{ood}}(v, B) = \|B^\top v - B_\star^\top v_\star\|_2$ is the same in both cases.

This means that we can just take the min over all rotation matrices $U$ (where the first step follows since the identity matrix is a rotation matrix, and the second step is from Equation A.155):

$$L_{\mathsf{ood}}(v_{\mathsf{lp}}^\infty(B_0), B_0) \leq \min_U L_{\mathsf{ood}}(v_{\mathsf{lp}}^\infty(UB_0), UB_0) \tag{A.168}$$

$$\leq \min_U \Big(\frac{c(\delta)}{\cos\theta_{\max}(S,R)}\Big)^2 \|UB_0 - B_\star\|_2 \|w*\|_2^2 \tag{A.169}$$

$$= \Big(\frac{c(\delta)}{\cos\theta_{\max}(S,R)}\Big)^2 d(B_0, B_\star) \|w*\|_2^2, \tag{A.170}$$

which is as desired. We repeat the same thing for Equation A.161 to get Equation A.134 in the Theorem statement. $\qquad\square$

A.4 LP vs. FT (OOD), NON-ASYMPTOTIC RESULT FOR GAUSSIAN COVARIATES

Theorem 3.4 showed an asymptotic result: if the error $d(B_0, B_\star) \to 0$, then linear probing (LP) achieves better out-of-distribution (OOD) error than fine-tuning (FT). Here we give a more quantitative version of Theorem 3.4 for Gaussian covariates. The result can be extended to the case there each entry of $P_z$ is independent and identically distributed, mean-zero, constant non-zero variance, but instead of Gaussian is sub-Gaussian with constant sub-Gaussian variance / moment—this can be shown using Theorem 1.1 in Rudelson & Vershynin (2009), which is a different matrix concentration inequality.

We show that LP does better than FT out-of-distribution if the error is less than a specific quantity (in terms of the representation dimension $k$, and the angles between the ID subspace $S$ and the important pretrained directions $R_* = \text{rowspace}(B_\star)$).

**Theorem A.16.** *In the linear overparameterized setting, under the ID subspace assumption, assume the non-degeneracy conditions $\cos\theta_{\text{max}}(R_*, S) > 0$ and $\cos\theta_{\text{max}}(R_*, S^\perp) > 0$ where $R_* = \text{rowspace}(B_\star)$. Suppose the covariates are generated from a Gaussian distribution on the ID subspace $S$, so $P_z = \mathcal{N}(0, I_m)$. Let $\|w_\star\|_2$ be a fixed constant. Given failure probability $1 \le \delta > 0$, for all $w_\star, B_0, n, d, k, \epsilon$, if $n \ge 5m$, and $n \ge 10\log\frac{1}{\delta}$, if the error of the pretrained representation is not too high:*

$$d(B_0, B_\star) < O\Big(\frac{\cos\theta_{\text{max}}(R_*, S^\perp)(\cos\theta_{\text{max}}(R_*, S))^2 \delta^2}{\sqrt{k}\log(n/\delta)}\Big), \tag{A.171}$$

*then with probability at least $1 - \delta$, the OOD error of linear probing is lower (better) than for fine-tuning at all time steps $t \ge 0$ in the fine-tuning trajectory:*

$$L_{\text{ood}}(v_{\text{lp}}^{\infty i}, B_0^i) < \inf_{t \ge 0} L_{\text{ood}}(v_{\text{lp}}^{\infty i}, B_0^i). \tag{A.172}$$

*Proof.* Let $\epsilon = d(B_0, B_\star)$. We first note that the condition in Equation A.171 implies that $d(B_0, B_\star) < O(\cos\theta_{\text{max}}(R_*, S^\perp))$ and $d(B_0, B_\star) < O(\cos\theta_{\text{max}}(R_*, S))$. This is because the cosine angles are between 0 and 1, $\delta$ is between 0 and 1, and $k$ and $n$ are at least 1. We now simplify and combine the linear probing and fine-tuning bounds.

Let $R_0 = \text{rowspace}(B_0)$. Warning: note that the Equation A.171 in the Theorem statement assumes conditions on the angles between $R_*$ (corresponding to the optimal representation) and the ID subspace $S$. However, our results that bounded the fine-tuning (Theorem 3.2) and linear probing (Lemma A.134) errors require conditions on the angles between $R_0$ (corresponding to the representation that linear probing and fine-tuning use) and $S$. So we have to be careful about this distinction, and use Lemma A.10 to relate the two, which we do below.

**Fine-tuning**: From Theorem 3.2, we get:

$$\sqrt{L_{\text{ood}}(v_{\text{ft}}(t), B_{\text{ft}}(t))} \ge O\Big(\frac{\cos\theta_{\text{max}}(R_0, S^\perp)}{\sqrt{k}} \frac{\min(\varphi, \varphi^2/\|w_\star\|_2)}{(1 + \|w_\star\|_2)^2}\Big) - \epsilon. \tag{A.173}$$

Where $\varphi$ is the head-error, which we lower bounded in Lemma A.13—substituting this bound and noting that $\min(\varphi, \varphi^2) = O(\varphi^2)$, $\|v_\star\|_2 = \|w_\star\|_2$ (which we assumed is a constant), this gives us:

$$\sqrt{L_{\text{ood}}(v_{\text{ft}}(t), B_{\text{ft}}(t))} \ge O\Big(\frac{\cos\theta_{\text{max}}(R_0, S^\perp)}{\sqrt{k}} \delta^2\Big) - \epsilon \tag{A.174}$$

Now, since $d(B_0, B_\star) = \epsilon$, we use Lemma A.10 to get that:

$$\cos\theta_{\text{max}}(R_0, S^\perp) \ge \cos\theta_{\text{max}}(R_*, S^\perp) - \epsilon \tag{A.175}$$

Substituting this into Equation A.174, we get (notice the $R_*$ instead of $R_0$ below):

$$\sqrt{L_{\text{ood}}(v_{\text{ft}}(t), B_{\text{ft}}(t))} \ge O\Big(\frac{\cos\theta_{\text{max}}(R_*, S^\perp) - \epsilon}{\sqrt{k}} \delta^2\Big) - \epsilon \tag{A.176}$$

Since $\epsilon \le O(\cos\theta_{\text{max}}(R_*, S^\perp))$, this can be simplified to:

$$\sqrt{L_{\text{ood}}(v_{\text{ft}}(t), B_{\text{ft}}(t))} \ge O\Big(\frac{\cos\theta_{\text{max}}(R_*, S^\perp)}{\sqrt{k}} \delta^2\Big) - \epsilon \tag{A.177}$$

**Linear probing**: From Lemma A.134, we get:

$$\sqrt{L_{\mathsf{ood}}(v_{\mathsf{lp}}^{\infty},B_0)} \leq O\Big( \frac{\log(n/\delta)}{(\cos\theta_{\mathsf{max}}(R_0,S))^2} \epsilon \|w_{\star}\|_2 \Big) \tag{A.178}$$

Again, we use Lemma A.10 to get:

$$\cos\theta_{\mathsf{max}}(R_0,S) \geq \cos\theta_{\mathsf{max}}(R_*,S) - \epsilon \tag{A.179}$$

Substituting into Equation A.178, and using the fact that $\epsilon \leq O(\cos\theta_{\mathsf{max}}(R_*,S))$, and since we assumed $\|w_{\star}\|_2$ is a constant, we get:

$$\sqrt{L_{\mathsf{ood}}(v_{\mathsf{lp}}^{\infty},B_0)} \leq O\Big( \frac{\log(n/\delta)}{(\cos\theta_{\mathsf{max}}(R_*,S))^2} \epsilon \Big) \tag{A.180}$$

**Combining the two**: We want to show that the OOD error of LP is less than for fine-tuning:

$$O\Big( \frac{\log(n/\delta)}{(\cos\theta_{\mathsf{max}}(R_*,S))^2} \epsilon \Big) \leq O\Big( \frac{\cos\theta_{\mathsf{max}}(R_*,S^{\perp})}{\sqrt{k}} \delta^2 \Big) - \epsilon \tag{A.181}$$

We can bring the $\epsilon$ to the LHS, so this is equivalent to showing:

$$O\Big( \frac{\log(n/\delta)}{(\cos\theta_{\mathsf{max}}(R_*,S))^2} \epsilon \Big) + \epsilon \leq O\Big( \frac{\cos\theta_{\mathsf{max}}(R_*,S^{\perp})}{\sqrt{k}} \delta^2 \Big) \tag{A.182}$$

Since $\log(n/\delta) \geq 1$ and $\cos\theta_{\mathsf{max}}(R_*,S))^2$ is between 0 and 1, this is equivalent to folding the $\epsilon$ inside the big-oh on the LHS:

$$O\Big( \frac{\log(n/\delta)}{(\cos\theta_{\mathsf{max}}(R_*,S))^2} \epsilon \|w_{\star}\|_2 \Big) \leq O\Big( \frac{\cos\theta_{\mathsf{max}}(R_*,S^{\perp})}{\sqrt{k}} \delta^2 \Big) \tag{A.183}$$

But assuming the condition on $\epsilon$ in Equation A.171 of the Theorem statement, this is easy to show with a bit of algebra. $\qquad\square$

## A.5   PRINCIPAL ANGLES ARE LIKELY NON-ZERO

In Theorems 3.2, 3.4, and 3.5, we assumed the cosine of the largest principal angle between the representations and ID subspace (or complement of the ID subspace) was non-zero. For example, Theorem 3.4 assumed the largest principal angle between $R_* = \mathsf{rowspace}(B_{\star})$ and the ID subspace $S$ is non-zero, and similarly for the angle between $R_*$ and $S^{\perp}$. Having an angle of 0 is a degenerate condition. As an example, look at Figure 2—here the input dimension $d = 2$, the representation dimension $k = 1$, and the ID subspace $S$ has dimension 1. The only way these angles can be 0 is if $B_{\star}^{\top}$ is exactly in the same direction as $S$ or $S^{\perp}$, which seems like too much of a coincidence. intuitively, if nature introduces even a small amount of randomness in either the optimal representation or ID subspace, the angle will be non-zero.

This example was in two dimensions—to make this intuition a bit more formal in higher dimensions, we prove a simple claim. Lemma A.17 shows that if the $S$ is a randomly selected $m$ dimensional subspace, then the angles $\cos\theta_{\mathsf{max}}(R_*,S)$ and $\cos\theta_{\mathsf{max}}(R_*,S^{\perp})$ are non-zero (and we get quantitative lower bounds on them).

**Lemma A.17.** *Let $R$ be a fixed $k$ dimensional subspace, and let $S$ be a uniformly random $m$ dimensional subspace (formally, a uniform measure on the Grassmannian manifold) in $\mathbb{R}^d$ with $m > k$. Then with probability at least $1 - \delta$,*

$$\cos\theta_{\mathsf{max}}(R,S) \geq \frac{\sqrt{m} - \sqrt{k} - \sqrt{2\log\frac{1}{\delta}}}{\sqrt{d\log\frac{2d}{\delta}}} \tag{A.184}$$

*In addition, we get that $\cos\theta_{\mathsf{max}}(R,S) > 0$ almost surely (with probability 1).*

*If $m \geq 5k$ and $m \geq 10\log\frac{1}{\delta}$, then we get with probability at least $1 - \delta$:*

$$\cos\theta_{\mathsf{max}}(R,S) \geq O\Big( \sqrt{\frac{m}{d\log\frac{2d}{\delta}}} \Big) \tag{A.185}$$

*Recall that big-oh notation here means that the RHS is true for some universal constant (independent of any other problem parameters).*

*Proof.* Note that principal angles are invariant if we rotate $R$ and $S$ by the same rotation matrix $U$. That is, if we let $U \in R^{d \times d}$ be a rotation matrix, and $E \in \mathbb{R}^{d \times k}$, $F \in \mathbb{R}^{d \times m}$ have orthonormal columns which form a basis for $R$ and $S$ respectively, then we have:

$$\cos\theta_{\mathsf{max}}(R,S) = \sigma_k(E^\top F) = \sigma_k((UE)^\top(UF)) \tag{A.186}$$

This symmetry means that we can fix $S$ and instead consider $R$ to be a uniform random $k$ dimensional subspace on the Grassmannian manifold. Without loss of generality, we can also fix $S$ to be the span of the first $m$ standard basis vectors: $(e_1,...,e_m)$, where $e_i \in \mathbb{R}^d$ has a 1 in the $i$-th entry and a 0 in every other entry.

Equivalently, let $M_R$ be a $d$-by-$k$ matrix, where each column is sampled independently from $N(0,I_d)$—since the columns of $M_R$ span a uniformly random $k$-dimensional subspace, we can let $R$ be range of $M_R$. This is equivalent to sampling each entry of $M_R$ from $N(0,1)$.

Let $c = \cos\theta_{\mathsf{max}}(R,S)$. From Lemma A.2, $c$ can be written as:

$$c = \min_{r \in R, \|r\|_2 = 1} \|F^\top r\|_2 = \min_{r \in R, \|r\|_2 \geq 1} \|F^\top r\|_2 \tag{A.187}$$

Since $R$ is the range of $M_R$, any $r \in R$ can be written as $r = M_R \lambda$ for some $\lambda \in \mathbb{R}^k$. We first show that $\|\lambda\|_2$ cannot be much smaller than $\|r\|_2$. This is because:

$$\|r\|_2 = \|M_R \lambda\|_2 \leq \sigma_{\max}(M_R)\|\lambda\|_2 \tag{A.188}$$

So this gives us:

$$\|\lambda\|_2 \geq \frac{\|r\|_2}{\sigma_{\max}(M_R)} \tag{A.189}$$

So every $r \in R$ can be written as $M_R \lambda$ where $\|\lambda\|_2$ is lower bounded as above.

We now simplify the definition of $c$, starting from Equation A.187.

$$c = \min_{r \in R, \|r\|_2 \geq 1} \|F^\top r\|_2 \tag{A.190}$$

$$\geq \min_{\|\lambda\| \geq 1/\sigma_{\max}(M_R)} \|F^\top M_R \lambda\|_2 \tag{A.191}$$

$$\geq \min_{\|\lambda\| \geq 1/\sigma_{\max}(M_R)} \sigma_{\min}(F^\top M_R)\|\lambda\|_2 \tag{A.192}$$

$$= \frac{\sigma_{\min}(F^\top M_R)}{\sigma_{\max}(M_R)} \tag{A.193}$$

So now we want to lower bound the ratio of two random matrices. We note that $F^\top M_R$ is a matrix of size $(m,k)$ with each entry sampled independently from $N(0,1)$ (this is because $F^\top$ simple selects the first $m$ rows of $M_R$). $M_R$ is a matrix of size $(d,k)$ with each entry sampled independently from $N(0,1)$.

Now, as in the Gaussian assumption step of the proof of Lemma A.15, we can apply standard matrix concentration bounds (page 4, below Equation 1.11, in Rudelson & Vershynin (2009) for the bound on $\sigma_{\min}$, and Theorem 4.1.1 in Tropp (2015) for the bound on $\sigma_{\max}$). We get that with probability at least $1 - \delta$:

$$\sigma_{\min}(F^\top M_R) \geq \sqrt{m} - \sqrt{k} - \sqrt{2\log\frac{1}{\delta}} \tag{A.194}$$

$$\sigma_{\max}(M_R) \leq \sqrt{d\log\frac{2d}{\delta}} \tag{A.195}$$

Note that we can use alternate bounds for $\sigma_{\min}$ in Rudelson & Vershynin (2009) that are sometimes tighter.

For the ratio of the two, we get that with probability at least $1 - \delta$, we have:

$$c \geq \frac{\sigma_{\min}(F^\top M_R)}{\sigma_{\max}(M_R)} \geq \frac{\sqrt{m} - \sqrt{k} - \sqrt{2\log\frac{2}{\delta}}}{\sqrt{d\log\frac{2d}{\delta}}} \tag{A.196}$$

For interpretability, ignoring log factors this is approximately:

$$c \gtrsim \frac{\sqrt{m} - \sqrt{k}}{\sqrt{d}} \tag{A.197}$$

The result when $m \geq 5k$ and $n \geq 10 \log \frac{2}{\delta}$ follows with simple algebra.

For the result where we show $\cos\theta_{\max}(R,S) > 0$ almost surely, we recall that $F^\top M_R$ is a matrix of size $(m,k)$ with each entry sampled independently from $N(0,1)$. Then applying Lemma 3 in Xie et al. (2021a), we get that $\sigma_{\min}(F^\top M_R) > 0$ almost surely. Since $\sigma_{\max}(M_R)$ is finite, this gives us $\cos\theta_{\max}(R,S) > 0$ almost surely.

$\square$

In our case, the dimension of the ID subspace $S$ is $m$, and the dimension of $R_* = \mathrm{rowspace}(B_\star)$ is $k$, with $k < m$ and $k < d - m$. If $S$ is a uniformly random $m$-dimensional subspace, then $S^\perp$ is a uniformly random $d - m$ dimensional subspace. In this case, Lemma A.17 tells us that $\cos\theta_{\max}(R_*, S) > 0$ and $\cos\theta_{\max}(R_*, S^\perp) > 0$ almost surely, and gives us quantitative lower bounds for these angles.

### A.6 LP vs. FT (ID)

We prove Proposition 3.5, where we show that if the representation is imperfect, then fine-tuning does better than linear probing, in-distribution.

**Restatement of Proposition 3.5.** *In the linear overparameterized setting, under the ID subspace assumption (Assumption 3.3), let $R_0 = rowspace(B_0)$, and $R_{\mathsf{aug}} = Span(\{w_\star\} \cup R_0)$. Suppose $w_\star \notin R_0$, $\cos\theta_{\max}(S, R_{\mathsf{aug}}) \neq 0$, and that fine-tuning converges to a local minimum of its loss, then fine-tuning does better ID almost surely: $L_{\mathsf{id}}(v_{\mathsf{ft}}^\infty, B_{\mathsf{ft}}^\infty) < L_{\mathsf{id}}(v_{\mathsf{lp}}^\infty, B_0)$ with probability 1 (over the randomness of the training examples).*

*Proof.* **Fine-tuning gets** $0$ **ID loss**: It is well known from prior work (Laurent & von Brecht, 2018) that all local minima are global for optimizing two layer linear networks under convex losses (which is our setting), so if fine-tuning converges to a local minimum, it actually converges to a global minimum of the train loss. Since there exists parameters that achieve $0$ loss on the training data (namely, $B_\star, v_\star$), this means fine-tuning gets $0$ loss on the training data as well. So for all training examples $x$ (that is, rows of $X$):

$$v_{\mathsf{ft}}^{\infty \top} B_{\mathsf{ft}}^\infty x = w_\star^\top x. \tag{A.198}$$

Since the models are linear, this implies that fine-tuning gets all examples in the span of the training examples correct as well. Since $P_z$ has density, and the number of training examples $n$ is at least as large as the ID subspace dimension $m$, the training examples span the ID subspace almost surely, so fine-tuning gets every example in $x \in S$ correct almost surely, giving us:

$$L_{\mathsf{id}}(v_{\mathsf{ft}}^\infty, B_{\mathsf{ft}}^\infty) = 0 \tag{A.199}$$

**Linear probing gets positive ID loss**: Lemma A.20 shows that the ID error of linear probing is greater than zero under the same assumptions as this Proposition, so

$$L_{\mathsf{id}}(v_{\mathsf{lp}}^\infty, B_0) > 0, \tag{A.200}$$

which finishes the proof.

$\square$

We now state and prove the Lemmas that we used to lower bound the ID error of linear probing.

Lemma A.18 gives conditions for when the projection $F^\top w$ of a vector $w$ is not contained in the projection $\mathrm{Range}(F^\top E_0)$ of the column space of a matrix $E_0$.

**Lemma A.18.** *Let $w \in \mathbb{R}^d$ be a vector and $F \in \mathbb{R}^{d \times m}, E_0 \in \mathbb{R}^{d \times k}, E_{\mathsf{aug}} \in \mathbb{R}^{d \times (k+1)}$ have orthonormal columns, with $Range(E_{\mathsf{aug}}) = Span(\{w\} \cup Range(E_0))$. If $m > k$, we have:*

$$F^\top E_{\mathsf{aug}} \text{ is full rank} \tag{A.201}$$

$$\overset{(a)}{\Longrightarrow} F^\top E_{\mathsf{aug}} \text{ has higher rank than } F^\top E_0 \tag{A.202}$$

$$\overset{(b)}{\Longleftrightarrow} F^\top w \notin Range(F^\top E_0) \tag{A.203}$$

*Proof.* The proof of (a) is clear—$F^\top E_{\mathsf{aug}} \in \mathbb{R}^{m \times (k+1)}$ has rank $k + 1$ (since it is full rank and $m \geq k+1$), but $F^\top E_{\mathsf{aug}} \in \mathbb{R}^{m \times k}$ has rank at most $k$ and is therefore lower rank. The assumption that $m > k$ is crucial here.

For (b), let $a_1, \dots, a_k$ be the columns of $E_0$, which form a basis for $\mathrm{Range}(E_0)$. Then $F^\top a_1, \dots, F^\top a_k, F^\top w$ spans $\mathrm{Range}(F^\top E_{\mathsf{aug}})$, while $F^\top a_1, \dots, F^\top a_k$ spans $\mathrm{Range}(F^\top E_0)$. So (notice the first list of vectors has an additional $F^\top w$) this means that $\dim(\mathrm{Range}(F^\top E_{\mathsf{aug}})) \neq \dim(\mathrm{Range}(F^\top E_0))$ iff $F^\top w$ is linearly independent from the rest, that is, $F^\top w \notin \mathrm{Range}(F^\top E_0)$. Note that the rank of a matrix is the dimension of its range (column space), that is, $\dim(\mathrm{Range}(A)) = \mathrm{rank}(A)$ so this is what we wanted to show. $\square$

The next Lemma says that if the projection $F^\top w_\star$ of the optimal linear model $w_\star$ onto the ID subspace $S$, is not contained in the projection $\mathrm{Range}(F^\top E_0)$ of the features, then linear probing incurs non-zero ID error.

**Lemma A.19.** *In the linear overparameterized setting, under the ID subspace assumption, if $F^\top w_\star \notin Range(F^\top E_0)$, then $L_{\mathsf{id}}(v_{\mathsf{lp}}^\infty, B_0) > 0$, where $E_0 \in \mathbb{R}^{d \times k}$ and $F \in \mathbb{R}^{d \times m}$ have orthonormal columns that form a basis for the feature rowspace $R_0 = rowspace(B_0)$ and ID subspace $S$ respectively.*

*Proof.* We prove the contrapositive. Suppose $L_{\mathsf{id}}(v_{\mathsf{lp}}^\infty, B_0) = 0$. This means that:

$$L_{\mathsf{id}}(v_{\mathsf{lp}}^\infty, B_0) = \underset{x \sim P_{\mathsf{id}}}{\mathbb{E}} [(v_\star^\top B_\star x - v_{\mathsf{lp}}^{\infty \top} B_0 x)^2] = 0 \tag{A.204}$$

Since the squared error is always non-negative, this means that $v_{\mathsf{lp}}^{\infty \top} B_0 x = w_\star^\top x$ almost surely when $x \sim P_{\mathsf{id}}$ (recall that we defined $w_\star = B_\star^\top v_\star$). Recall $P_{\mathsf{id}}$ is defined as: first pick $z \in P_z$ (which has density) and then output $x = Fz$. Since $P_z$ has density, this implies that we get all examples in the ID subspace $S$ correct:

$$v_{\mathsf{lp}}^{\infty \top} B_0 x = w_\star^\top x \text{ for all } x \in S. \tag{A.205}$$

Since the columns of $F$ form an orthonormal basis for $S$, this gives us (since each column of $F$ is in $S$):

$$v_{\mathsf{lp}}^{\infty \top} B_0 F = w_\star^\top F. \tag{A.206}$$

Note that the rows of $B_0$ also form an orthonormal basis for $R_0$ just like the columns of $E_0$. So we can choose $v$ with $v^\top E_0^\top = v_{\mathsf{lp}}^{\infty \top} B_0$. Then we have:

$$v^\top E_0^\top F = w_\star^\top F \Leftrightarrow F^\top E_0 v = F^\top w_\star \tag{A.207}$$

$$\Leftrightarrow F^\top w_\star \in \mathrm{Range}(F^\top E_0), \tag{A.208}$$

where we took the transpose of both sides in the first step. This finishes the proof of the contrapositive. $\square$

Finally, Lemma A.20 combines Lemma A.18 and Lemma A.19 to give a more interpretable condition for the ID error of linear probing: when the ID subspace $S$ has some components along the optimal linear model $w_\star$ and the feature rowspace $R_0$, then linear probing has non-zero error. This is measured in terms of the principal angle $\cos\theta_{\mathsf{max}}(R_{\mathsf{aug}}, S)$ between the ID subspace $S$ and $R_{\mathsf{aug}}$ which is the span of $R_0$ combined with $w_\star$. This angle will typically be non-zero—as an illustrative example, from Lemma A.17 we have that this angle will be non-zero almost surely if the ID subspace $S$ is a uniformly random subspace.

**Lemma A.20.** *In the linear overparameterized setting, under the ID subspace assumption, let $R_0 = rowspace(B_0)$, and $R_{\mathsf{aug}} = Span(\{w_\star\} \cup R_0)$. If $w_\star \notin R_0$ and $\cos\theta_{\mathsf{max}}(R_{\mathsf{aug}}, S) > 0$, then $L_{\mathsf{id}}(v_{\mathsf{lp}}^\infty, B_0) > 0$.*

*Proof.* After a bit of setup, the proof simply combines Lemma A.18 and Lemma A.19. If $w_\star \notin R_0$, then $R_{\mathsf{aug}}$ has dimension $k+1$. Let $E_{\mathsf{aug}} \in \mathbb{R}^{d \times (k+1)}, F \in \mathbb{R}^{d \times m}$ have orthonormal columns which form a basis for $R_{\mathsf{aug}}$ and $S$ respectively. We assumed $\cos\theta_{\mathsf{max}}(R_{\mathsf{aug}}, S) = \sigma_{\mathsf{min}}(F^\top E_{\mathsf{aug}}) > 0$ which means that $F^\top E_{\mathsf{aug}}$ is full rank. The ID subspace assumption assumes that $m > k$. So from Lemma A.18, $F^\top w_\star \notin \mathrm{Range}(F^\top E_0)$ where $E_0 \in \mathbb{R}^{d \times k}$ has orthonormal columns that form a basis for $R_0$. Then from Lemma A.19, $L_{\mathsf{id}}(v_{\mathsf{lp}}^\infty, B_0) > 0$. $\square$

A.7   LP-FT

We start by showing a simple proposition, that if the initial feature extractor is perfect, then linear probing recovers the optimal weights.

**Proposition A.21.** *In the overparameterized linear setting, let $R = rowspace(B_0)$. If $B_0 = B_\star$, and $\cos\theta_{\mathsf{max}}(S,R) > 0$, then $L_{\mathsf{ood}}(v_{\mathsf{lp}}^\infty, B_0) = 0$ for all $t$.*

*Proof.* We first show that because $\cos\theta_{\mathsf{max}}(R,S) > 0$, the training loss for linear probing is strongly convex. Recall that the training loss is:

$$\widehat{L}(v,B) = \|XB^\top v - Y\|_2^2 \tag{A.209}$$

Linear probing keeps $B$ fixed as $B_0 = B_\star$ and only tunes $v$, so we are interested in the Hessian of the loss with respect to $v$ evaluated at $v, B_\star$:

$$\mathrm{Hess}_v\widehat{L}(v,B_\star) = 2(B_\star X^\top)(B_\star X^\top)^\top \tag{A.210}$$

For strong convexity, it suffices to show that the min singular value of the Hessian is bounded away from 0 by a constant. Recall the definition of $\cos\theta_{\mathsf{max}}(R,S)$. For some $F$ whose columns form an orthonormal basis for $S$, we have (since the rows of $B_\star$ form an orthonormal basis for $R$):

$$\sigma_k(B_\star F) = \cos\theta_{\mathsf{max}}(R,S) > 0 \tag{A.211}$$

Note that $B_\star F$ is a $k$-by-$n$ matrix, so if the $k$-th singular value is positive it must be full rank. Since the columns of $X^\top$ span $F$ (since we defined $F$ to be such that the columns of $F$ are an orthonormal basis for $S$, i.e. the rows of $X$), this means $B_\star X^\top$ is rank $k$. But that means the Hessian $(B_\star X^\top)(B_\star X^\top)^\top$ is rank $k$ as well. So the linear probing loss is strongly convex.

Since the loss is strongly convex, there is a unique minimizer, and gradient flow converges to that. However, since we are in the well-specified setting, we know the training loss is:

$$\widehat{L}(v,B_\star) = \|XB_\star^\top v - XB_\star^\top v_\star\|_2^2 \tag{A.212}$$

So $v = v_\star$ achieves 0 loss and must be the (unique) minimizer. Therefore we have shown that linear probing converges to the unique minimizer $v_{\mathsf{lp}}^\infty = v_\star$, which attains 0 loss, as desired.

Note that the entire proof works out if $B_0 = UB_\star$ for some rotation matrix $U$. In that case, the Hessian becomes $2U(B_\star X^\top)(B_\star X^\top)^\top U^\top$ which is still rank $k$, since multiplying by square rotation matrices does not change the rank. In this case, the minimizer of the loss is $v = Uv_\star$, since $(UB_\star)^\top(Uv_\star) = B_\star^\top v_\star$. So linear probing converges to $v_{\mathsf{lp}}^\infty = Uv_\star$, which achieves 0 loss, as desired. ☐

**Restatement of Proposition 3.6.** *Given perfect pretrained features $B_0 = UB_\star$ for some rotation $U$. Let $R_0 = rowspace(B_0)$. Under the non-degeneracy conditions $\cos\theta_{\mathsf{max}}(R_0,S) \neq 0, \cos\theta_{\mathsf{max}}(R_0,S^\perp) \neq 0$:*

$$\forall t, L_{\mathsf{ood}}(B_{\mathsf{ft}}(t)^\top v_{\mathsf{ft}}(t)) > 0, \text{ if } v_0 \sim \mathcal{N}(0,\sigma^2 I) \text{ is randomly initialized (FT)}, \tag{A.213}$$

$$\forall t, L_{\mathsf{ood}}(B_{\mathsf{ft}}(t)^\top v_{\mathsf{ft}}(t)) = 0, \text{ if } v_0 \text{ is initialized to } v_{\mathsf{lp}}^\infty \text{ (LP-FT)}. \tag{A.214}$$

*Proof.* We first use Proposition A.21, which in the proof we showed still works if $B_0 = UB_\star$ for some rotation matrix $U$ (which doesn't have to be identity). We get that $v_{\mathsf{lp}}^\infty = Uv_\star$. Then we have $B_0^\top v_{\mathsf{lp}}^\infty = B_\star^\top v_\star = w_\star$.

We now just show that the gradients with respect to the training loss $\widehat{L}$ at $(v_{\mathsf{lp}}^\infty, B_0)$ is 0, so gradient flow does not update the parameters at all.

The training loss is:

$$\widehat{L}(v,B) = \|XB^\top v - XB_\star^\top v_\star\|_2^2 \tag{A.215}$$

The derivative with respect to $v$ is:

$$\partial_v\widehat{L}(v,B) = 2BX^\top(XB^\top v - XB_\star^\top v_\star) \tag{A.216}$$

Then since $B_0^\top v_{\mathsf{lp}}^\infty = B_\star^\top v_\star$, we have:

$$\partial_v\widehat{L}(v_{\mathsf{lp}}^\infty, B_0) = 0 \tag{A.217}$$

Next, the derivative with respect to $B$ is:

$$\partial_B \widehat{L}(v,B) = 2v(XB^\top v - XB_\star^\top v_\star)^\top X \tag{A.218}$$

Then since $B_0^\top v_{\mathsf{lp}}^\infty = B_\star^\top v_\star$, we have:

$$\partial_B \widehat{L}(v_{\mathsf{lp}}^\infty, B_0) = 0 \tag{A.219}$$

So since both the derivatives are 0, we have $\partial_t v_{\mathsf{ft}}(t) = 0$ and $\partial_B B_{\mathsf{ft}}(t) = 0$, which means the parameters don't change at all—at all times $t$ we have $v_{\mathsf{ft}}(t) = Uv_\star$ and $B_{\mathsf{ft}}(t) = UB_\star$ which gives us zero OOD loss: $L_{\mathsf{ood}}(B_{\mathsf{ft}}(t)^\top v_{\mathsf{ft}}(t)) = 0$ as desired. □

## B  MORE INFORMATION ON EXPERIMENTS

In this Appendix, we include more details on the datasets, pretraining methods, and adaptation methods. We also include the OOD accuracies for fine-tuning and linear probing if we early stop and choose the learning rate based on OOD data, where we see that linear probing is still typically better than fine-tuning OOD. Finally, we include results for additional baselines, pretraining models, and conclude with a discussion about the effective robustness of LP-FT.

### B.1  OVERVIEW OF DATASETS

We first give an overview of the datasets used in our paper, before diving into more details of the exact training procedures (e.g., number of epochs, pretraining method, etc). The datasets we use are:

- **DomainNet** (Peng et al., 2019) is a standard domain adaptation dataset. Here, our ID dataset contains "sketch" images (e.g., drawings of apples, elephants, etc), and the OOD dataset contains "real", "clipart", and "painting" images of the same categories. We use the version of the dataset from Tan et al. (2020).

- **Living-17** and **Entity-30** are sub-population shift datasets from the BREEDS benchmark (Santurkar et al., 2020). In Living-17 the goal is to classify an image as one of 17 animal categories such as "bear"—for example, the ID dataset contains images of black bears and sloth bears and the OOD dataset has images of brown bears and polar bears. In Entity-30 the goal is to classify an image as one of 30 entities such as "fruit" or "insect".

- **FMoW Geo-shift** is adapted from the satellite remote sensing dataset *Functional Map of the World* (Christie et al., 2018; Koh et al., 2021). The goal is to classify a satellite image into one of 62 categories such as "impoverished settlement" or "hospital". Our ID dataset contains images from North America, and the OOD dataset contains images from Africa and Europe.

- **CIFAR-10 → STL** is a standard domain adaptation dataset (French et al., 2018), where the ID is CIFAR-10 (Krizhevsky, 2009), and the OOD is STL (Coates et al., 2011). The task is to classify an image into one of 10 categories such as "dog", "cat", or "airplane"—as usual, we remove the "monkey" class in STL since CIFAR-10 has no "monkey" images.

- **CIFAR-10 → CIFAR-10.1** (Recht et al., 2018) is a dataset collected using a very similar protocol to CIFAR-10, and the authors describe it as "a minute distributional shift". The hope is that a classifier trained on CIFAR-10 gets high accuracy on CIFAR-10.1.

- **ImageNet-1K** (Russakovsky et al., 2015) is a large scale dataset containing over a million images, where the goal is to classify an image into one of 1000 categories such as "Yorkshire terrier", "Labrador retriever", "acoustic guitar", "library", "school bus", etc. We fine-tune on ImageNet as the ID dataset, and evaluate on four standard OOD datasets: **ImageNetV2** (Recht et al., 2019), **ImageNet-R** (Hendrycks et al., 2020), **ImageNet-A** (Hendrycks et al., 2019b), and **ImageNet-Sketch** (Wang et al., 2019).

### B.2  DATASET AND METHOD DETAILS

We use a diverse range of datasets and pretraining strategies.

- **CIFAR-10 → STL**: We fine-tune or linear probe on CIFAR-10 (Krizhevsky, 2009) and test on STL (Coates et al., 2011). This is a benchmark used in domain adaptation papers (French et al., 2018). CIFAR-10 and STL share 9 classes, so we follow the common practice of omitting the unshared class in STL (which is the 'monkey' class) when reporting accuracies. We use a publicly available MoCo-v2 ResNet-50 checkpoint pretrained on unlabeled examples from ImageNet-1k (Russakovsky et al., 2015), and fine-tune for 20 epochs.

Table 3: **OOD accuracies** with 90% confidence intervals over 3 runs, for each of the three OOD domains in the split of DomainNet used by Tan et al. (2020); Prabhu et al. (2021). LP does better than FT across the board, and LP-FT does the best.

|                | Real         | Painting     | Clipart      |
|----------------|--------------|--------------|--------------|
| Fine-tuning    | 55.29 (0.52) | 50.26 (0.98) | 60.93 (2.15) |
| Linear probing | **87.16 (0.18)** | 74.50 (0.58) | 77.29 (0.12) |
| LP-FT          | **86.82 (0.51)** | **75.91 (0.73)** | **79.48 (0.90)** |

- **DomainNet**: We use the dataset splits in Tan et al. (2020) which is also used by follow-up work, e.g., in Prabhu et al. (2021). This is different from the original version of the DomainNet dataset (Peng et al., 2019), specifically Tan et al. (2020) note that some domains and classes contain many mislabeled outliers, so they select the 40 most common classes from the 'sketch', 'real', 'clipart' and 'painting' domains. We use the 'sketch' domain as ID, and all other domains ('real', 'clipart', 'painting') as OOD, and in the main paper we report the average accuracies across the OOD domains. In Table 3 we see that the *same trends hold for each of the three OOD domains*. We use a CLIP (Radford et al., 2021) pretrained ResNet-50 model, and fine-tune for 50 epochs (since this is a smaller dataset).

- **Living-17** and **Entity-30**: We use a publicly available MoCo-v2 ResNet-50 checkpoint pretrained on unlabeled examples from ImageNet-1k (Russakovsky et al., 2015), and fine-tune for 20 epochs. Note that Living-17 and Entity-30 are subpopulation shifts derived from ImageNet, but the pretraining is done on unlabeled data and does not see any OOD labels, following the pretraining and fine-tuning strategy in Cai et al. (2021). Entity-30 is a relatively large dataset that contains around 140K training examples.

- **FMoW Geo-shift**: We adapt the version of the dataset from (Koh et al., 2021). We use training data from 'North America' to fine-tune or linear probe, and then evaluate on validation data from Africa and Europe. We use a MoCo-TP (Ayush et al., 2020) checkpoint, pretrained on unlabeled FMoW satellite images. We fine-tune for 50 epochs here since the ID training dataset is smaller (around 20K examples).

- **CIFAR-10 → CIFAR-10.1** (Recht et al., 2018): We follow the same protocols as CIFAR-10 → STL, except we test on CIFAR-10.1.

- **ImageNet**: we linear probe or fine-tune on ImageNet (Russakovsky et al., 2015), and evaluate on **ImageNetV2** (Recht et al., 2019), **ImageNet-R** (Hendrycks et al., 2020), **ImageNet-A** (Hendrycks et al., 2019b), and **ImageNet-Sketch** (Wang et al., 2019). We use a CLIP pretrained ViT-B/16 (vision transformer), the largest publicly available CLIP model (Radford et al., 2021). We ran fine-tuning for 10 epochs, linear probing for 10 epochs. To equalize the runtime for LP-FT, we ran the linear probing stage for 5 epochs, and then the fine-tuning stage for 5 epochs. We used a batch size of 128 for all methods.

**Tuning for ImageNet experiments.** We swept over three learning rates for fine-tuning (0.0001, 0.0003, 0.001) and linear probing (0.01, 0.03, 0.1)—as is standard we use larger learning rates for linear probing. For LP-FT, we swept over 3 learning rates (0.01, 0.03, 0.1) for the 5-epoch linear probing step. We took the run that had the best ImageNet (ID) validation accuracy, and then swept over 3 learning rates (0.00001, 0.00003, 0.0001) for the 5-epoch fine-tuning step—we use a lower learning rate for LP-FT since the experiments on the other datasets suggested that the optimal learning rate that maximizes *ID validation accuracy* for LP-FT is smaller. We did not find the comparisons to be particularly sensitive to learning rate choice.

**Augmentations for ImageNet experiments.** We used augmentations for fine-tuning, and no augmentations for linear probing, following Kornblith et al. (2019). This might raise a question of whether linear probing and LP-FT do better OOD because of the lack of augmentations. So as an ablation we also tried fine-tuning without augmentations, however that led to worse accuracy (than fine-tuning with augmentations) both ID and OOD. We now give details on the preprocessing and augmentations that we used. On ImageNet, for linear probing and LP-FT, we used no augmentations—we just resized each image so that the smaller side has size 224 with bicubic interpolation, and then center-crop to a 224-by-224 image. For fine-tuning, we used augmentations:

Table 4: **OOD accuracies** with 90% confidence intervals over 3 runs, when fine-tuning gets to choose learning rate and early stop, and linear probing gets to choose $\ell_2$ regularization weights, on OOD data. We see that linear probing still typically does better OOD (the only flip from before is on FMoW).

|    | CIFAR-10.1 | STL | Ent-30 | Liv-17 | DomNet | FMoW |
|----|-----------|-----|--------|--------|--------|------|
| FT | **92.27 (0.36)** | 85.97 (0.38) | 64.09 (0.19) | 78.63 (0.53) | 59.43 (2.49) | **40.23 (3.12)** |
| LP | 82.67 (0.22) | **86.53 (0.01)** | **69.15 (0.13)** | **82.39 (0.14)** | **79.91 (0.24)** | 37.12 (0.01) |

|    | ImNetV2 | ImNet-R | ImNet-Sk | ImNet-A | Average |
|----|---------|---------|----------|---------|---------|
| FT | **71.5 (-)** | 52.4 (-) | 40.5 (-) | 27.8 (-) | 61.3 |
| LP | 69.7 (-) | **70.9 (-)** | **46.4 (-)** | **46.1 (-)** | 67.1 |

Table 5: **In-distribution (ID)**: Average distance that features move before and after fine-tuning or LP-FT, multiplied by 100 to make things easier to read. For linear probing the numbers are all 0, since the features are not tuned. As predicted by our theory, we see that features for ID examples (this table) move more than features for OOD examples (Table 6). Both sets of features change substantially less for LP-FT. As usual we show 90% confidence intervals over three runs.

|       | CIFAR-10 | Entity-30 | Living-17 | DomainNet | FMoW |
|-------|----------|-----------|-----------|-----------|------|
| FT    | 2.23 (0.03) | 3.05 (0.02) | 1.88 (0.01) | 207.6 (12.31) | 4.87 (0.15) |
| LP-FT | 0.07 (0.00) | 0.03 (0.01) | 0.11 (0.01) | 0.19 (0.03) | 0.57 (0.19) |

specifically we use RandomResizedCrop in TorchVision, with the default arguments and setting the size of the crop to 224, and then apply a random horizontal flip.

**Notes on pretrained model choice.** We note that our results say that the pretraining has to be good (e.g., at least get reasonable accuracy ID) for linear probing to outperform fine-tuning OOD. So, for example, we use a model pretrained on unlabeled satellite images for the satellite image dataset—if we pretrain the model on ImageNet, we expect that fine-tuning might do better. Similarly, for DomainNet we use a CLIP pretrained model, which is pretrained on the very large WebImageText dataset, and sees a variety of photo and sketch like images. Pretraining on ImageNet alone does not lead to high accuracies on DomainNet (features are not very good), so we do not necessarily expect linear probing to outperform fine-tuning with these lower quality features (for example, see the MoCo ablation in our main paper where we used a worse pretrained model, and fine-tuning did better OOD).

**Sanity check of fine-tuning implementation.** As a sanity check of our implementation, fine-tuning did substantially better than training from scratch on all datasets (both ID and OOD) and matched existing fine-tuning numbers where available (e.g. ResNet50 on CIFAR-10 (Chen et al., 2020b) and Entity-30 (Cai et al., 2021)). Fine-tuning and linear probing also both do substantially better than training from scratch, ID and OOD, across the datasets. For example, on Living-17, training from scratch gets 89.3% ID and 58.2% OOD (Santurkar et al., 2020) which is over 5% worse ID and nearly 20% worse OOD, than all the adaptation methods. For reference linear probing gets 96.5% ID and 82.2% OOD, and fine-tuning gets 97.1% ID and 77.8% OOD. This is even though training from scratch was run for 300 epochs, which is 15 times longer than fine-tuning and LP-FT.

### B.3 TARGET EARLY STOPPING

In the main paper, one ablation we mention is early stopping each fine-tuning method and choose the best learning rate based on target validation accuracy. As expected, fine-tuning does improve a little, but linear probing (average accuracy: 67.1%) is still better than fine-tuning (average accuracy: 61.3%). Table 4 shows the full results for all datasets.

### B.4 FEATURE CHANGE

We examine how much the features changed for ID and OOD examples in each dataset. Specifically, for each dataset, for each input example in the held out validation set, we computed the Euclidean distance of the ResNet-50 features before and after fine-tuning. We averaged these numbers across the dataset, showing the results for ID validation examples in Table 5, and for OOD examples in Table 6.

Table 6: **Out-of-distribution (OOD)**: Average distance that features move before and after fine-tuning or LP-FT, multiplied by 100 to make things easier to read. For linear probing the numbers are all 0, since the features are not tuned. As predicted by our theory, we see that features for ID examples (Table 5) move more than features for OOD examples (this table). Both sets of features change substantially less for LP-FT. As usual we show 90% confidence intervals over three runs.

|  | STL | Entity-30 | Living-17 | DomainNet | FMoW |
|---|---|---|---|---|---|
| FT | 1.70 (0.04) | 2.60 (0.02) | 1.67 (0.01) | 159.97 (16.23) | 5.62 (0.30) |
| LP-FT | 0.04 (0.00) | 0.02 (0.00) | 0.09 (0.01) | 0.18 (0.02) | 0.54 (0.17) |

Table 7: ID and OOD accuracies on Living-17 using a CLIP ResNet-50 model pretrained on the WebImageText dataset, instead of unlabeled ImageNet examples. Similar findings hold—here fine-tuning does similarly to linear probing ID, but does worse than linear probing OOD. LP-FT does better than both ID, and closes 86% of the gap OOD. As usual we show 90% confidence intervals over three runs.

|  | ID | OOD |
|---|---|---|
| LP | 94.7 (0.2) | **78.6 (0.5)** |
| FT | 94.7 (0.1) | 67.3 (0.8) |
| LP-FT | **95.6 (0.2)** | 77.0 (0.6) |

The feature distortion theory predicts that the features for ID examples change more than for OOD examples. This bears out in 9 out of 10 cases, that is all cases except for FT on FMoW. To see this, compare each cell in Table 5 with the corresponding cell in Table 6—the former is higher in 9 out of 10 cases.

The feature distortion theory says that this large feature change is caused because the head is randomly initialized—since the head needs to be updated by a large amount, the feature extractor is also updated a lot because the updates are coupled. Our theory predicts that if the head is initialized via linear probing then the feature extractor should change a lot less for both ID and OOD examples. As predicted by the theory, across all the datasets in Table 5 and Table 6, the features change a lot less for LP-FT than for FT. For example, on CIFAR-10, the features change $30\times$ less for LP-FT than for FT.

These results suggest that fine-tuning underperforms OOD, and LP-FT does well ID and OOD, for the reasons predicted by the feature distortion theory.

### B.5 ADDITIONAL ARCHITECTURES, FINE-TUNING METHODS

The main contributions of our paper are conceptual understanding and theory. However, to strengthen the empirical investigation we ran two additional models (a CLIP vision transformer and CLIP ResNet-50), as well as three additional fine-tuning heuristics. We focus on the Living-17 dataset because some of these ablations require lots of compute and can take a long time to run on all the datasets.

**Architectures and pretraining source**: In the main paper, we showed results when initializing with a MoCo-v2 ResNet-50 model pretrained on unlabeled ImageNet examples. Here we examine how the results change when we 1. Use a ResNet-50 model pretrained on CLIP's WebImageText dataset (Table 7), and, 2. Use a much larger vision transformer model (ViT-B/16) pretrained on CLIP's WebImageText dataset (Table 8)—this is the largest publicly available CLIP model at the time of writing. We see that similar findings to our main paper hold—fine-tuning does better than linear probing ID, but does worse than linear probing ('underperforms') OOD. Finally, LP-FT does better than both methods ID, and closes most (75%-90%) of the gap OOD.

These results are from early stopping on ID validation data. If we early stop on OOD validation data, LP-FT achieves $87.9\pm0.4\%$ OOD accuracy, and LP gets $88.3\pm0.2\%$ OOD accuracy and here there is no statistically significant difference between the two. On the other hand, even if we early stop on OOD validation data, fine-tuning gets $84.4\pm0.5\%$ OOD accuracy which is lower.

**Fine-tuning heuristics**: Transfer learning (initializing with a pretrained model, and then adapting it to a downstream task) is the standard way to build modern ML models, because it improves accuracy

Table 8: ID and OOD accuracies on Living-17 using a CLIP ViT-B/16 (Vision Transformer) model pretrained on the WebImageText dataset, instead of unlabeled ImageNet examples. This is the largest publicly available CLIP model that we could find. The same findings hold—fine-tuning does better than linear probing ID, but does worse than linear probing OOD. LP-FT does better than both ID, and closes 75% of the gap OOD. As usual we show 90% confidence intervals over three runs.

|       | ID              | OOD             |
|-------|-----------------|-----------------|
| LP    | 97.5 (0.1)      | **87.6 (0.5)**  |
| FT    | 97.8 (0.0) | 81.5 (2.1)      |
| LP-FT | **98.0 (0.0)**  | 86.1 (0.1) |

and speeds up training. Since this paradigm is so widely used, there are many heuristics people use when training their models (as mentioned in the main paper, LP-FT has sometimes been used as a heuristic as well, although not in the context of OOD). We showed that LP-FT is one way to do well ID and OOD, but we hope that our theory leads to even better fine-tuning algorithms.

In this section, we compare LP-FT with additional fine-tuning heuristics: using a larger learning rate for the head layer, regularizing the features towards their original values, and side-tuning (Zhang et al., 2020) where we freeze the features but add a side-network.

The intuitions from our theory suggest two other potential ways to improve OOD accuracy: 1. We could use a higher learning rate on the linear layer, so that the linear layer learns quicker and the features do not get as distorted, and 2. We could regularize the weights of the feature extractor towards the pretrained initialization, to prevent feature distortion. These heuristics have been used in prior work on fine-tuning as well, for example method 2 corresponds to L2-SP in (Li et al., 2018).

We run these two approaches on Living-17. For approach (1), we use a $10\times$ higher learning rate for the linear layer, and for approach (2) we regularize the Euclidean distance between the current feature extractor weights (so ignoring the linear head) from the pretrained weights, multiplying by a hyperparameter $\lambda$. We grid search over the same learning rates as fine-tuning for both methods, and in addition for (2) we grid search over $\lambda \in \{1.0, 0.1, 0.01, 0.001, 0.0001\}$, so this amounts to sweeping over 30 hyperparameters as opposed to just 6 for fine-tuning and LP-FT. For each hyperparameter configuration we run 3 replication runs with different seeds to reduce the estimation variance, and early stop and model select using ID data just like for fine-tuning and LP-FT. Just like for fine-tuning and LP-FT, we use a cosine learning rate decay and train for the same number of epochs. Indeed, we find that both (1) and (2) are able to close part of the OOD gap between fine-tuning and linear probing. However, LP-FT does better than both methods ID and OOD. The full results are in Table 9.

We also compare with another method, (3) side-tuning (Zhang et al., 2020). Side-tuning freezes the pretrained features $g(x)$ but trains another 'side' model $s(x)$, and then outputs $v^\top(g(x) + h(x))$, where the head $v$ and the parameters of the side model $s$ are tuned. The intuition for trying this is that side-tuning also preserves the pretrained features which likely reduces feature distortion. In the supplementary of Zhang et al. (2020) they use a ResNet-50 for both the original model and the side model in their vision experiments, so we do the same. We sweep over twelve learning rates $(3 \cdot 10^{-5}, 1 \cdot 10^{-4}, 3 \cdot 10^{-4}, ..., 1.0, 3.0, 10.0)$, with three replication runs with different seeds for each learning rate. Just like for fine-tuning and LP-FT, we use a cosine learning rate decay and train for the same number of epochs, and we early stop and model select using ID validation data. We checked that the best learning rate was not at the boundary of the grid search. On OOD, side-tuning (81.0%) improves over fine-tuning (77.7%). However, side-tuning doesn't do as well ID. LP-FT did better ID and OOD. This could be because side-tuning does not get to refine the pretrained features for the ID task—while the side-network is powerful enough to learn good features, it is initialized randomly and effectively trained from scratch, so it might not be able to learn these good features on the limited sized training dataset (around 40K examples). The results are also in Table 9.

We also include results for training from scratch in Table 9—these results are from Santurkar et al. (2020). Note that training from scratch was done for 450 epochs, whereas fine-tuning was done for 20 epochs. As a sanity check, all the fine-tuning methods and linear probing do substantially better than training from scratch, both ID and OOD.

Table 9: ID and OOD accuracies on Living-17 including three additional fine-tuning heuristics, where we (1) Use a $10\times$ larger learning rate for the head, or (2) Regularize the Euclidean distance of the feature extractor weights to the pretrained initialization, and (3) side-tuning where we freeze the pretrained model but add a side network that is fine-tuned. As a sanity check, all methods do better than training from scratch ID and OOD, and we show 90% confidence intervals over three runs. As per the intuitions from the feature distortion theory, these methods do mitigate feature distortion to some extent and improve OOD accuracy over fine-tuning. LP-FT does better than all methods ID and OOD—nonetheless, we believe that LP-FT is just the first step and hope that our theory can be used to inspire or derive better algorithms.

|  | ID | OOD |
| --- | --- | --- |
| Scratch | 92.4 (1.3) | 58.2 (2.4) |
| LP | 96.5 (0.1) | 82.2 (0.2) |
| FT | 97.1 (0.1) | 77.7 (0.7) |
| FT (10x Linear) | 97.2 (0.2) | 80.4 (0.3) |
| FT (regularized) | 97.1 (0.2) | 80.0 (0.4) |
| Side-tuning | 95.5 (0.4) | 81.0 (0.7) |
| LP-FT | **97.8 (0.1)** | **82.6 (0.3)** |

### B.6 DISCUSSION OF EFFECTIVE ROBUSTNESS

LP-FT gets higher OOD accuracy than fine-tuning, but it sometimes gets higher ID accuracy as well. Taori et al. (2020) and Miller et al. (2021) show that OOD accuracy can often be correlated with ID accuracy, and suggest examining the effective robustness: intuitively the extra gain in OOD accuracy than can be predicted from improved ID accuracy alone. Is LP-FT simply better in-distribution, or does it have higher effective robustness as well?

We start out by noting that linear probing clearly has higher effective robustness in most of our datasets. Linear probing does worse than fine-tuning ID so based on the effective robustness framework we would expect it to do worse than fine-tuning OOD as well. However, linear probing does better than fine-tuning OOD and therefore has higher effective robustness.

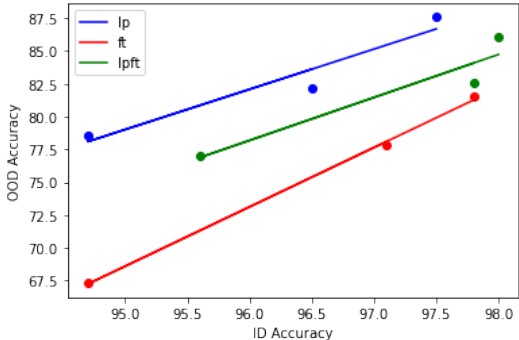

Figure 3: We plot the OOD accuracy against ID accuracy on Living-17 for the three methods we consider, when we start from three different pretrained models (CLIP ResNet-50, CLIP ViT-B/16, MoCo-V2 ResNet-50). The line for linear probing and LP-FT lie above fine-tuning which suggests that they have higher effective robustness. Each point is produced by averaging over three random seeds.

The solutions found by LP-FT also appear to have higher effective robustness than fine-tuning, because when they have similar ID accuracy, LP-FT does much better OOD. For a few pieces of evidence:

1. On CIFAR-10 $\rightarrow$ STL, there is no statistically significant difference between FT and LP-FT on ID, but LP-FT gets 8% higher accuracy OOD in Table 2.

2. If we look at checkpoints earlier in training for CIFAR-10 $\rightarrow$ STL we can exactly equalize ID accuracy and compare OOD accuracies. In-distribution, LP-FT and FT both get 97.2% accuracy, but OOD, LP-FT (90.2%) is much better than FT (81.8%).

3. Finally, in Figure 3 we plot the OOD accuracy against the ID accuracy for fine-tuning and LP-FT on Living-17. We plot these for three different pretrained models (CLIP ResNet-50, CLIP ViT-B/16, MoCo-V2 ResNet-50). We see that the ID-OOD line for LP-FT is above the line for FT indicating effective robustness.

Note that higher effective robustness does not mean a method is better. For example, a method A can have higher effective robustness B by doing a lot worse in-distribution even when they have the same OOD accuracy. In this case, A is clearly inferior since it does worse ID and same OOD, but has higher effective robustness because of its worse ID accuracy.

We believe the finding that linear probing and LP-FT has higher effective robustness than fine-tuning when the distributon shift is large is particularly interesting because Taori et al. (2020) and Miller et al. (2021) show that it is uncommon for methods to have higher effective robustness. In our case linear probing and LP-FT appear to consistently have higher effective robustness which suggests that with good transfer learning methods we can get both high in-distribution accuracy and higher effective robustness.

## C  ADDITIONAL RELATED WORK

**Theoretical analysis of overparameterized models.**    Modern deep learning presents an interesting paradigm for theoretical analysis where the number of parameters is much larger than the number of training points. The model class is highly expressive and several solutions obtain zero training loss even in the presence of noise. Such overparameterized models have received a lot of interest recently especially with a focus on understanding "benign overfitting" or the phenomenon where fitting noisy training data to zero loss leads to classifiers that generalize well. By analyzing different linear overparameterized settings Belkin et al. (2019); Hastie et al. (2019); Bartlett et al. (2019); Muthukumar et al. (2020); Mei & Montanari (2019); Bibas et al. (2019) study various statistical properties such as the "double descent curve" in addition to benign overfitting. One important aspect of overparameterized models is that there is no unique minimizer of the training loss. We need some *inductive bias* which is typically implicit via the optimization procedure. Prior works study the statistical properties of the explicit inductive bias of minimum norm interpolation. In contrast, we study the effect of gradient based optimization from a particular pretrained initialization where we effectively capture the exact implicit inductive bias of gradient based fine tuning.

