# OpenReview forum: "Fine-Tuning can Distort Pretrained Features and Underperform Out-of-Distribution"
_ICLR.cc/2022/Conference — ICLR 2022 Oral_

### Official Review · Reviewer_VgHa · 2021-10-18

**Correctness:** 3
**Technical Novelty And Significance:** 3
**Empirical Novelty And Significance:** 3
**Recommendation:** 8
**Confidence:** 3

**Main Review:**

This paper has many strengths:
- It is not well known that FT often outperforms LP OOD -- the empirical and theoretical study of this phenomenon may be very important to the community.
- LP-FT outperforms LP and FT on many distribution shifts.

In addition, the paper has some weaknesses with associated actionable items:

1) The main weakness of this paper was contextualizing LP-FT. In the abstract and introduction, it seems as though LP-FT is a method that is being introduced by this paper. For instance, the abstract states "our analysis suggests the simple two step-strategy of linear probing then full fine-tuning". It is not until the final page of the main paper that the related work section states "LP-FT has sometimes been used as a fine-tuning heuristic". Moreover, this is presented without any citations. I would very much appreciate clarity on this issue, is LP-FT something that has been used in the past? If so, by which papers and why?  Does this reference that in fine-tuning, sometimes the hyperparameters in the final layer are decoupled from those used for the encoder [1] and performance improves? As shown by [2, 3], performance improvement ID lead to performance improvement OOD, and so this could be an alternative explanation for  the OOD boost of LP-FT that is orthogonal to the authors theory. For instance, in the effective robustness framework of [2], do the solutions found by LP-FT exhibit more effective robustness than the FT solutions?

2) Section 4 is hard to follow in its current form as it is not clear exactly from where numbers are derived. For instance in the first line of the  "Results." paragraph of 4.2, where is the 76.5 number from?

3) One of the networks studied by this paper is CLIP, for which a zero-shot final layer can be constructed (e.g., the final linear layer does not need to be constructed from scratch before fine-tuning). One possible addition to the paper could be discussion of this setting, and in particular if LP-FT is still required.

4) There are a few claims in the paper which could benefit from additional support. In particular, in the conclusion it is stated that the "gap between FT and LP grows as the quality of pretrained features improve". Are the authors referencing the single MOCO-v1 vs. MOCO-v2 experiment or is there additional support for this claim? Are the authors referring to absolute or relative difference between FT and LP? I am wondering as this claim seems a bit counterintuitive, if the features are already good shouldn't LP be sufficient?

5) The empirical verification of the theory (Section 4.3) is very interesting and could be more thorough. In particular, are there associated error bars for the 0.019 and 0.017 numbers as this seems quite close to conclude that one is larger. This euclidean distance experiment is very interesting, why is it only conducted for one distribution shift? Moreover, what happens when analyzing cosine distance instead of cosine distance?

[1] https://arxiv.org/pdf/2106.04560.pdf
[2] https://arxiv.org/abs/2007.00644
[3] https://arxiv.org/abs/2107.04649

**Summary Of The Paper:**

This paper contrasts fine-tuning (i.e., modifying all network weights) and linear probing based on their relative ID/OOD performance. It is known that fine-tuning (FT) outperforms linear probing (LP) ID. This paper presents that the reverse is true OOD (FT outperforms LP). This paper suggests that this occurs because fine-tuning distorts features in conjunction with the final linear layer. Instead, if a final linear layer is trained first, the features do not have to move that much during full fine-tuning. The authors refer to this method as LP-FT and show it often outperforms LP and FT both OOD and ID.

**Summary Of The Review:**

This paper is very interesting and will be an important contribution if concerns are properly addressed. An essential concern is the contextualization of the method LP-FT --- is this method (or a modification) something that people have explored previously and if so in what context. In the current framing of the abstract and introduction, LP-FT seems to be introduced by this paper. In addition, this papers empirical verification of the theory (Sec 4.3) appears very promising but could benefit from additional detail and experiments (e.g., more than one distribution shift in the euclidean distance experiments).

Edit: authors have addressed many concerns and I have changed my score to 6.

---

> ### Author Response · Authors · 2021-11-19
> **(2/2) Other clarifications**
>
> > The reviewer asked if we have "additional support for this claim" that the "gap between FT and LP grows as the quality of pretrained features improve" and asked "if the features are already good shouldn't LP be sufficient?"
>
> We meant to say that the **OOD gap (important clarification!) between FT and LP grows as the quality of the pretrained features improve**. This comes from **Theorem 3.2**: as the features improve the ratio of the OOD error of LP to FT goes to 0, so LP does much better OOD. **As you said, the ID gap between LP and FT will narrow when the features are good but the OOD gap will increase**, so LP is sufficient if we have good features. We have clarified this in the updated manuscript.
>
> > "Results." paragraph of 4.2, where is the 76.5 number from
>
> We apologize for this important typo, we copied from the table to the body of the text incorrectly. The number should be 74.4% as in Table 2, bottom row, right-most column. We did another pass over Section 4 to check for typos---please let us know if you have other concerns.
>
> > CLIP, for which a zero-shot final layer can be constructed… LP-FT is still required?
>
> We like this suggestion - CLIP zero-shot models are another example where we have tradeoffs (zero-shot does worse than fine-tuning ID, but often better OOD), so we could initialize with the zero-shot final layer and then fine-tune the network. We have added this to the discussion.
>
>
> References:
>
> [4] Learning Transferable Visual Models From Natural Language Supervision. A. Radford et al. ICML 2021.
>
> [5] Appendix B.5 of the SimCLR paper: A Simple Framework for Contrastive Learning of Visual Representations. T. Chen et al. ICML 2020.
>
> [6] Page 5, paragraph 1, of SimCLR-v2: Big Self-Supervised Models are Strong Semi-Supervised Learners. T. Chen et al. NeurIPS 2020.
>
> [7] Appendix A.6 and A.7 of MoCo: Momentum Contrast for Unsupervised Visual Representation Learning. K. He et al. CVPR 2020.
>
> [8] Appendix D.3 of Bootstrap your own latent: A new approach to self-supervised Learning. JB Grill et al. NeurIPS 2020.
>
> [9] In-N-Out: Pre-Training and Self-Training using Auxiliary Information for Out-of-Distribution Robustness. S. M. Xie et al. ICLR 2021.
>
> [10] Using Pre-Training Can Improve Model Robustness and Uncertainty. D. Hendrycks et al. ICML 2019.
>
> [11] Using Self-Supervised Learning Can Improve Model Robustness and Uncertainty. D. Hendrycks et al. NeurIPS 2019.
>
> [12] Wilds: A Benchmark of in-the-Wild Distribution Shifts. PW Koh et al. ICML 2021.
>
> [13] The Evolution of Out-of-Distribution Robustness Throughout Fine-Tuning. A. Andreassen et al. Arxiv 2021.
>
> [14] Accuracy on the Line: On the Strong Correlation Between Out-of-Distribution and In-Distribution Generalization. J. Miller et al. ICML 2021.
>
> [15] https://github.com/fastai/course-v3/blob/master/nbs/dl1/lesson1-pets.ipynb (the tutorial first linear probes, and then fine-tune, although it does not say why they do this or compare with just fine-tuning which is why we said some people have used it as a heuristic).
>
> [16] Partial transfusion: on the expressive influence of trainable batch norm parameters for transfer learning. F. Kanavati et al. Medical Imaging with Deep Learning 2021. (their method FC-then-full first linear probes, and then fine-tunes, although it does not say why they do this or compare with just fine-tuning which is why we said some people have used it as a heuristic, in addition they do not look at OOD).

---

> > ### Comment · Reviewer_VgHa · 2021-11-19
> > **Thank you for your response & minor additional point**
> >
> > Thank you for clarifying many of my concerns, in particular the euclidean distance tables in addition to clarifying the typos. I will increase my score accordingly.
> >
> > One minor point with regard to your response to "CLIP, for which a zero-shot final layer can be constructed… LP-FT is still required?"
> >
> > > We like this suggestion - CLIP zero-shot models are another example where we have tradeoffs (zero-shot does worse than fine-tuning ID, but often better OOD), so we could initialize with the zero-shot final layer and then fine-tune the network. We have added this to the discussion.
> >
> > To better contextualize things in your added discussion, I believe this is what is done in a paper you've cited https://arxiv.org/abs/2109.01903.

---

> > > ### Author Response · Authors · 2021-11-19
> > > **Thanks for the quick response - anything else we can do to improve the work?**
> > >
> > > Thank you for the quick update! Sorry to follow up, but we just wanted to check what improvements could turn this into an "accept" recommendation? We would love to get feedback to improve our work going forward!
> > >
> > > So far it looked like the review found the paper "very interesting" and an "important contribution" but the "main weakness" was "contextualizing LP-FT" - can we do better on this front? The other remaining point seemed to be about "empirical verification" of the the theory. Again, really appreciate you taking the time to write a thoughtful review so we apologize for bothering you about this.
> > >
> > > To show that our findings are more general we now compare linear probing and fine tuning on ImageNet (ID) and four additional OOD datasets suggested by another reviewer, in Appendix B.5. We believe this is a strong support for our claims since these were datasets selected by another reviewer, and a larger architecture (vision transformer) but the observation that linear probing does better than fine-tuning for large distribution shifts still holds.
> > >
> > > > To better contextualize things in your added discussion, I believe this is what is done in a paper you've cited https://arxiv.org/abs/2109.01903.
> > >
> > > Thank you for bringing this up, and we have edited the discussion. Minor note: our understanding is that this paper (concurrent work, also submitted to ICLR) takes a weighted sum of the fine-tuned and zero-shot model in weight space, as opposed to LP-FT, but we will be happy to edit our discussion if we misunderstood.

---

> > > > ### Comment · Reviewer_VgHa · 2021-11-19
> > > > **Thanks -- will look at other discussions**
> > > >
> > > > I haven't thoroughly read through the discussions with the other reviewers yet, at that point if no issues arise I will change to accept recommendation. Thanks for the paper and the thorough response.
> > > >
> > > > With regards to the concurrent work I have the same understanding, what I meant is that for the fine-tuning part they do what you've suggested in your original response above:
> > > > > so we could initialize with the zero-shot final layer and then fine-tune the network

---

> > > > > ### Author Response · Authors · 2021-11-20
> > > > > **Thank you -- makes sense**
> > > > >
> > > > > That makes sense, and sounds fair! Sorry again to have bothered you about this.
> > > > >
> > > > > We agree with your latest clarification, and have edited the discussion again based on that - thank you.

---

> > > > > > ### Comment · Reviewer_VgHa · 2021-11-20
> > > > > > **Accept**
> > > > > >
> > > > > > I've read through the responses to other reviewers and changed my rating to accept.

---

> ### Author Response · Authors · 2021-11-19
> **(1/2) Main concerns: contextualizing LP-FT and more datasets for empirical verification of theory**
>
> > "The main weakness of this paper was contextualizing LP-FT"
>
> > "is LP-FT something that has been used in the past?"
>
> We apologize for the poor and confusing contextualization of LP-FT.
>
> Fine-tuning is widely studied so there are many heuristics but they are poorly understood. **Our contribution is to theoretically understand** these better. This is particularly important for robustness/OOD because we find that the de-facto standard method of full fine-tuning is sub-optimal, but widely used in the OOD/robustness literature.
>
> More specifically for LP-FT: Some people have used LP-FT as a heuristic for regular in-distribution fine-tuning [15,16]. However there isn't an understanding of why or when it does better, and LP-FT has not been used for OOD / robustness [9,10,11,12,13,14] or in popular recent pretraining papers like CLIP [4], SimCLR [5,6], MoCo [7], BYOL [8].
>
> We have edited this in the manuscript and apologize for miscommunications. We've **clarified in the intro that the method has sometimes been used as a heuristic (although not for OOD), with some cites**. Please let us know what else we can do to contextualize better.
>
> > "performance improvement ID lead to performance improvement OOD"
>
> > "so this could be an alternative explanation for the OOD boost of LP-FT that is orthogonal to the authors theory"
>
> > "do the solutions found by LP-FT exhibit more effective robustness than the FT solutions?"
>
> This is a great question. The **solutions found by LP-FT have higher effective robustness than FT**, because when they have similar ID accuracy, LP-FT does much better OOD. We have added Appendix B.6 to discuss this. Details:
> - On CIFAR-10 $\to$ STL, there is no statistically significant difference between FT and LP-FT on ID, but LP-FT gets 8% higher accuracy OOD (Table 1).
> - If we look at checkpoints earlier in training for CIFAR-10 $\to$ STL we can exactly equalize ID accuracy and compare OOD accuracies. For ID: LP-FT and FT both get 97.2%. For OOD: LP-FT (90.2%) > FT (81.8%).
> - In Appendix B.6 we plot the OOD accuracy against the ID accuracy for fine-tuning and LP-FT on Living-17. We plot these for three different pretrained models. In these new experiments added for the rebuttal, we see that the ID-OOD line for LP-FT is above the line for FT indicating effective robustness (as in [2]).
>
> We thank the reviewer for the suggestion - we believe the finding that **LP-FT has higher effective robustness than FT makes our paper stronger**. This is particularly interesting because [2] and [3] show that it is uncommon for methods to have higher effective robustness.
>
> > The reviewer says the "empirical verification of the theory (Section 4.3) is very interesting" and asked if there are "associated error bars?" and said the "euclidean distance experiment is very interesting why is it only conducted for one distribution shift?"
>
> We agree with this point and for the rebuttal we have **added error bars and computed the distance that ID and OOD features move across the five datasets for both FT and LP-FT**. We have added these to Table 5 and Table 6 in Appendix B.3. We use Euclidean distance because in our feature distortion theory the norm of the change, and not just the direction, matters.
>
> We see that in 9/10 cases (all except FT for FMoW), the **OOD features change less than the ID features** as predicted by the theory, and this is statistically significant. For example, on CIFAR->STL, fine-tuning, the distance moved is 1.70 \pm 0.04 for OOD features and 2.23 \pm 0.03 for ID features.
>
> Additionally, across all 10 cases the **features change orders of magnitude less for LP-FT** than fine-tuning, suggesting that LP-FT indeed works for the reason predicted by the theory.
>
> Euclidean distance for ID examples (multiplied by 100):
>
> |       | CIFAR-10    | Entity-30   | Living-17   | DomainNet     | FMoW        |
> |-------|-------------|-------------|-------------|---------------|-------------|
> | FT    | 2.23 (0.03) | 3.05 (0.02) | 1.88 (0.01) | 207.6 (12.31) | 4.87 (0.15) |
> | LP-FT | 0.07 (0.00) | 0.03 (0.01) | 0.11 (0.01) | 0.19 (0.03)   | 0.57 (0.19) |
>
> Euclidean distance for OOD examples (multiplied by 100):
>
> |       | STL         | Entity-30   | Living-17   | DomainNet      | FMoW        |
> |-------|-------------|-------------|-------------|----------------|-------------|
> | FT    | 1.70 (0.04) | 2.60 (0.02) | 1.67 (0.01) | 159.97 (16.23) | 5.62 (0.3)  |
> | LP-FT | 0.04 (0.00) | 0.02 (0.00) | 0.09 (0.01) | 0.18 (0.02)    | 0.54 (0.17) |
>
> (Minor note: Yes, the features appear to change much more for DomainNet, which is perhaps why LP and LP-FT are substantially better there)

---

### Official Review · Reviewer_5UET · 2021-10-23

**Correctness:** 3
**Technical Novelty And Significance:** 3
**Empirical Novelty And Significance:** 2
**Recommendation:** 8
**Confidence:** 4

**Main Review:**

**Strengths:**

1. This paper explores two increasingly impactful research directions, fine-tuning pre-trained models and generalization under distribution shifts. I believe their results, both theoretical and empirical, would be of interest to many in the community

2. The theoretical results, despite studying a very simple setting that is unlikely to be used in any real experiments, offer intuitions that transferred well to practical results.

3. The paper is clear and well written.

**Weaknesses:**

1. Experiments could be more comprehensive. For instance, while this paper analyses several distribution shifts (Breeds-Living17, Breeds-Entity30, DomainNet, CIFAR->STL, CIFAR 10.1 and FMoW), it was surprising that distribution shifts like ImageNetV2, ImageNet-R, ImageNet-A, ObjectNet and ImageNet Sketch were not considered by this work. Moreover, only a single architecture, ResNet-50, is used for the experiments. As they stand, it is unclear whether the results from this paper would hold at larger scales.

2. This work overlooks simple baselines. For instance, given the finding that end-to-end fine-tuning significantly changes the weights of the backbone (which arguably hurts OOD performance), it would be natural to consider having a smaller learning rate for the backbone, and a higher one for the untrained final linear layer. It is not uncommon to do so in practice. Moreover, authors could have explored regularizing the difference of the weights of the backbone to the original weights, encouraging them to not be changed too much. Finally, the authors do not mention the fact that models like CLIP allow the possibility of starting with a good set of initial weights for the final layer, as they can be used in a zero-shot setting. If the intuitions presented in this paper hold, this prevent the "distortion" that happens when the last layer is far from the optimum.

3. The theoretical results are disconnected from realistic settings. Some clear examples are the assumption of a two-layer network, a squared error loss (while the de facto standard for classification is cross-entropy), and considering the worst case loss over distributions of bounded norm.

4. Apart from the theoretical results, there is not a lot of novelty introduced by this work. The two-stage fine-tuning strategy where end-to-end fine-tuning follows linear probing is commonplace and thought in introductory courses in deep learning, e.g. in the first lesson of https://github.com/fastai/course-v3/blob/master/nbs/dl1/lesson1-pets.ipynb

5. Comparison to previous work is lacking. In particular, explicit comparisons of LP-FT with other recent robustness-oriented fine-tuning methods would greatly strengthen this work [1-4, among others].


**References:**

[1] Wortsman, Mitchell, et al. "Robust fine-tuning of zero-shot models." arXiv preprint arXiv:2109.01903 (2021).

[2] Aghajanyan, Armen, et al. "Better fine-tuning by reducing representational collapse." arXiv preprint arXiv:2008.03156 (2020).

[3] Jiang, Haoming, et al. "Smart: Robust and efficient fine-tuning for pre-trained natural language models through principled regularized optimization." arXiv preprint arXiv:1911.03437 (2019).

[4] Zhu, Chen, et al. "Freelb: Enhanced adversarial training for natural language understanding." arXiv preprint arXiv:1909.11764 (2019).

**Summary Of The Paper:**

This paper explores how different strategies for fine-tuning affect in- and out-of-distribution performance. The authors contrast linear probing (updating only the parameters of the final linear layer), end-to-end fine-tuning (updating all parameters of the model) and a two-stage approach, where linear probing is followed by end-to-end fine-tuning. While end-to-end fine-tuning typically improves in-distribution performance, the authors show that it can also underperform linear probing out-of-distribution. The paper theoretically analyzes the tradeoffs in a simplified scenario with two-layer networks, finding that end-to-end fine-tuning can "distort" pre-trained features. Their experiments on a number of datasets including CIFAR, WILDS-FMoW and others, confirm the intuitions from their theory. The proposed mitigation strategy, a two-stage fine-tuning approach where end-to-end fine-tuning follows linear probing is found to be beneficial, especially out-of-distribution.


**Update:** The authors addressed most of the concerns raised by this and other reviews, and I am raising my score accordingly.

**Summary Of The Review:**

Overall, while there are several points of concern that could strengthen the paper, I believe the results and theory presented by this work would still be of interest to many in the community, so I recommend its acceptance.

---

> ### Author Response · Authors · 2021-11-19
> **(3/3) Other responses on related work**
>
> > "Apart from the theoretical results, there is not a lot of novelty introduced by this work."
>
> > "two-stage fine-tuning strategy where end-to-end fine-tuning follows linear probing… in the first lesson of https://github.com/fastai/course-v3/blob/master/nbs/dl1/lesson1-pets.ipynb"
>
> In addition to the theoretical contributions, we believe **the finding that linear probing does better than fine-tuning for large distribution shifts is novel.**
>
> We thank the reviewer for this link, and have added a cite to the tutorial---in our original paper, we say "LP-FT has sometimes been used as a fine-tuning heuristic, we show that it addresses the ID-OOD tradeoff theoretically and empirically".
>
> The reviewer asked for better contextualization of LP-FT, and what the novel aspects are:
> - LP-FT is a direct implication of our theory (which says that fine-tuning underperforms OOD because the initial random head is far from the optimal head). The goal of our experiments is to test if the theory's predictions hold up on real datasets.
> - Some people have used LP-FT as a heuristic. However there isn't an understanding of why or when it does better, and **LP-FT has not been used for OOD / robustness** [5,12,13,14,15,16] or popular recent pretraining papers like CLIP [4], SimCLR [5,6], MoCo [7], BYOL [8]---they use linear probing or vanilla fine-tuning.
>
> That said, we appreciate your point so please let us know how we can contextualize things better.
>
> > Other related work
>
> In the related works we now clarify that [1] is a promising approach, and have added [2]-[4]. We note that [1] appears to be concurrent work based on the ICLR guidelines.
>
> [1] mentions on page 8 that they got comparable performance by ensembling the two models in output space. **For the rebuttal, we ran an additional comparison on Living-17**, where we ensemble the outputs of the linear probed and fine-tuned models, choosing a weight $\alpha$ to maximize ID validation accuracy. Averaged over three runs, ID accuracy of this ensemble was 97.1\% and OOD accuracy was 80.8\%, so better than fine-tuning but LP-FT is better both ID (97.8%) and OOD (82.6%). Finally, we note that we do not think LP-FT is the SOTA method, and we hope that combining our insights with other methods could lead to better performance.
>
> > "authors do not mention the fact that models like CLIP allow the possibility of starting with a good set of initial weights for the final layer"
>
> This is a great point, we have added this to the discussion.
>
> References:
>
> [5] In-N-Out: Pre-Training and Self-Training using Auxiliary Information for Out-of-Distribution Robustness. S. M. Xie et al. ICLR 2021.
>
> [6] On the Theory of Transfer Learning: The Importance of Task Diversity. N. Tripuraneni et al. NeurIPS 2020.
>
> [7] Learning Transferable Visual Models From Natural Language Supervision. A. Radford et al. ICML 2021.
>
> [8] Appendix B.5 of the SimCLR paper: A Simple Framework for Contrastive Learning of Visual Representations. T. Chen et al. ICML 2020.
>
> [9] Page 5, paragraph 1, of SimCLR-v2: Big Self-Supervised Models are Strong Semi-Supervised Learners. T. Chen et al. NeurIPS 2020.
>
> [10] Appendix A.6 and A.7 of MoCo: Momentum Contrast for Unsupervised Visual Representation Learning. K. He et al. CVPR 2020.
>
> [11] Appendix D.3 of Bootstrap your own latent: A new approach to self-supervised Learning. JB Grill et al. NeurIPS 2020.
>
> [12] Using Pre-Training Can Improve Model Robustness and Uncertainty. D. Hendrycks et al. ICML 2019.
>
> [13] Using Self-Supervised Learning Can Improve Model Robustness and Uncertainty. D. Hendrycks et al. NeurIPS 2019.
>
> [14] Wilds: A Benchmark of in-the-Wild Distribution Shifts. PW Koh et al. ICML 2021.
>
> [15] The Evolution of Out-of-Distribution Robustness Throughout Fine-Tuning. A. Andreassen et al. Arxiv 2021.
>
> [16] Accuracy on the Line: On the Strong Correlation Between Out-of-Distribution and In-Distribution Generalization. J. Miller et al. ICML 2021.

---

> ### Author Response · Authors · 2021-11-19
> **(2/3) Ran vision transformer, more datasets, more baselines**
>
> > "unclear whether the results from this paper would hold at larger scales."
>
> > "distribution shifts like ImageNetV2, ImageNet-R, ImageNet-A, ObjectNet and ImageNet Sketch were not considered"
>
> > "only a single architecture, ResNet-50, is used"
>
> In our original paper we used popular distribution shift datasets like DomainNet, Breeds, CIFAR $\to$ STL, and the remote sensing dataset "Functional Map of the World". We have now **added results on larger scale datasets**: we linear probe or fine-tune on ImageNet, and evaluate on **ImageNetV2, ImageNet-R, ImageNet-A, and ImageNet-Sketch**. This is for a **CLIP ViT-B/16 (vision transformer), the largest publicly available CLIP model** [7], which is much larger than a ResNet-50.
>
> Recall that our theory says that linear probing does better than fine-tuning when the shift is large. We expect linear probing to do better on ImageNet-R, ImageNet-A, ImageNet-Sketch. We expect fine-tuning to do better on ImageNetV2 because it is collected by "repeating the dataset curation process" of ImageNet (quote from the ImageNetV2 paper)---this is similar to CIFAR-10 $\to$ CIFAR-10.1 in our original paper.
>
> We find that fine-tuning gets 2% higher accuracy ID than linear probing, but averaged across the four OOD datasets fine-tuning gets 10% lower accuracy OOD. LP-FT is currently running and will be included, if accepted, in the camera ready.
>
> For the ImageNet-validation set (ID):
>
> | ID             | ImageNet |
> |----------------|----------|
> | Fine-tuning    |  **81.7**|
> | Linear probing |     79.7 |
>
> For the OOD datasets (ImageNetV2 is a replication of ImageNet so a small shift, so the results are similar to CIFAR-10 -> CIFAR-10.1 in our original paper):
>
> | OOD            |  ImageNetV2 (small shift) | Renditions | Sketch | ImageNet-A |*OOD Average*|
> |----------------|---------------------------|------------|--------|------------|-------------|
> | Fine-tuning    |                   **71.5**|       52.4 |   40.5 |       27.8 |       *48.1*|
> | Linear probing |                      69.7 |    **70.6**|**46.4**|    **45.7**|       *58.1*|
>
> We think this is a strong result because it suggests that **our theory and intuitions generalize to larger scale OOD datasets requested by the reviewer**. We include more details in Appendix B.5.
>
> We have also added **new results for a CLIP vision transformer** (ViT-B/16) on Living-17 to Appendix B.4. Indeed, we find that fine-tuning does better than linear probing ID, but underperforms OOD. LP-FT does better than both ID, and closes over 75% of the gap OOD. These results are from early stopping on ID validation data—-LP-FT actually achieves 87.9% OOD accuracy (better than LP) in the middle of training.
>
> |       | ID         | OOD        |
> |-------|------------|------------|
> | LP    | 97.5 (0.1) | 87.6 (0.5) |
> | FT    | 97.8 (0.0) | 81.5 (2.1) |
> | LP-FT | 98.0 (0.0) | 86.1 (0.1) |
>
> > "overlooks simple baselines"
>
> > "given the finding that end-to-end fine-tuning significantly changes the weights of the backbone (which arguably hurts OOD performance), it would be natural to consider"
>
> > (1) "smaller learning rate for the backbone, and a higher one for the untrained final linear layer"
>
> > (2) "regularizing the difference of the weights of the backbone to the original weights"
>
> As the reviewer mentions, intuitions from the feature distortion theory also suggest that some other methods like (1) and (2) reduce feature distortion and can improve OOD accuracy to some extent. This is the right insight, and LP-FT is an extreme version where the learning rate for the features is 0 initially. **We view (1) and (2) as other methods that can work based on our theory, as opposed to baselines.**
>
> We have **added results for methods (1) and (2) on Living-17** in Appendix B.4. For (1) we used a 10x larger learning rate for the linear classifier layer, repeating the grid search over 6 learning rates, and for (2) we grid searched over regularization weights in {1.0, 0.1, 0.01, 0.001, 0.0001}, on top of the grid search over 6 learning rates we did for fine-tuning and LP-FT (so 30 hyperparameter configurations * 3 replication runs = 90 runs). As expected, we found that these methods do better than fine-tuning OOD. LP-FT still does better ID and OOD.
>
> |                  | ID         | OOD        |
> |------------------|------------|------------|
> | LP               | 96.5 (0.1) | 82.2 (0.2) |
> | FT               | 97.1 (0.1) | 77.7 (0.7) |
> | FT (10x Linear)  | 97.2 (0.2) | 80.4 (0.3) |
> | FT (regularized) | 97.1 (0.2) | 80.0 (0.4) |
> | LP-FT            | 97.8 (0.1) | 82.6 (0.3) |
>
> We used Living-17 because that's what we used for all ablations in Section 4.3.

---

> ### Author Response · Authors · 2021-11-19
> **(1/3) Core contribution is theory of fine-tuning, which is challenging**
>
> We thank Reviewer 5UET for saying that the "results, both theoretical and empirical, would be of interest to many" and for "recommend(ing) its acceptance". We also thank the reviewer for the detailed and useful suggestions---**we have run suggested experiments (more datasets, more architectures, more baselines)**, which we believe makes our paper stronger.
>
> We believe the review focused more on the experiments so would like to clarify the focus of our paper. Many fine-tuning heuristics are used in practice, but it is unclear when they work---our **goal is theory and understanding** of when to use what method. We also believe that **the finding that linear probing does better than fine-tuning for large distribution shifts is novel** and interesting.
>
> > The reviewer likes that the theoretical results "offer intuitions that transferred well to practical results" but mentions that it is "disconnected from realistic settings... two-layer network... squared error loss"
>
> We agree---we would love to study more realistic models but want to emphasize that studying fine-tuning for our setting is already challenging and a big step forward.
> - **Analyzing fine-tuning is challenging even for two-layer models** because we need to analyze how the representations change. Prior transfer learning work [5, 6] recognizes this challenge and instead studies the simpler case where the representations are fixed.
> - We think it is a strength that our theory leads to insights that hold up on many real datasets, for example that ID features change more than OOD features, features change much less for LP-FT, and OOD early stopping does not solve the problem with fine-tuning (Section 4.3).
>
> We believe that theory is our strongest contribution and novelty so we hope the reviewer takes these responses into account.

---

> ### Author Response · Authors · 2021-11-20
> **Any other questions or suggestions?**
>
> Thank you again for the comprehensive and useful review, and sorry to bother you. We just wanted to check if you had any other questions or suggestions - we would love to get feedback to further improve our work.
>
> So far the review "recommend(ed) its acceptance." and there were some questions about larger scale datasets, architectures, and methods. We hope our response has addressed these and clarified the focus and novelty of the paper?

---

> ### Author Response · Authors · 2021-11-27
> **Larger scale datasets and models, LP-FT**
>
> As mentioned in the comment to all reviewers, the LP-FT runs on ImageNet-Renditions, ImageNet-sketch, ImageNet-A, and ImageNet-V2, with a larger scale CLIP vision transformer model, have finished running. **LP-FT did the best on all four OOD datasets and tied best on ID**. Please let us know if there's anything else we can do to further improve our paper. Thank you for your useful suggestions, and for engaging with us to improve our work.

---

### Official Review · Reviewer_VT2b · 2021-11-02

**Correctness:** 4
**Technical Novelty And Significance:** 3
**Empirical Novelty And Significance:** Not applicable
**Recommendation:** 8
**Confidence:** 4

**Main Review:**

Pros:

1. the observation that fine-tuning performance worse on the OOD test is new and interesting

2. the proposed method LP-FT is simple yet effective

3. theoretically analysis is further provided to show why LP-FT works

Cons:

1. some important baselines are missing like [a-c], which should be discussed and compared

[a]. Guo, Yunhui, et al. "Spottune: transfer learning through adaptive fine-tuning." Proceedings of the IEEE/CVF Conference on Computer Vision and Pattern Recognition. 2019.

[b]. Zhang, Jeffrey O., et al. "Side-Tuning: A Baseline for Network Adaptation via Additive Side Networks." Computer Vision–ECCV 2020: 16th European Conference, Glasgow, UK, August 23–28, 2020, Proceedings, Part III 16. Springer International Publishing, 2020.

[c]. Ge, Weifeng, and Yizhou Yu. "Borrowing treasures from the wealthy: Deep transfer learning through selective joint fine-tuning." Proceedings of the IEEE conference on computer vision and pattern recognition. 2017.

2. why different pre-trained models are utilized for different ID and OOD pairs? In fact, the authors could use different models for each ID and OOD pair.

**Summary Of The Paper:**

This paper studies the problem of how to fine-tune a pre-trained model and obtain better results for both ID and OOD. Two methods, fine-tuning and linear probing, are investigated and compared, then a new two-step variant called LP-FT is derived. Results further verify that LP-FT obtains the best performance for ID and OOD tests compared with FT and LP.

**Summary Of The Review:**

This paper discovers that vanilla fine-tuning performs worse than linear probing for the OOD tests and then develops a new method combining these two techniques sequentially. Results on several datasets verify its effectiveness. Even there exists some minor problems, this paper is interesting and easy to read. Thus, I tend to give the "weak accept" score.

-----POST REBUTTAL-----

The authors have addressed my concerns. Thus, I increase my score from 6 to 8.

---

> ### Author Response · Authors · 2021-11-19
> **Clarifying that our focus is theory; added more baselines, pretrained models, datasets**
>
> We thank the reviewer for the positive review and useful suggestions, and for agreeing that "the observation that fine-tuning performance worse on the OOD test is new and interesting" and appreciating the "theoretical analysis".
>
> We believe the review focused more on the experiments so would like to clarify the focus of our paper. Many fine-tuning heuristics are used in practice, but it is unclear when they work---our **goal is theory and understanding** of when to use what method.
> - Fine-tuning theory is novel and challenging because we need to analyze how the representations change. Prior transfer learning work [1,2] recognizes this challenge and instead studies the simpler case where the representations are fixed.
> - LP-FT is a direct implication of our theory (which says that fine-tuning underperforms OOD because the initial random head is far from the optimal head). The goal of our experiments is to test if the theory's predictions hold up on real datasets. Besides LP-FT, we test other predictions from our theory in Section 4.3.
>
> We believe that theory is our strongest contribution and novelty so we hope the reviewer takes these responses into account.
>
> > "some important baselines are missing" and the reviewer would like to see these "discussed and compared"
>
> We thank the reviewer for the useful pointers. We do not think LP-FT is a SOTA method, but an example insight of the theory---as predicted by the theory it gets the best of both worlds: the strong ID accuracy of fine-tuning, and strong OOD accuracy of linear probing. Combining it with ideas like spot-tuning, side-tuning, and joint fine-tuning is a good idea for future work and would likely boost its performance. We have added cites to these works and a discussion to the related works.
>
> The strength of our theory is it can give insights into fine-tuning methods. For example, side-tuning freezes the pretrained weights and tunes a smaller side network---in our theoretical setting this would get good OOD performance as well because it reduces feature distortion.
>
> That said, we have **added more fine-tuning methods for the rebuttal**---we now compare with l2-sp [3] and using a higher learning rate for the final layer, on the Living-17 dataset. LP-FT did better than both methods, ID and OOD. L2-sp requires an additional hyperparameter---we tuned this, doing a grid search over 5 regularization weights * 6 learning rate = 30 hyperparameters, even more so than fine-tuning and LP-FT where we just swept over 6 learning rates. Each configuration is run 3 times with different seeds. More details are in Appendix B.4.
>
> |                  | ID         | OOD        |
> |------------------|------------|------------|
> | LP               | 96.5 (0.1) | 82.2 (0.2) |
> | FT               | 97.1 (0.1) | 77.7 (0.7) |
> | FT (10x Linear)  | 97.2 (0.2) | 80.4 (0.3) |
> | FT (regularized) | 97.1 (0.2) | 80.0 (0.4) |
> | LP-FT            | 97.8 (0.1) | 82.6 (0.3) |
> L2-sp corresponds to FT (regularized). We used Living-17 because that's what we used for all ablations in Section 4.3.
>
> > "why different pre-trained models are utilized for different ID and OOD pairs"
>
> > "In fact, the authors could use different models for each ID and OOD pair."
>
> We used different pretraining methods because we require reasonably good pretrained features for the downstream task. For example, for the satellite dataset FMoW we use a model pretrained on unlabeled satellite images---an ImageNet pretrained model would not be suitable and performs poorly on satellite images. Similarly, a satellite image pretrained model would not be appropriate or do well on CIFAR.
>
> That said, we agree with your broader point and have **added more pretraining models for the rebuttal**---we ran the Living-17 experiments with a CLIP-ResNet-50 and a CLIP-ViT-B/16 (a vision transformer that is much larger than ResNet-50), and the same findings hold. We have added these to Appendix B.4.
>
> In addition, we have **added results for larger datasets**: ImageNet $\to$ ImageNet-v2, ImageNet-R, ImageNet-Sketch, ImageNet-A, in Appendix B.5. Fine-tuning still does better than linear probing ID, but worse OOD when the shift is large.
>
> We believe that these additional experiments make our paper stronger, and show that our findings are more general, so we thank the reviewer for the suggestions.
>
> [1] In-N-Out: Pre-Training and Self-Training using Auxiliary Information for Out-of-Distribution Robustness. S. M. Xie et al. ICLR 2021.
>
> [2] On the Theory of Transfer Learning: The Importance of Task Diversity. N. Tripuraneni et al. NeurIPS 2020.
>
> [3] Explicit Inductive Bias for Transfer Learning with Convolutional Networks. X. Li et al. ICML 2018.

---

### Official Review · Reviewer_Gpdo · 2021-11-03

**Correctness:** 4
**Technical Novelty And Significance:** 2
**Empirical Novelty And Significance:** 3
**Recommendation:** 8
**Confidence:** 3

**Main Review:**

The strength of this paper is the extensive and detailed toy and benchmark experiments. The reasoning and intuition of why fine-tuning underperforms on OOD data are well explained and discussed throughout the paper. The suggested solution by combining linear probing and fine-tuning also has good performance.

**Summary Of The Paper:**

This paper discovers an interesting behavior of model fine-tuning: the performance is worse compared to linear probing on OOD data (i.e., data from other domains), especially when the distribution shift between inner distribution and out of distribution are big. The explanation provided in the paper is that fine-tuning distorts the feature representations, overfits on inner distributions, and thus has a higher error on OOD data. The authors also provide a simple solution to this issue by fine-tuning with a classification head initialized from linear probing and had better results in all the benchmarks they have in the paper.

**Summary Of The Review:**

Although the solution proposed in this paper is not fancy, the reasoning, intuition, and experiments are well written.

---

> ### Author Response · Authors · 2021-11-19
> **Key technical novelty is theory of fine-tuning**
>
> We thank the reviewer for the positive review, expressing that we discover an "interesting behavior of model fine-tuning", the "reasoning and intuition… are well explained", and for appreciating the "extensive… toy and benchmark experiments."
>
> Regarding technical novelty, we wanted to clarify that **a key contribution of our paper is the theoretical analysis**. Recent transfer learning theory works [1,2] say that theoretical analysis of fine-tuning is missing and challenging because we need to analyze how the representations change, so we believe our theory is an important step forward. These prior works study training a linear classifier on frozen features, which has a closed form solution.
>
> We believe that **theoretically analyzing fine-tuning of overparameterized models, and characterizing feature distortion, is our key technical novelty and contribution**.
>
> [1] In-N-Out: Pre-Training and Self-Training using Auxiliary Information for Out-of-Distribution Robustness. S. M. Xie et al. ICLR 2021.
>
> [2] On the Theory of Transfer Learning: The Importance of Task Diversity. N. Tripuraneni et al. NeurIPS 2020.

---

### Author Response · Authors · 2021-11-23
**Overall Response**

We thank the reviewers for their time and thoughtful feedback. The reviewers agree that our work is "very interesting and will be an important contribution" (RVgHa), would "be of interest to many in the community" and "recommend its acceptance" (R5UET), "interesting and easy to read" (RVT2b), and appreciate that our paper "discovers an interesting behavior of model fine-tuning" and has "extensive and detailed toy and benchmark experiments" (RGpdo).

(i) Main contribution: the bulk of the research contribution of this work is to provide insight and theoretical analyses/tools to study fine-tuning. Based on our analysis, one can draw several interesting implications (one being LP-FT which we focus on experimentally).

(ii) Novelty: While we do not propose any new heuristics, our experimental contributions are novel with respect to robustness. We find a surprising result that linear probing often does better than fine-tuning OOD, and LP-FT improves both in-distribution and OOD accuracy. These are existing heuristics, but whose robustness gains haven't been investigated. We also provide insights into why different heuristics behave differently and how to design better fine-tuning methods keeping in mind robustness.

(iii) Connection between theory and practice: we acknowledge that the theoretical settings considered are more stylized than real world settings. However, to the best of our knowledge, there is no existing analysis on precise trajectories of fine-tuning. Transfer learning theory focuses on linear probing---analysis of fine-tuning is scarce and challenging, and we lay out precise technical challenges in page 2 of the Introduction. It is important to ensure that our insights from the theoretical setting aren't misleading and in order to do this, we tested out numerous implications of our theory on real datasets (Section 4.2 and 4.3).

We have incorporated reviewer feedback in our revision, and we believe this makes our paper much stronger. Thank you to the reviewers for these suggestions:

- Added results for linear probing and fine-tuning on four larger scale OOD datasets (ImageNetV2, ImageNet-R, ImageNet-Sketch, ImageNet-A) suggested by R5UET in Appendix B.5. We think this is a strong result because it suggests that our theory and intuitions generalize to larger scale OOD datasets requested by the reviewer.

- Added more architectures (e.g. vision transformers) in Appendix B.4 and B.5.

- Added comparisons to more fine-tuning methods (regularizing towards pretrained initialization, higher learning rate for head layer, side-tuning) in Appendix B.4. LP-FT does better both ID and OOD.

- Added plots of effective robustness in Appendix B.6 - we see that LP-FT has higher effective robustness than fine-tuning, which we think makes our results stronger.

- Tested the theoretical implications of Section 4.3 on more datasets, showing results in Appendix B.3.

---

> ### Author Response · Authors · 2021-11-27
> **LP-FT Runs Finished on ImageNet datasets**
>
> The LP-FT runs on ImageNet-Renditions, ImageNet-sketch, ImageNet-A, and ImageNet-V2 have finished running. **LP-FT did the best on all four OOD datasets and tied best with fine-tuning on ID**---as predicted by the theory it gets the strong OOD accuracy of linear probing and the strong ID accuracy of fine-tuning. We apologize for the delay because it took some time for these larger scale jobs to finish running given our compute resources.
>
> For the OOD datasets (ImageNetV2 is a replication of ImageNet so a small shift, so the results are similar to CIFAR-10 -> CIFAR-10.1 in our original paper):
>
> | OOD            | ImageNetV2 (Small shift) | Renditions | Sketch | ImageNet-A |Average      |
> |----------------|--------------------------|------------|--------|------------|------|
> | Fine-tuning    |**71.5**                  | 52.4       | 40.5   | 27.8       |        48.1 |
> | Linear probing | 69.7                     | 70.6       | 46.4   | 45.7       |        58.1 |
> | LP-FT          |**71.6**                  |**72.9**    |**48.4**|**49.1**    |       *60.5*|
>
> For the ImageNet-validation set (ID):
>
> | ID             | ImageNet |
> |----------------|----------|
> | Fine-tuning    |**81.7**  |
> | Linear probing |  79.7    |
> | LP-FT          |**81.7**  |
>
> We are especially excited that our **theoretical insights carry over to larger scale datasets and models that we did not consider when writing the paper, and brings us to a total of 10 OOD datasets**. Thank you to all the reviewers for their time and thoughtful suggestions.
>
> Details:
> - We used a CLIP-ViT-B/16, vision transformer, around 4 times larger than a ResNet-50. This is the largest publicly available CLIP model.
> - LP-FT, LP, and FT used exactly the same amount of compute. For fine-tuning we fine-tuned for 10 epochs, for LP-FT we did 5 epochs of LP and then 5 epochs of FT. For linear probing we ran 10 epochs of linear probing.
> - We swept over three learning rates for each method, and early stopped and picked the best model based on ID (ImageNet Val) accuracy. The trends are the same if one uses OOD val to early stop and model select.

---

### Public Comment · ~Fábio_Ferreira1 · 2022-11-09
**What is the x-axis of Figure 2?**

With all respect, what does the x-axis of Figure 2 resemble? An axis for data - how? Thank you.

---

> ### Public Comment · ~Ananya_Kumar1 · 2023-01-20
> **x-axis is the span of the in-distribution (training data)**
>
> Thanks for the question! For this simplified figure, we assume that the in-distribution data is entirely along the x-axis. For example, in-distribution examples may be (1, 0) or (2, 0) or (-1, 0). But they must have 0 in the y-component, so they cannot be like (1,1).
>
> Formally, at the top of page 4 we define the training data, which is a matrix $X \in \mathbb{R}^{n \times d}$. In this example, $d = 2$ so the matrix has $n$ rows and $2$ columns. Each row of $X$ is a 2-dimensional data point, and we assume the second coordinate is 0. So the second column of $X$ is entirely 0. Does this help?

---

### Decision · Program_Chairs · 2022-01-20

**Decision:**

Accept (Oral)

**Comment:**

The paper provides a solid and thorough analysis to the two basic methods of fine-tuning, linear probing (LP) and fine-tuning (FT). The authors provide an important and highly interesting observation about the performance of both in and out of domain (OOD) setting. They validate the known phenomena that FT outperforms LP in the in-domain (ID) setting, but demonstrate that when tested on OOD data, LP is in fact more performant and back this observation with a theoretical and empirical analysis.
The remedy provided is also a known, yet slightly less popular technique of setting the final layer (LP) first, then finte-tuning (FT-LP). The authors provide thorough experiments showing that this technique enjoys the best of both worlds, meaning ID and OOD. I found it worth noting that during the rebuttal period the authors provided experiments on additional larger scale datasets and models and the results of the paper carried over to these new setting.
The reviews agree that the analysis provided is both interesting and novel. Even though the paper does not provide a new technique, there is a consensus that the understanding it gives on known techniques is a welcome addition to ICLR.